# Towards Solving the Gilbert-Pollak Conjecture via Large Language Models

Yisi Ke [* 1]   Tianyu Huang [* 2]   Yankai Shu [* 1]   Di He [3]   Jingchu Gai[† 4]   Liwei Wang[† 3 5]

## Abstract

The Gilbert-Pollak Conjecture (Gilbert & Pollak, 1968), also known as the Steiner Ratio Conjecture, states that for any finite point set in the Euclidean plane, the Steiner minimum tree has length at least $\sqrt{3}/2 \approx 0.866$ times that of the Euclidean minimum spanning tree (the Steiner ratio). A sequence of improvements through the 1980s culminated in a lower bound of $0.824$, with no substantial progress reported over the past three decades. Recent advances in LLMs have demonstrated strong performance on contest-level mathematical problems, yet their potential for addressing open, research-level questions remains largely unexplored. In this work, we present a novel AI system for obtaining tighter lower bounds on the Steiner ratio. Rather than directly prompting LLMs to solve the conjecture, we task them with generating rule-constrained geometric lemmas implemented as executable code. These lemmas are then used to construct a collection of specialized functions, which we call verification functions, that yield theoretically certified lower bounds of the Steiner ratio. Through progressive lemma refinement driven by reflection, the system establishes a new certified lower bound of 0.8559 for the Steiner ratio. The entire research effort involves only thousands of LLM calls, demonstrating the strong potential of LLM-based systems for advanced mathematical research.

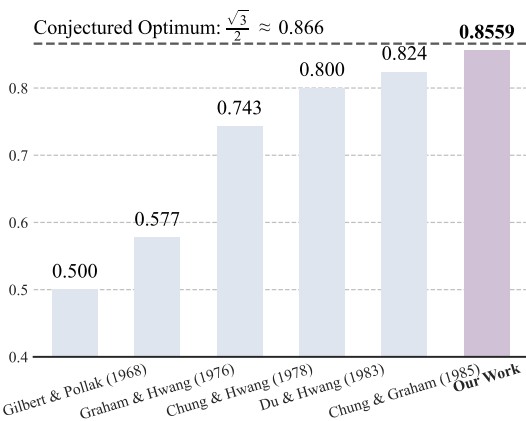

*Figure 1. Progress on Steiner Ratio Lower Bound*

## 1. Introduction

The Gilbert-Pollak Conjecture (Gilbert & Pollak, 1968) stands as a seminal open problem in combinatorial optimization and computational geometry, with profound implications for geometric network design (Kahng & Robins, 1994) and operations research. Given a finite set of points in the Euclidean plane, the Steiner Minimal Tree (SMT) is the shortest network that connects all points, allowing the introduction of additional auxiliary nodes, known as Steiner points. The Steiner ratio is the infimum of the ratio between the length of the SMT and that of the Minimum Spanning Tree (MST). Gilbert & Pollak (1968) conjectured that this ratio is lower-bounded by $\sqrt{3}/2$. Subsequent work has sought to establish tight lower bounds approaching this conjectured optimum. Remarkably, no further improvements have been reported over nearly four decades, reflecting the difficulty of the problem. See Figure 1 and the Preliminaries for a summary of the relevant literature.

Recently, Large Language Models (LLMs) have demonstrated remarkable results in solving contest-level mathematics (Achim et al., 2025; Chen et al., 2025; Chervonyi et al., 2025). A natural next step is to explore whether these capabilities can be extended to tackling unsolved mathematical conjectures, such as the Gilbert-Pollak Conjecture. However, naïvely prompting LLMs to address such tasks often yields incorrect proofs with subtle logical gaps and fundamental misunderstandings of key concepts. Unlike canonical definitions commonly found in textbooks, research-level concepts appear only sparsely in training data. Compared to

---

[*]Equal contribution [†]Co-corresponding authors. [1]School of EECS, Peking University [2]School of Mathematical Sciences, Peking University [3]State Key Laboratory of General Artificial Intelligence, Peking University, Beijing, China [4]Carnegie Mellon University, Machine Learning Department [5]Center for Machine Learning Research, Peking University. Correspondence to: Jingchu Gai <jgai@andrew.cmu.edu>, Liwei Wang <wanglw@pku.edu.cn>.

*Proceedings of the 43rd International Conference on Machine Learning*, Seoul, South Korea. PMLR 306, 2026. Copyright 2026 by the author(s).

well-structured competition problems, open research problems exhibit a vastly larger and less constrained search space. As a result, models struggle to identify valid proof strategies and become substantially more prone to hallucination.

To address this challenge, we propose a novel AI system for automating theoretical discovery in the context of the Gilbert–Pollak Conjecture. We first introduce a new mathematical tool, termed verification functions. We prove that identifying a valid collection of such functions yields a rigorous lower bound on the Steiner ratio, which serves as the system's reward signal. To facilitate the search for valid collections of verification functions, we further decompose each verification function into two parametric lemmas, whose implementations are generated by LLMs as rule-constrained executable code. We also develop a reflection process that identifies key elements limiting further reward improvements and provides this information to the LLM, guiding subsequent generations to overcome the bottleneck. An overview of the system is illustrated in Figure 2.

Our proposed system has several key advantages over general LLM-based reasoning systems. By narrowing and concretizing the search space, the LLM can focus on generating simpler, more isolated, and more fundamental lemmas in executable code, making the overall objective significantly easier than attempting to solve the conjecture end to end. The verification function then integrates these lemmas into a rigorous computation of the lower bound, ultimately determining the final outcome. Moreover, rather than relying solely on a scalar reward signal, our reflection mechanism returns mathematically structured feedback to the LLM, enabling more effective and targeted iterative refinement.

Leveraging this AI system tailored to the Gilbert–Pollak Conjecture, we successfully improve the lower bound of the Steiner ratio from 0.824—the previous state-of-the-art result established by Chung & Graham (1985)—to 0.8559. Notably, the iterative refinement process requires only about ten iterations. The entire research effort involves only thousands of LLM calls, incurring a total cost of just a few hundred dollars, thereby demonstrating the strong potential of LLM-based systems for advanced mathematical research.

Reproducible codes and the final proof can be found in https://github.com/keyisi2006/Steiner-Ratio. *If you are a mathematician interested solely in the mathematical results presented in this paper, you may proceed directly to Appendix H, where we provide a simplified version that excludes content related to LLMs.*

## 2. Preliminary

Before presenting our method, we review the background of the Gilbert-Pollak Conjecture and introduce the notation used throughout this paper.

## 2.1. Gilbert-Pollak Conjecture and Steiner Tree

We begin by formally defining the Minimum Spanning Tree and the Steiner Minimum Tree.

---

**Minimum Spanning Tree & Steiner Minimal Tree**

**Definition 1** (Minimum Spanning Tree). *Consider a set $V$ of $n$ points in the Euclidean plane $\mathbb{R}^2$. A spanning tree on $V$ is a connected, acyclic graph with vertex set $V$. When the length of each edge is defined as the Euclidean distance between its endpoints, a spanning tree that minimizes the total length is called a* Minimum Spanning Tree.

**Definition 2** (Steiner Minimal Tree). *Consider a set $V$ of $n$ points in the Euclidean plane. The shortest network interconnecting all points in $V$, where the length of each edge is measured by Euclidean distance, is necessarily a tree, referred to as a* Steiner Minimal Tree. *A Steiner Minimal Tree may contain auxiliary vertices not in $V$. These additional vertices are called* Steiner points, *while the points in $V$ are referred to as* regular points.

---

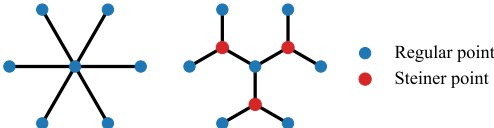

*Figure 3. Illustration of minimum spanning tree (left) and Steiner minimal tree (right).*

Based on the definitions of Minimum Spanning Tree and Steiner Minimum Tree, we formally define the Steiner ratio and verbalize Gilbert-Pollak Conjecture as follows:

---

**Steiner Ratio and Gilbert-Pollak Conjecture**

**Definition 3** (Steiner Ratio). *The Steiner ratio is defined as the infimum of the ratio of the length of the Steiner Minimal Tree to the length of the Euclidean Minimum Spanning Tree over all finite sets of points $V$:*

$$\rho_{Steiner} = \inf_V L_S(V)/L_m(V),$$

*where $L_S(V)$ and $L_m(V)$ denote the lengths of Steiner Minimal Tree and Minimum Spanning Tree, respectively.*

**Conjecture** (Gilbert-Pollak). Gilbert & Pollak (1968) famously conjectured that this ratio is lower-bounded by: $\rho_{\text{Steiner}} = \sqrt{3}/2 \approx 0.866$.

---

**Progress on the Steiner Ratio Lower Bound.** Gilbert & Pollak (1968) originally conjectured that $\rho_{\text{Steiner}} = \sqrt{3}/2$. Given the intractability of a direct proof, extensive research has historically focused on tightening the lower bound.

*Figure 2. Illustration of our proposed LLM Math Research System for the Gilbert-Pollak Conjecture*

This bound was successively improved from an initial 0.5 (Moore, as reported in Gilbert & Pollak (1968)) to 0.577 (Graham & Hwang, 1976), 0.743 (Chung & Hwang, 1978), 0.8 (Du & Hwang, 1983), and finally 0.824 (Chung & Graham, 1985). Remarkably, however, progress has stalled entirely since 1985. The primary obstacle lies in the combinatorial explosion of topological structures required for analysis, which far exceeds the capacity of human derivation.

## 2.2. Mathematical Framework for Proving Steiner Ratio Lower Bound

In this section, we briefly describe the mathematical framework for proving lower bound for Steiner ratio. Many of the mathematical details can be found in Appendix B.1.

We first give a very brief and informal description. The framework for establishing the Steiner ratio lower bound is mathematical induction (Du & Hwang, 1983). Assuming that for any set of points with cardinality at most $n - 1$, the Steiner ratio is lower-bounded by $\rho$. To extend this to a set $V$ of $n$ points, the core strategy involves a reduction argument: identifying a specific subset $V' \subset V$ of size $n - r$ (where $r \geq 1$) such that the marginal gain in the Steiner tree length satisfies $\Delta L_S / \Delta L_m \geq \rho$. Here, $\Delta L_S = L_S(V) - L_S(V')$ and $\Delta L_m = L_m(V) - L_m(V')$ denote the reduction in the lengths of the Steiner Minimal Tree and the Minimum Spanning Tree, respectively. Since $|V'| \leq n - 1$, the inductive hypothesis guarantees $L_S(V')/L_m(V') \geq \rho$. Combining this with the marginal condition $\Delta L_S \geq \rho \Delta L_m$ completes the inductive step, proving $L_S(V)/L_m(V) \geq \rho$.

However, executing this reduction presents two formidable challenges: (1) Combinatorial Search: Identifying the optimal subset $V'$—specifically, determining the reduction size $r$ (e.g., reducing to $n - 1$, $n - 2$, etc.) and selecting the exact points—requires searching a vast combinatorial space. (2) Geometric Verification: For a chosen $V'$, proving the inequality $\Delta L_S \geq \rho \Delta L_m$ is non-trivial. It necessitates formulating novel geometric theorems and analyzing complex functional properties, such as monotonicity and extremal values over high-dimensional continuous domains.

Throughout the sequence of previous works on the lower bound, the complexity of these geometric constructions and functional analyzes has progressively escalated, eventually exceeding the limits of manual tractability. This intractability motivates our introduction of an LLM-based system to automate the discovery process.

We now present the precise mathematical framework. The crux of the induction method is to **prune** some regular points $V^* \subsetneq V$ from the SMT by performing certain operations, thereby reducing the problem size to $V' = V \setminus V^*$. The key is to prove that the Steiner ratio $\rho$ can still be maintained during these operations.

The pruning operation can be summarized into three steps: (1) *Steiner Tree Pruning*: Remove a subset of edges $S^-$ from the SMT such that $V^*$ becomes disconnected from the rest. (2) *Spanning Tree Connection*: Restore connectivity between $V^*$ and the remaining points by adding a set of edges $t^*$ between regular points, which can be viewed as part of a spanning tree. (3) *Steiner Tree Reconstruction*: Connect the remaining points using a new Steiner tree $S^+$, thereby forming a complete Steiner tree.

We refer to the tuple $\tau = (V^*, S^-, S^+, t^*)$ arising from this pruning procedure as a **splitting**; a formal definition is provided in Appendix B.1.2.

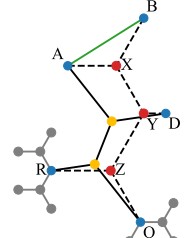

In this pruning process, the decrease in the Steiner tree length is given by $\Delta L_S \geq L_{S^-} - L_{S^+}$, while the increase in the spanning tree length is given by $\Delta L_m \leq L_{t^*}$. Here, $L_X$ denotes the total edge length of graph $X$. The requirement for the induction to hold is that $\Delta L_S / \Delta L_m \geq \rho$, which is equivalent to $\rho \cdot L_{t^*} + L_{S^+} - L_{S^-} \leq 0$. If this condition holds, then combining it with the inductive hypothesis $L_S(V')/L_m(V') \geq \rho$ allows us to derive $L_S(V)/L_m(V) \geq \rho$, thereby completing the induction step. Therefore, we can reformulate the verification of a candidate

*Figure 4. Illustration of Pruning Process. We try to prune $V^* = \{B\}$. Black dashed lines are $S^-$. Black solid lines are $S^+$. Green line is $t^*$. Yellow points are Steiner points in $S^+$.*

lower bound $\rho$ for the Steiner ratio as the minimax condition in Theorem 4. Let $\mathcal{W} = [0, +\infty)^n$ denote the parameter space of edge lengths; see Definition 12 in Appendix B.1.1 for its precise construction.

---

**Sufficient Condition for Lower Bound**

**Theorem 4.** *For any splitting $\tau = (V^*, S^-, S^+, t^*)$ and geometric configuration $\boldsymbol{w} \in \mathcal{W}$, we define the splitting function $F_\tau(\boldsymbol{w}, \rho)$ as:*

$$F_\tau(\boldsymbol{w}, \rho) = \rho \cdot L_{t^*} + L_{S^+} - L_{S^-}.$$

*Let $\mathcal{F} = \{F_\tau : \tau \text{ is a splitting}\}$ denote the collection of all splitting functions. The lower bound $\rho_{Steiner} \geq \rho$ is established if:*

$$\max_{\boldsymbol{w} \in \mathcal{W}} \min_{F \in \mathcal{F}} F(\boldsymbol{w}, \rho) \leq 0. \tag{1}$$

*We say that a value $\rho$ is feasible if the condition equation 1 holds.*

---

In the following sections, we introduce an LLM-based mathematical framework designed to address this minimax problem.

# 3. LLM Math System for the Conjecture

To effectively leverage LLMs for the Gilbert–Pollak Conjecture, we must systematically address three fundamental challenges: (1) Search Space. Spanning graph theory and computational geometry, the conjecture entails a search space far larger than that of contest-level mathematics, leading to severe hallucinations when LLMs are applied naïvely. We therefore need to construct a compact search space that constrains LLMs to generate reasonable reasoning trajectories with high probability. (2) Rigorous Verification. We require the system to produce correct derivations and rigorously verified lower bounds, which serve as a quantitative reward for evaluating the LLM's generations. (3) Iterative Reflection. As demonstrated by recent advances like DeepSeek-R1 (Guo et al., 2025), reflection plays a critical role in solving complex problems. We therefore need to build a mechanism designed to identify bottlenecks in intermediate outputs, enabling the model to reflect and iteratively refine its reasoning to obtain improved results.

To address these challenges, we first introduce a class of *verification functions* and theoretically show that, given a suitable collection, the corresponding lower bound can be computed efficiently. Consequently, obtaining improved bounds is equivalent to constructing richer and more informative valid verification functions, providing a tractable and systematic solution for Challenge (2).

Furthermore, the verification function provides significant structural advantages for leveraging LLMs. Specifically, we show that any verification function can be decomposed into

two parameterized lemmas. Rather than tasking the model with solving the conjecture directly, we cast the generation objective as producing rule-constrained implementations of these lemmas, expressed as executable code, which significantly simplifies the generation space, thereby effectively addressing Challenge (1).

Finally, we develop a bottleneck region-based reflection mechanism. Leveraging the reward signal and the verification process, we introduce an algorithm that automatically identifies key bottleneck regions limiting further improvements of the lower bound and feeds this information back to the LLM in natural language, guiding subsequent generations to overcome these bottlenecks. Experiments show that this signal significantly improves the effectiveness of LLMs, thereby addressing Challenge (3).

---

**Overview & Section Roadmap**

An overview of our system is illustrated in Figure 2. In Section 3.1, we begin by formally defining the *verification function* and establishing a method to compute lower bounds based on this formulation. The lower bound serves as the reward model of the system. In Section 3.2, we demonstrate that the search for a verification function is theoretically equivalent to identifying specific classes of lemmas. Leveraging this insight, we design an LLM-driven agent to automatically generate and filter these lemmas. Finally, Section 3.3 details the iterative interaction between the agent and the reward model with a *reflection mechanism*.

---

## 3.1. Reward Model

The general objective of the reward in the LLM is to evaluate the generated output. In our context, the reward is to determine the tightest lower bound for the Steiner ratio from the "generated proof". Theorem 4 establishes that the statement "the Steiner ratio is lower bounded by $\rho$" is equivalent to verifying that $\max_{\boldsymbol{w} \in \mathcal{W}} \min_{F \in \mathcal{F}} F(\boldsymbol{w}, \rho) \leq 0$. However, directly evaluating this continuous-domain minimax problem is computationally intractable.

To make verification tractable, we employ a Branch-and-Bound strategy that recursively partitions the continuous parameter space $\mathcal{W}$ into smaller hyperrectangles, as illustrated in Figure 5 for the 2D case. For each hyperrectangle, our objective is to identify a single function $f \in \mathcal{F}$ that ensures $f(\boldsymbol{w}) \leq 0$ globally within the subdomain. How-

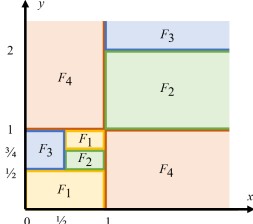

*Figure 5.   Branch-and-Bound on 2D Plane*

ever, direct verification is infeasible due to the continuous nature of the hyperrectangle. To address this challenge, we introduce verification functions, which serve as upper bounds for the underlying splitting functions while satisfying specific structural constraints. We formally define these functions as follows.

---

**Verification Function**

**Definition 5.** *A function $f : \mathcal{W} \to \mathbb{R} \cup \{+\infty\}$ is called a **verification function** if it satisfies the following two conditions:*

1. ***Shape Constraint:** The restriction of $f$ to any line parallel to a coordinate axis is unimodal (first non-increasing and then non-decreasing).*

2. ***Bounding Constraint:** There exists a splitting function (defined in Theorem 4) $g \in \mathcal{F}$ such that $g(\boldsymbol{w}) \leq f(\boldsymbol{w})$, $\forall \boldsymbol{w} \in \mathcal{W}$.*

*We denote the set of all verification functions by $\mathcal{F}_{ver}$.*

---

*Remark:* The central challenge in verifying $f(\boldsymbol{w}) \leq 0$ throughout each hyperrectangle is that exhaustive point-wise checking is infeasible in a continuous domain. The following theorem addresses this challenge by establishing that the maximum value of our verification function is attained at the vertices. This property effectively reduces the verification problem to a finite set of vertex checks. We now state the theorem as follows:

---

**Vertex Maximum Property**

**Theorem 6.** *Let $H \subset \mathcal{W}$ be a bounded axis-aligned hyperrectangle and let $f$ be any function in $\mathcal{F}_{ver}$. A key property is that the global maximum of $f$ over $H$ is attained at one of its vertices. Consequently, if*

$$\max_{\boldsymbol{w} \in \mathcal{V}(H)} f(\boldsymbol{w}) \leq 0, \qquad (2)$$

*then $f(\boldsymbol{w}) \leq 0$ for all $\boldsymbol{w} \in H$, which implies that the underlying splitting function $g(\boldsymbol{w}) \leq 0$. Note that $\mathcal{V}(H)$ is the set of vertices of $H$.*

---

Leveraging these theoretical properties of the verification function, we construct a reward model to compute the lower bound of the Steiner ratio. We present the core algorithm below, deferring the full pseudocode and detailed explanations to Appendix F.1.

**Algorithm for Reward Model.** To determine the maximal valid lower bound, the reward model employs a binary search, using a feasibility oracle grounded in Theorem 6. This oracle recursively partitions the continuous parameter space $\mathcal{W}$ into hyperrectangles and certifies each region via the following steps:

(1) *Region Partitioning.* We recursively partition the infinite parameter space into hyperrectangle regions. (2)*Vertex*

*Verification.* For bounded regions, we select a verification function $f$ and invoke the Vertex Maximum Property. By verifying $f(\boldsymbol{w}) \leq 0$ at all vertices, we mathematically guarantee that the inequality holds for every point $\boldsymbol{w}$ within the region. (3) *Monotonicity Check for Unbounded Regions.* For unbounded regions, we first verify that $f$ is non-increasing along those unbounded dimensions, ensuring the maximum value occurs at the finite lower boundary (a "slice" of the region). We then apply vertex verification to this bounded slice. Details of the monotonicity check are provided in Appendix F.2. (4) *Refinement via Subdivision.* Unverified regions are subdivided along one dimension into two halves, and recursively attempt verification.

The procedure terminates when either all regions are certified within the region number threshold, in which case the candidate $\rho$ is declared *feasible*, or when the budget is exhausted with remaining uncertified regions, in which case $\rho$ is reported as *infeasible*.

### 3.2. LLM Agent

Identifying effective verification functions that yield tight lower bounds remains challenging, as it requires deep geometric intuition, creative lemma construction, and careful analysis of function properties. Human mathematicians may struggle to systematically search and explore potentially useful geometric lemmas for this task. To address this challenge, we develop an LLM agent to automate and scale this discovery process.

We demonstrate that finding a verification function reduces to identifying two types of simpler geometric lemmas: (1) the Trapped Regular Point Lemma and (2) the Valid 4-Point Steiner Tree Lemma. (This equivalence is illustrated in Figure 8 and formally proven in Theorem 20, Appendix C.2). Within our framework, the LLM generates candidate lemmas in the form of structured code snippets. This approach significantly simplifies the generation process and facilitates rigorous verification. In the subsequent sections, we detail how we leverage the LLM to discover these lemmas.

#### 3.2.1. FINDING TRAPPED REGULAR POINT LEMMA

In this section, we detail the methodology for discovering Trapped Regular Point Lemmas using LLMs. We define two distinct categories of rules: Type A rules encode necessary properties that any Steiner Minimal Tree (SMT) must satisfy, while Type B rules specify sufficient conditions that guarantee the existence of an implicit regular point. Given the vast combinatorial space of rule interactions—which renders manual exploration intractable (see Appendix G for details)—we employ an LLM integrated with local tools (e.g., Mathematica). By iteratively validating hypotheses, the LLM discovers effective rule combinations and synthesizes conditions that imply regularity. These conditions

are subsequently simplified by the model to derive novel geometric lemmas.

**Theoretical Foundation.** Consider traversing a path on the Steiner tree. At each intermediate Steiner point, we consistently select the branch corresponding to a $60°$ clockwise turn ("turning right"). This traversal traces a maximal chain of vertices $A_0, A_1, \ldots, A_n$, where $A_n$ is a regular point and $A_1, \ldots, A_{n-1}$ are Steiner points.

Due to the geometric constraints established in Lemma 11, every internal angle $\angle A_{i-1} A_i A_{i+1}$ along this chain is exactly $120°$. We abstract this configuration as the **Steiner Spiral Chain**, as illustrated in Figure 6. For analysis, we denote the edge lengths as $a_i = |A_{i-1} A_i|$ for $i = 1, \ldots, n$, and normalize the system such that $a_1 = 1$.

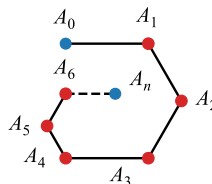

*Figure 6. The Steiner Spiral Chain structure*

Crucially, this chain is non-intersecting and constitutes the Steiner Minimal Tree for the set of vertices $\{A_0, \ldots, A_n\}$. Our goal is to derive conditions under which $A_n$ is trapped inside a bounded polygon. We construct a discrete **Structured Reasoning Space** consisting of axiomatic templates. We formalize the constraints of the Steiner Spiral using two sets of predicates, denoted $\mathcal{C}^A$ and $\mathcal{C}^B$, which serve as the atomic actions for the LLM agent.

---

**Type A/B Condition**

**Lemma 7** (Type A: Intrinsic Properties). *For any $0 \le u < s \le v \le n$, the inequality $|A_u A_v| \ge a_s$ (denoted $C^A_{u,v,s}$) is always satisfied.*

**Lemma 8** (Type B: Sufficient Trapping Conditions). *For $0 \le u < s \le v < \min(6, n)$, let $H$ be the orthogonal projection of $A_u$ onto line $A_v A_{v+1}$. We define the predicate $C^B_{u,v,s}$ as (1) $H$ lies on ray $A_v A_{v+1}$ and (2) $|A_u H| < a_s$. If $C^B_{u,v,s}$ holds, then $A_n$ is trapped inside the polygon $A_u \ldots A_v H A_u$.*

---

These predicates correspond to algebraic inequalities on $w$ (e.g., $C^B_{0,2,1} \iff (a_2 \le 1) \land (\sqrt{3}/2 \cdot (1 + a_2) \le 1)$). Full derivations are in Appendix C.1.

---

**Trapped Regular Point Theorem**

**Theorem 9.** *Let $\{C^A_i\}_{i=1}^k \subseteq \mathcal{C}^A$ and $\{C^B_j\}_{j=1}^l \subseteq \mathcal{C}^B$ be subsets of conditions. Suppose a linear constraint $C$ on $w$ satisfies the implication: $C \land \left( \bigwedge_{i=1}^k C^A_i \right) \implies \bigvee_{j=1}^l C^B_j$. Then, if $w$ satisfies $C$, the regular point $A_n$ is trapped inside the corresponding polygonal region.*

---

**LLM-Guided Lemma Generation.** This module operationalizes Theorem 9 through the synthesis of valid constraints via an LLM agent. Here, we delineate the process of guiding the LLM to identify trapped regular point lemmas. Comprehensive details are available in Appendix G.

● *Preparation: Geometric Definition.* We begin by providing the LLM with the geometric definition of the coordinate system and the structure of the polygonal chain.

● *Constraint Selection.* Since reasoning over the complete sets $\mathcal{C}^A$ and $\mathcal{C}^B$ is computationally intractable, we supply the LLM with the full sets of Type A and Type B conditions, and task the LLM with selecting a concise subset of relevant conditions (typically 1–3 items). The goal is to identify constraints that collectively enforce the trapping condition.

● *Derivation and Simplification.* The agent first employs Mathematica's `Reduce` function to compute the feasible region of $w$ that satisfies the implication $\bigwedge C^A_i \implies \bigvee C^B_j$. However, the raw region derived by Mathematica is often complex and non-linear, rendering it incompatible with our reward model. Consequently, we task the LLM with simplifying this region into explicit *linear constraints*, followed by a verification step using Mathematica.

---

**Derivation & Verification Prompt**

**Step 4.1: Attempt Derivation**
Run the following logic in Mathematica using the expressions copied from the list:
```
(* Define the logic: ForAll tail_vars,
(Steiner_Constraints) implies (Trap_Goal) *)
Reduce[
    ForAll[{a5, a6},
        a5 > 0 && a6 > 0 && <Selected_A_Props>,
        <Selected_B_Props>
    ] && a2 > 0 && a3 > 0 && a4 > 0,
    {a2, a3, a4}
]
```
*Tip*: If the result is non-linear, relax it to a linear inequality that still covers the Box.

**Step 4.2: Verification**
Verify your derived condition `<Cond>`:
```
Reduce[
    !ForAll[{a5, a6}, a5 > 0 && a6 > 0 &&
<Selected_A_Props>, <Selected_B_Props>]
    && a2 > 0 && a3 > 0 && a4 > 0 && <Cond>,
    {a2, a3, a4}
]
```
The output MUST be `False`.

---

● *Coding and Output.* Finally, we instruct LLM to encapsulate the derived constraints into two code snippets: boolean function `X_cond` checks if $w$ satisfies the derived linear constraints, while floating-point function `AX_upper_bound` computes the distance upper bound from $A_0$ to trapped $A_n$. Then they are transformed into verification functions and added to $\mathcal{F}_{\text{ver}}$. We present an example of LLM-generated

Trapped Regular Point Lemma, accompanied by its corresponding structured code snippet, below.

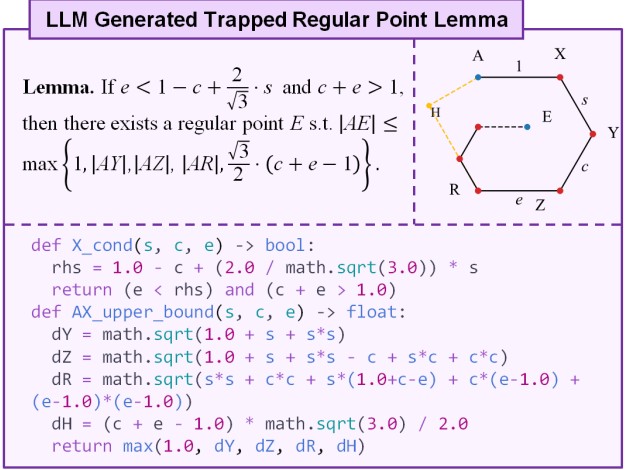

**LLM Generated Trapped Regular Point Lemma**

**Lemma.** If $e < 1 - c + \frac{2}{\sqrt{3}} \cdot s$ and $c + e > 1$, then there exists a regular point $E$ s.t. $|AE| \leq \max\left\{1, |AY|, |AZ|, |AR|, \frac{\sqrt{3}}{2} \cdot (c + e - 1)\right\}$.

```
def X_cond(s, c, e) -> bool:
    rhs = 1.0 - c + (2.0 / math.sqrt(3.0)) * s
    return (e < rhs) and (c + e > 1.0)
def AX_upper_bound(s, c, e) -> float:
    dY = math.sqrt(1.0 + s + s*s)
    dZ = math.sqrt(1.0 + s + s*s - c + s*c + c*c)
    dR = math.sqrt(s*s + c*c + s*(1.0+c-e) + c*(e-1.0) +
(e-1.0)*(e-1.0))
    dH = (c + e - 1.0) * math.sqrt(3.0) / 2.0
    return max(1.0, dY, dZ, dR, dH)
```

### 3.2.2. FINDING VALID 4-POINT STEINER TREE LEMMA

In this section, we focus on discovering valid 4-point Steiner tree lemmas. Our goal is to identify regions in the parameter space where a specific topology is guaranteed to exist and be computable. Analogous to Section 3.2.1, our core strategy involves constructing a structured reasoning space, within which the LLM performs reasoning and deduction.

**Theoretical Foundation.** Consider four points $A$, $B$, $C$, and $D$ forming a convex quadrilateral in clockwise order. We focus on the $(AB)$-$(CD)$ topology, defined as follows:

**Steiner Tree Type**

**Definition 10** (Steiner Tree Type). *The $(AB)$-$(CD)$ type Steiner tree consists of two Steiner points $S$ and $T$, and five edges: $(A, S), (B, S), (S, T), (T, C), (T, D)$. The edges incident to each Steiner point meet at $120°$ angles, as illustrated in Figure 7.*

We adopt the Valid 4-Point Steiner Tree Theorem from Du et al. (1987) to determine the existence of the $(AB)$-$(CD)$ topology. The theorem's details are provided in Appendix B.2 as Theorem 16. The LLM agent's task is to translate the geometric conditions of the theorem into algebraic constraints on the parameter space $w$, simplifying them to satisfy the orthogonally convex property required by our reward model for subsequent conversion into verification functions.

*Figure 7.* $(AB)$-$(CD)$ type Steiner tree.

**LLM-Guided Lemma Generation.** The task of the LLM agent is to transform the existence conditions of the Steiner tree involving four specific points into a form that can be used for verification functions. The process proceeds as follows:

• *Translation:* Given a specific set of four terminals, the LLM translates the geometric conditions of Theorem 16 into algebraic constraints on the parameters $w$.

• *Simplification:* Following the translation step, the resulting raw algebraic constraints are often complex and non-convex. To address this, the LLM performs a heuristic simplification to identify a tractable constraint set that implies the original complex conditions.

• *Verification:* Finally, the LLM must verify that the resulting condition is *orthogonally convex* and implies the validity conditions. Upon validation, new verification functions with $S^+$ being the specific topology will be added to $\mathcal{F}_{\text{ver}}$.

• *Coding and Output.* The LLM is instructed to encapsulate the derived constraints into two code snippets: boolean function `steiner_cond` checks if $w$ satisfies the derived constraints, while floating-point function `steiner_length` computes the 4-point Steiner tree length. Then they are transformed into verification functions and added to $\mathcal{F}_{\text{ver}}$. We present an example of an LLM-generated 4-Point Steiner Tree Lemma, accompanied by its corresponding structured code snippet, below.

**LLM Generated 4-Point Steiner Tree Lemma**

**Lemma.** If $c \geq 1$ and $e \geq 1$, then $(AD)$-$(QR)$ type Steiner tree is valid.

```
def steiner_cond(c, e) -> bool:
    return (c >= 1.0) and (e >= 1.0)
def steiner_length(c, d, s, e) -> float:
    S = 1.0 + s + c + e + 2.0*d
    return math.sqrt(S*S - S + 1.0)
```

### 3.3. Iterative Process

In this subsection, we describe the iterative closed-loop interaction between the LLM agent and the reward model. While self-reflection mechanisms have proven pivotal in solving contest-level math problems (Jaech et al., 2024; Singh et al., 2026; Lai et al., 2026)—enabling models to review generation history, detect errors, and iteratively refine accuracy—they are not directly applicable here. In our system, the objective is not to solve the Gilbert-Pollak conjecture directly, but to synthesize intermediate lemmas. Consequently, generic reflection strategies are insufficient. Instead, we design a targeted supervisory signal that explicitly identifies areas for improvement, guiding the LLM to overcome specific bottlenecks during the refinement process.

Once the LLM generates lemmas, they are converted into

verification functions to compute the current lower bound. To iteratively tighten this bound, the reward model attempts to certify an incremented candidate $\rho + \delta$. If the verification process fails, leaving a set of uncertified subregions $\{B_1, \ldots, B_k\}$, the model performs a geometric abstraction by computing the minimal axis-aligned hyperrectangle $B^\star$ that encloses them. We refer to $B^\star$ as the bottleneck region. This region is converted into a structured prompt by explicitly providing the LLM with the coordinate intervals of each dimension in $B^\star$. Consequently, the LLM is instructed to generate lemmas specifically targeting this region, thereby enabling the construction of new verification functions to close the gap. We now formally describe the entire iterative process incorporating this bottleneck reflection mechanism.

• **Evaluate.** The reward model evaluates the feasible lower bound using the current set of verification functions.

• **Reflection.** A bottleneck region is extracted from the verification process and fed back to the LLM agent.

• **Propose.** Guided by the current bottleneck region, the LLM agent proposes new lemmas targeting the limiting part of the parameter space to overcome the current optimization bottleneck.

• **Translate.** After the proposed lemmas are manually validated for correctness, they are translated into verification functions that can be consumed by the reward model, resulting in an augmented set of verification functions.

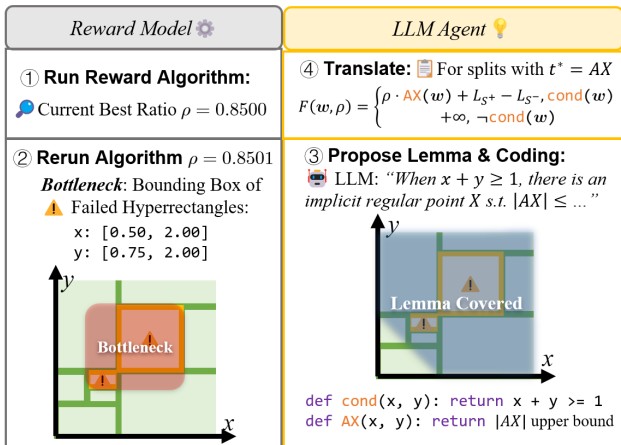

*Figure 8.* Interaction Between Reward Model & LLM Agent

**Verification of the Proof Correctness.** We ensure the reliability of our results through a dual-verification strategy. First, the LLM acts solely as a reasoning engine, offloading all validation tasks to Mathematica to perform exact computations. This significantly reduces hallucinations in the generated lemmas. Second, we performed independent manual verification of the generated proofs to confirm that the overall logical argument is sound. Consequently, the final lower bound is a standard mathematical proof that stands

independent of the generative AI pipeline.

# 4. Experiments and Results

We evaluate our system along several key dimensions: lemma quality, robustness across different LLM backbones, the role of reflection in iterative reasoning, and the final improvement of the Steiner ratio lower bound.

**Lemma Quality.** Across approximately $100$ rounds of experiments, more than $80\%$ of LLM-proposed lemmas are effective (both mathematically correct and helpful in improving the lower bound for the current step) when instantiated as verification functions. This high success rate highlights the advantages of operating within *narrow reasoning spaces* and using a *localized, lemma-driven approach*, rather than attempting end-to-end optimization over the full parameter space.

**Robustness Across LLM Backbones.** We conducted experiments using GPT-5 and Gemini 3 Pro. In all cases, the LLMs successfully proposed valid lemmas, and after roughly a dozen iterative rounds, the system consistently converged to a Steiner ratio lower bound of approximately $0.8559$. These results demonstrate that the framework is robust and largely insensitive to the choice of LLM backbone.

**Reflection Ablation.** To evaluate the importance of the reflection mechanism, we performed ablation experiments where the reflection bottleneck was removed. In this setting, the system failed to improve the lower bound over $10$ iterative rounds, confirming that reflection is essential for guiding lemma discovery and enabling improvement.

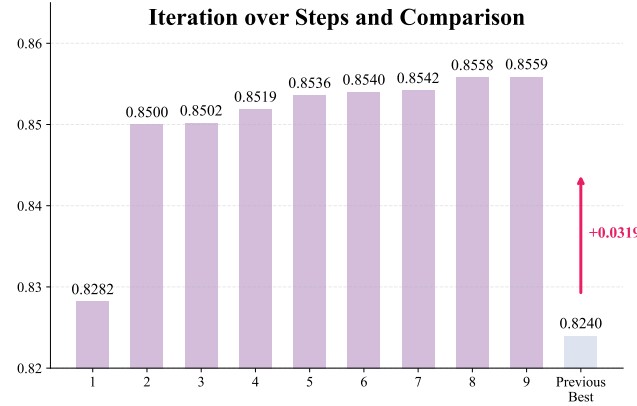

*Figure 9. Illustration of the ratio change throughout the iteration over steps and comparison with best results of previous works.*

**Typical Trajectory and Final Result.** Figure 9 summarize a representative trajectory of the system. Through iterative interaction between the LLM agent and the reward model, the lower bound progressively improves and stabilizes at $0.8559$. Importantly, this bound has been rigorously verified

across the entire parameter space, and all lemmas proposed along this trajectory have been manually checked for correctness, ensuring the validity of the overall proof independent of the specific generative AI pipeline. Across this trajectory, the LLM consumed approximately 0.5M tokens ($\sim$ 35K per round) and 4.6 hours of reasoning, while computations of the reward model required roughly 11.7 hours. The detailed content of each step of improvement can be found in the Appendix E.

## 5. Conclusion

We present an LLM-based system designed to advance the Gilbert-Pollak Conjecture. By guiding the LLM to generate structured geometric lemmas and instantiating them as verification functions, the system iteratively refines its reasoning in a localized search space, producing a significant improvement over the previous lower bound for the Steiner ratio. Our experiments demonstrate that this approach is robust across different LLM backbones and relies on only a modest number of model calls, highlighting its potential as a scalable paradigm for leveraging LLMs in math research.

Two design principles in our system, we believe, transfer to other mathematical-discovery problems.

**Construct a structured reasoning space.** Rather than tasking the LLM with end-to-end proofs, we decompose the search into a compositional space of parameterized lemmas. The LLM proposes and simplifies candidates; symbolic tools verify each one. This division of labor plays to the LLM's strength in creative search while delegating formal correctness to tools that cannot hallucinate.

**Reflect on localized verifier failures.** We summarize each verifier failure into a structured reflection signal—a minimal sub-region of the search space where the current set of lemmas is insufficient—and feed it back to the LLM. This search–verify–reflect loop should generalize to any setting that admits an exact verifier and whose failures can be localized to a bounded sub-problem.

## 6. Limitations and Future Work

We highlight two limitations of the current framework, each pointing to a concrete direction for future work.

**Symbolic verification does not scale gracefully.** Our verifier relies on Cylindrical Algebraic Decomposition (CAD), whose cost grows rapidly with the number of variables and the algebraic degree of the constraints. As a consequence, we adopt a conservative policy in which high-complexity 4-point lemmas are excluded from regimes with unbounded parameters, since CAD over variable parameters becomes intractable in those cases. Extending the inductive structure to more edges and points—or moving to 5-point or

larger Steiner-tree topologies—would amplify this bottleneck. Lighter-weight certificates (e.g., interval arithmetic, sum-of-squares relaxations, or learned proof sketches that invoke CAD only on hard sub-regions) are a natural next step.

**Human verification remains in the loop.** Although the final lower bound is mathematically certified by symbolic tools, every LLM-proposed lemma is manually checked for correctness before being instantiated as a verification function. The framework is therefore not fully autonomous. Reducing this dependence—via formal proof assistants such as Lean, or via stronger automated lemma checkers—is required for end-to-end autonomous mathematical discovery.

## Acknowledgements

LW is supported by National Science Foundation of China (NSFC92470123, NSFC62276005) and the State Key Laboratory of General Artificial Intelligence.

## Impact Statement

This paper advances the use of machine learning for open mathematical research. The cost of our experiments is modest—on the order of a few hundred dollars per discovery—suggesting that LLM-assisted theoretical research is accessible to researchers without large compute budgets. We note two broader considerations. First, as autonomous reasoning agents are scaled to many problems in parallel, their aggregate compute and energy footprint becomes non-trivial and deserves explicit accounting. Second, the paradigm of LLMs proposing structured artifacts that are then checked by exact verifiers is dual-use across any domain with a strong verifier (formal theorem proving, software verification, program synthesis); we encourage future work to remain transparent about which steps are mechanically verified and which still rely on human judgment.

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

# Appendix

## Contents

# A. Related Works

**The Gilbert-Pollak Conjecture.** Originally proposed by Gilbert & Pollak (1968), this conjecture posits that the Steiner ratio is $\rho = \sqrt{3}/2$. Beyond its theoretical importance in combinatorial optimization, the conjecture holds significant implications for **Very Large Scale Integration (VLSI)** design and **network optimization** (Kahng & Robins, 1994; Du & Hu, 2008). While Gilbert and Pollak verified the conjecture for $n = 3$, subsequent work confirmed it for specific cases: $n = 4$ by Pollak (1978), $n = 5$ by Du et al. (1985), and $n = 6$ by Rubenstein & Thomas (1991). regarding the general lower bound for any $n$, significant efforts have pushed the value from an initial 0.5 to 0.57 (Graham & Hwang, 1976), 0.74 (Chung & Hwang, 1978), 0.8 (Du & Hwang, 1983), and finally to 0.824 (Chung & Graham, 1985). However, improving this bound further has remained an open challenge since 1985 due to the prohibitive complexity of the search space.

**Solving Math Problems with LLMs.** Recent advancements in LLMs have demonstrated remarkable potential in mathematical reasoning, extending from standard benchmarks to competition-level challenges. In specialized domains, Trinh et al. (2024) introduced AlphaGeometry, a neuro-symbolic system capable of solving IMO geometry problems without human demonstrations. To tackle broader formal reasoning, HyperTree Proof Search (Lample et al., 2022) pioneered the integration of Transformer-based policies with Monte-Carlo Tree Search (MCTS). Building on this direction, recent works leverage Reinforcement Learning (RL) within formal environments: AlphaProof (Google DeepMind, 2024) achieves silver-medal standards in Lean, while DeepSeek-Prover (Xin et al., 2024; 2025) employs RL to establish state-of-the-art performance in open-ended theorem proving. To enhance reasoning reliability, InternLM-Math (Ying et al., 2024) leverages process supervision to verify intermediate steps. On the infrastructure front, Yang et al. (2023) developed LeanDojo, an open environment enabling large-scale training of these systems. Most recently, Goedel-Prover (Lin et al., 2025; 2026) was introduced as a frontier model, achieving top performance among open-weight models on the MiniF2F benchmark.

**AI for Scientific Discovery.** Artificial Intelligence has catalyzed paradigm shifts across various scientific disciplines. In biology, AlphaFold (Jumper et al., 2021; Senior et al., 2020; Krokidis et al., 2025) revolutionized structural biology by predicting protein structures with atomic accuracy. In materials science, Merchant et al. (2023) leveraged graph neural networks to discover 2.2 million new crystals (GNoME), dramatically expanding the scale of stable materials known to humanity. Similarly, GraphCast (Lam et al., 2023) demonstrated that AI can model complex physical dynamics to outperform traditional weather forecasting. Moving towards algorithmic discovery, AlphaTensor (Fawzi et al., 2022) utilized reinforcement learning to discover matrix multiplication algorithms surpassing human-designed heuristics. More recently, the focus has shifted towards fully automated research frameworks; for instance, Lu et al. (2024); Yamada et al. (2025) proposed "The AI Scientist", where LLM-based agents independently generate ideas, execute experiments, and write manuscripts. Following this trajectory, we extend the exploratory capabilities of AI to the domain of Theoretical Mathematics, specifically targeting combinatorial optimization problems.

# B. Mathematical Preliminaries

## B.1. Background on the Gilbert-Pollak Conjecture

### B.1.1. BASIC PROPERTIES OF STEINER MINIMAL TREE

Let $V$ be a finite set of points in the Euclidean plane. We denote the lengths of the Steiner Minimal Tree (SMT) and the Minimum Spanning Tree (MST) on $V$ as $L_S(V)$ and $L_m(V)$, respectively. For an arbitrary geometric graph $G$, let $L_G$ denote the sum of the Euclidean lengths of its edges. We first recall fundamental geometric properties of SMTs.

> **Properties of Steiner Minimal Trees**
>
> **Lemma 11** (Gilbert & Pollak (1968)). *Let $S$ be a Steiner Minimal Tree for a set $V$. The following properties hold:*
> 1. *Every Steiner point in $S$ has a degree of exactly $3$, and the three incident edges meet at $120°$ angles.*
> 2. *If $|V| = n$, then $S$ contains at most $n - 2$ Steiner points. A tree with exactly $n - 2$ Steiner points is called a Full Steiner Tree (FST).*
> 3. *Any SMT can be decomposed into a union of edge-disjoint Full Steiner Trees. Consequently, it suffices to consider only Full Steiner Trees when proving lower bounds on the Steiner ratio.*

Based on Lemma 11, we restrict our analysis to Full Steiner Trees without loss of generality. A defining topological feature of a Full Steiner Tree is that all regular points (terminals) are leaves with a degree of 1. We use **induction** to establish the lower bound. The base case where $|V| \leq 4$ is shown by Pollak (1978). Thus, we may assume $|V| \geq 5$.

Let $V$ be the set of regular points, and let $S$ be $\mathrm{SMT}(V)$. To complete the induction step, we focus on a terminal substructure of $S$. Consider a leaf $A$ of maximum depth (choosing an arbitrary point as root) in $S$. Let $X$ be the parent of $A$. Since $A$ is at maximum depth, the sibling of $A$ must also be a terminal, denoted as $B$. Let $Y$ be the parent of $X$. The sibling component $D$ incident to $Y$ cannot have a height exceeding that of $X$; consequently, $D$ is necessarily either a single terminal or a Steiner point connected to two terminals.

This results in the two cases illustrated in Figure 10: in the left case, $D$ is a single regular point; in the right case, $D$ is a Steiner point connecting two regular points, $U$ and $V$. The notations $(R)$ and $(Q)$ denote the residual subtrees connected to the main structure at nodes $R$ and $Q$. The labels on the edges denote their Euclidean lengths. Because the Steiner ratio is scale-invariant, we normalize the scale of the tree such that the edge $AX$ has length 1 ($|AX| = 1$).

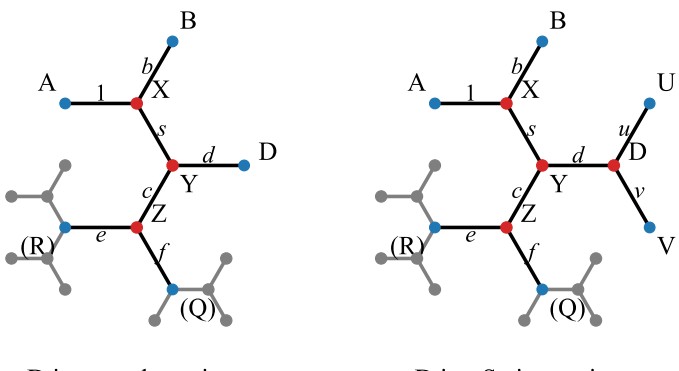

D is a regular point        D is a Steiner point

*Figure 10.* Local substructures of a Steiner Minimal Tree.

**Definition 12.** *Given a local structure in Figure 10, let $n$ be the number of edges excluding the normalized edge $AX$. The **parameter space** $\mathcal{W}$ is defined as $[0, +\infty)^n$, where the $i$-th entry represents the length of the $i$-th edge. (Therefore $\boldsymbol{w} \in \mathcal{W}$ is something like $(b, c, d, s, \cdots)$.)*

Since we only consider full Steiner trees, for any $\boldsymbol{w} \in \mathcal{W}$, the positions of all points in the local structure are uniquely determined and depend affinely on $\boldsymbol{w}$.

B.1.2. CORE LEMMAS

In this section, we provide a detailed formulation of the induction argument underlying Theorem 4. We first formalize the notion of a *splitting*, which was introduced informally in Section 2.2. We then state the reduction argument of Du & Hwang (1983) in terms of this notion.

**Definition 13.** *Given a Steiner tree $S$ for a set of points $V$, a tuple $\tau = (V^*, S^-, S^+, t^*)$ is called a **splitting** if:*

1. *$V^* \subsetneq V$ is a non-empty subset of $V$.*

2. *$S^-$ is a subset of the edges of $S$. Let $S_{rem} = S \setminus S^-$. No point in $V^*$ is incident to $S_{rem}$.*

3. *$S^+$ is a graph (may contain auxiliary vertices not in $S$) such that $S^+ \cup S_{rem}$ forms a valid Steiner tree for $V \setminus V^*$.*

4. *$t^*$ is a forest interconnecting $V^*$ and $V \setminus V^*$ using only vertices in $V$. It must satisfy: (a) For every $x \in V^*$, there exists a path in $t^*$ connecting $x$ to some $y \in V \setminus V^*$. (b) There are no paths in $t^*$ connecting any two distinct points $y_1, y_2 \in V \setminus V^*$.*

> **Du & Hwang (1983) Lemma 3**
>
> **Lemma 14** (Du & Hwang (1983) Lemma 3). *Let $S = \mathrm{SMT}(V)$ be the Steiner Minimal Tree for a set $V$ of $n$ points. Suppose that for all point sets with fewer than $n$ points, the Steiner ratio is at least $\rho'$. If there exists a splitting $\tau = (V^*, S^-, S^+, t^*)$ satisfying $(L_{S^-} - L_{S^+})/L_{t^*} \geq \rho'$, then the Steiner ratio for $V$ also satisfies $L_S(V)/L_m(V) \geq \rho'$.*

Based on Lemma 14, we reformulate the verification of a candidate lower bound $\rho$ for the Steiner ratio as a minimax condition.

> **Sufficient Condition for the Steiner Ratio Lower Bound**
>
> **Corollary 15** (Theorem 4 restated). *For any splitting $\tau = (V^*, S^-, S^+, t^*)$ and geometric configuration $\boldsymbol{w} \in \mathcal{W}$, we define the splitting function $F_\tau(\boldsymbol{w}, \rho)$ as:*
>
> $$F_\tau(\boldsymbol{w}, \rho) = \rho \cdot L_{t^*} + L_{S^+} - L_{S^-}.$$
>
> *Let $\mathcal{F} = \{F_\tau : \tau \text{ is a splitting}\}$ denote the collection of all splitting functions. The lower bound $\rho_{Steiner} \geq \rho$ is established if:*
>
> $$\max_{\boldsymbol{w} \in \mathcal{W}} \min_{F \in \mathcal{F}} F(\boldsymbol{w}, \rho) \leq 0. \tag{3}$$
>
> *We say that a value $\rho$ is feasible if the condition equation 3 holds.*

Intuitively, equation 3 states that for *any* possible geometric configuration of the tree ($\max_{\boldsymbol{w}}$), there must exist *at least one* valid strategy for splitting the tree ($\min_F$) such that the splitting function is non-positive.

## B.2. Geometric Properties of 4-Point Steiner Trees

Let $U$ be the vertex of an equilateral triangle $\triangle ABU$ constructed such that $A, B, U$ appear in counterclockwise order. Similarly, let $V$ be the vertex of an equilateral triangle $\triangle CDV$ constructed such that $C, D, V$ appear in counterclockwise order. The existence of the $(AB)$-$(CD)$ topology is governed by the following conditions:

> **Valid 4-Point Steiner Tree Theorem**
>
> **Theorem 16** (Du et al. (1987) Theorem 6). *We say an $(AB)$-$(CD)$ type Steiner tree is valid if (1) $A, B, C$ and $D$ form a convex quadrilateral. (2) $\max(\angle UCD, \angle UDC, \angle VAB, \angle VBA) \leq 120°$. (3) The intersection angle of the diagonals, $\angle AOB$, is at most $120°$, where $O$ is the intersection of $AC$ and $BD$.*

When these conditions are met, the length of the tree follows a concise formula:

---

**Length of the $(AB)$-$(CD)$ Type Steiner Tree**

**Lemma 17.** *The length of the $(AB)$-$(CD)$ type Steiner tree is precisely given by $|UV|$, where $U$ denote the position of point $B$ after segment $AB$ is rotated counterclockwise $60°$ about point $A$ and $V$ denote the position of point $D$ after segment $CD$ is rotated counterclockwise $60°$ about point $C$.*

---

*Proof.* By construction, points $A$, $U$, $B$ are arranged in clockwise order and form an equilateral triangle, as are points $C$, $V$, $D$. Consequently, $\triangle ABU$ and $\triangle CVD$ are equilateral triangles.

Recall Definition 10. A key property of the geometry is that the angle between the segments connecting to the non-bridge endpoints at a Steiner point is $120°$. Therefore, we note that $\angle ASB = 2\pi/3$. Since $\triangle ABU$ is equilateral, $\angle AUB = \pi/3$. The sum of these opposite angles in the quadrilateral $AUBS$ is $\angle ASB + \angle AUB = 2\pi/3 + \pi/3 = \pi$. This implies that the points $A, U, B, S$ are concyclic.

Since the points are concyclic, the angle $\angle ASU$ subtends the same arc $AU$ as the angle $\angle ABU$. Thus, $\angle ASU = \angle ABU = \pi/3$. From the definition of the Steiner point $S$, the segments $SA$, $SB$, and $ST$ form $120°$ angles. Specifically, $\angle AST = 2\pi/3$. It follows that $\angle ASU + \angle AST = \pi/3 + 2\pi/3 = \pi$, which proves that the points $U, S, T$ are collinear. By a similar argument for the points $C, V, D, T$, we can show that $V, T, S$ are also collinear. Combining these results, we conclude that the four points $U, S, T, V$ lie on a single straight line in that order.

Applying Ptolemy's theorem to the cyclic quadrilateral $AUBS$, we have:

$$|AB| \cdot |US| = |AU| \cdot |BS| + |BU| \cdot |AS|.$$

Since $\triangle ABU$ is equilateral, $|AU| = |BU| = |AB|$. Substituting these into the equation and dividing by $|AB|$ yields:

$$|US| = |AS| + |BS|.$$

Similarly, applying Ptolemy's theorem to the cyclic quadrilateral $CVDT$, we obtain:

$$|VT| = |CT| + |DT|.$$

As the points $U, S, T, V$ are collinear, the length of the segment $|UV|$ is the sum of the lengths of the constituent segments:

$$|UV| = |US| + |ST| + |TV| = (|AS| + |BS|) + |ST| + (|CT| + |DT|).$$

This sum is precisely the total length of the edges in the $(AB)$-$(CD)$ type Steiner tree. $\qquad\square$

# C. Our Theoretical Contributions

This appendix consolidates the original theoretical contributions of our work. Our primary objective is to elaborate on how we transform the computationally intractable minimax problem $\max_w \min_F F(w, \rho) \leq 0$ (from Theorem 4 of the main text) into a mathematically rigorous and tractable verification pipeline.

## C.1. The Type A and B Predicate Framework

> **Type A and B Predicate Framework**
>
> **Lemma 7** (restated). *For any $0 \leq u < s \leq v \leq n$, $|A_u A_v| \geq a_s$.*

*Proof.* We prove the lemma by contradiction. Assume for the sake of contradiction that $|A_u A_v| < a_s$. We can then construct a new Steiner tree by removing the edge $A_s A_{s-1}$ from the original tree and adding the edge $A_u A_v$. This operation results in a new, valid Steiner tree whose total length is less than that of the original, since an edge of length $a_s$ was replaced by one of length $|A_u A_v|$. This contradicts the assumption that the original tree is a minimal Steiner tree. $\square$

> **Type A and B Predicate Framework**
>
> **Lemma 8** (restated). *For any $0 \leq u < s \leq v < \min(6, n)$, let $H$ be the foot of the perpendicular from $A_u$ to line $A_v A_{v+1}$. If $H$ lies on the ray $A_v A_{v+1}$ and $|A_u H| < a_s$, then $A_n$ will be trapped inside polygon $A_u A_{u+1} \ldots A_v H A_u$.*

*Proof.* First, we need to prove a more specific form of this lemma. To this end, we introduce the following lemma:

**Lemma 18.** *Consider a simple polygonal chain $X_1 X_2 \ldots X_n$ in the plane, formed by $n$ segments connected end-to-end, satisfying the following conditions:*

1. *The angle between any two consecutive segments $X_i X_{i+1}$ and $X_{i+1} X_{i+2}$ is $120°$.*

2. *When traversing the chain in the order $X_1, X_2, \ldots, X_n$, every turn at each vertex $X_i$ is a right turn.*

*If $n \geq 7$ and the ray starting from $X_6$ through $X_7$ intersects one of the segments $X_1 X_2, X_2 X_3, \ldots, X_4 X_5$, then all subsequent vertices $X_7, X_8, \ldots, X_n$ lie within the closed region $\Omega$ enclosed by the segments $X_1 X_2, X_2 X_3, X_3 X_4, X_4 X_5, X_5 X_6$ and the line passing through $X_6$ and $X_1$.*

*Proof.* We establish a complex coordinate system by setting the segment $X_1 X_2$ on the real axis. Let each point $X_i$ be represented by its corresponding complex number, also denoted by $X_i$.

The vector from $X_i$ to $X_{i+1}$ can be expressed as $X_{i+1} - X_i$. Due to the right-turn condition, each subsequent vector is rotated. This gives us the relation:

$$X_{i+1} - X_i = \lambda_i \cdot \varepsilon^{i-1},$$

where $\varepsilon = e^{-\pi i/3}$ represents a $120°$ right turn, and $\lambda_i > 0$ is the length of the segment $|X_i X_{i+1}|$.

From this, the position of any vertex $X_i$ can be written as a sum:

$$X_i = X_1 + \sum_{k=1}^{i-1} \lambda_k \varepsilon^{k-1}.$$

Based on this formula, in conjunction with the condition that the ray $X_6 X_7$ intersects one of the segments in the initial chain $\{X_1 X_2, \ldots, X_4 X_5\}$, we can deduce the imaginary parts of the first few vertices:

$$\operatorname{Im} X_4 = \operatorname{Im} X_5 < \operatorname{Im} X_6 < \operatorname{Im} X_7 < \operatorname{Im} X_1 = \operatorname{Im} X_2 = 0.$$

Then for the real parts, since $X_1 \ldots X_n$ does not intersect itself ,we have:

$$\operatorname{Re} X_7 < \operatorname{Re} X_3$$

These inequalities imply that the ray $X_7 X_8$ must intersect either segment $X_2 X_3$ or $X_3 X_4$. This geometrically places $X_8$ inside the region $\Omega$, so $X_8 \in \Omega$.

We can repeat this process. The position of $X_9$ relative to $X_8$ follows the same rotation. The conditions will imply that the ray $X_8 X_9$ must also intersect one of the boundary segments, which in turn implies $X_9 \in \Omega$. For example, for $X_{10}$, the ray $X_9 X_{10}$ will intersect $X_3 X_4$ or $X_4 X_5$, leading to $X_{10} \in \Omega$.

By induction, we conclude that all subsequent vertices $X_k$ for $k \geq 7$ lie within $\Omega$. $\qquad\square$

Returning to the proof of the main lemma, in the following argument, we will use the techniques from Lemma 18.

Without loss of generality, we can assume $u = 0$. Let $\Omega$ be the disk centered at $A_0$ with radius $a_s$. According to the lemma's hypothesis, the point $H$ lies on the ray $A_v A_{v+1}$ and $|A_0 H| < a_s$, so the point $H$ is inside the disk $\Omega$. If the point $H$ lies on the line segment $A_v A_{v+1}$, then similar to the proof of Lemma 7, this would contradict the definition of a Minimal Steiner Tree. Therefore, the point $H$ must lie on the extension of the ray $A_v A_{v+1}$. Let $\mathcal{D}_1$ be the closed region enclosed by the polygon $A_u A_{u+1} \ldots A_v H A_u$, and let $\Gamma_1 = \partial \mathcal{D}_1$. For some $r > s > 0$ to be determined, we establish a complex coordinate system with the origin at $A_{v+1}$ and the positive real axis along the ray $A_{v+1} A_v$. Let

$$
X_k = \begin{cases} s + \sum_{i=1}^{6-k} r\omega^i, & 1 \leq k \leq 6 \\ A_{v+k-6}, & 7 \leq k \leq n+6-v \end{cases}
$$

where $\omega = e^{\pi i/3}$, and any undefined summation part is taken as $0$. Then $\operatorname{Im} X_7 = \operatorname{Im} X_6 = \operatorname{Im} X_1 = 0$. Let $\mathcal{D}_2$ be the closed region enclosed by the polygon $X_1 X_2 \ldots X_7$, and let $\Gamma_2 = \partial \mathcal{D}_2$. Note that since $v < 6$, $\mathcal{D}_1$ is contained in the upper half-plane $\mathbb{H}$. Therefore, we can choose $r - s, s$ sufficiently large such that $\mathcal{D}_1 \subseteq \mathcal{D}_2$.

We prove by induction on $m$ that $A_{v+m-6} = X_m \in \mathcal{D}_1$ (for $7 \leq m \leq n+6-v$), and the ray $X_{m-1} X_m$ intersects one of the closed segments $X_{m-6} X_{m-5}$ or $X_{m-5} X_{m-4}$.

For the base case $m = 7$, it is clear that $A_{v+1} = X_7 \in \mathcal{D}_1$. Since $\operatorname{Im} X_7 = \operatorname{Im} X_6 = \operatorname{Im} X_1 = 0$, the ray $X_6 X_7$ intersects the closed segment $X_1 X_2$.

Assume the conclusion holds for $m = k < n+6-v$. Consider the case $m = k+1$:

By the induction hypothesis, $A_{v+k-6} = X_k \in \mathcal{D}_1$. If the segment $X_k X_{k+1} = A_{v+k-6} A_{v+k-5}$ intersects the polygonal chain $X_1 \ldots X_k$, then because $X_k \in \mathcal{D}_1 \subseteq \mathcal{D}_2$, the segment $X_k X_{k+1}$ must intersect $\Gamma_1$. This contradicts the fact that $A_0 \ldots A_n$ is non-self-intersecting. Therefore, the segment $X_k X_{k+1}$ does not intersect the polygonal chain $X_1 \ldots X_k$. Thus, similar to the proof of lemma 18, by the induction hypothesis, since the ray $X_{k-1} X_k$ intersects the closed segment $X_{k-6} X_{k-5}$ or $X_{k-5} X_{k-4}$, the ray $X_k X_{k+1}$ must intersect the closed segment $X_{k-5} X_{k-4}$ or $X_{k-4} X_{k-3}$. Hence, $X_{k+1} \in \mathcal{D}_2$. Since $\Gamma_1$ divides $\mathcal{D}_2$ into two connected components, $\mathcal{D}_1$ and $\overline{(\mathcal{D}_2 - \mathcal{D}_1)}$, if $X_{k+1} \in (\mathcal{D}_2 - \mathcal{D}_1)$, then the segment $X_k X_{k+1}$ would intersect $\Gamma_1$, which contradicts the non-self-intersecting property of $A_0 \ldots A_n$. Therefore, $X_{k+1} \in \mathcal{D}_1$. By the principle of induction, we conclude that $A_n = X_{n+6-v} \in \mathcal{D}_1$.

$\qquad\square$

Full list of Type A and Type B Conditions when $n = 6$ can be found in Table 1 and Table 2.

*Table 1.* Type A conditions when $n = 6$.

| Type A Conditions | |
|---|---|
| **Condition** | **Description** |
| $C_{0,1,s}^A$ | $1 \geq a_s$ |
| $C_{0,2,s}^A$ | $1 + a_2 + a_2^2 \geq a_s^2$ |
| $C_{0,3,s}^A$ | $1 + a_2 + a_2^2 - a_3 + a_2 a_3 + a_3^2 \geq a_s^2$ |
| $C_{0,4,s}^A$ | $a_2^2 + a_3^2 + a_2(1 + a_3 - a_4) + a_3(-1 + a_4) + (-1 + a_4)^2 \geq a_s^2$ |
| $C_{0,5,s}^A$ | $1 + a_2^2 + a_3^2 - 2a_4 + a_4^2 + a_2(1 + a_3 - a_4 - 2a_5) + a_3(-1 + a_4 - a_5) - a_5 + a_4 a_5 + a_5^2 \geq a_s^2$ |

| | |
|---|---|
| $C_{0,6,s}^A$ | $1 + a_2^2 + a_3^2 - 2a_4 + a_4^2 - a_5 + a_4 a_5 + a_5^2 + a_3(-1 + a_4 - a_5 - 2a_6) + a_2(1 + a_3 - a_4 - 2a_5 - a_6)$ $+ a_6 - a_4 a_6 + a_5 a_6 + a_6^2 \geq a_s^2$ |
| $C_{1,2,s}^A$ | $a_2 \geq a_s$ |
| $C_{1,3,s}^A$ | $a_2^2 + a_2 a_3 + a_3^2 \geq a_s^2$ |
| $C_{1,4,s}^A$ | $a_2^2 + a_3^2 + a_2(a_3 - a_4) + a_3 a_4 + a_4^2 \geq a_s^2$ |
| $C_{1,5,s}^A$ | $a_2^2 + a_3^2 + a_4^2 + a_2(a_3 - a_4 - 2a_5) + a_3(a_4 - a_5) + a_4 a_5 + a_5^2 \geq a_s^2$ |
| $C_{1,6,s}^A$ | $a_2^2 + a_3^2 + a_4^2 + a_4 a_5 + a_5^2 + a_3(a_4 - a_5 - 2a_6) + a_2(a_3 - a_4 - 2a_5 - a_6) - a_4 a_6 + a_5 a_6 + a_6^2 \geq a_s^2$ |
| $C_{2,3,s}^A$ | $a_3 \geq a_s$ |
| $C_{2,4,s}^A$ | $a_3^2 + a_3 a_4 + a_4^2 \geq a_s^2$ |
| $C_{2,5,s}^A$ | $a_3^2 + a_4^2 + a_3(a_4 - a_5) + a_4 a_5 + a_5^2 \geq a_s^2$ |
| $C_{2,6,s}^A$ | $a_3^2 + a_4^2 + a_5^2 + a_3(a_4 - a_5 - 2a_6) + a_4(a_5 - a_6) + a_5 a_6 + a_6^2 \geq a_s^2$ |
| $C_{3,4,s}^A$ | $a_4 \geq a_s$ |
| $C_{3,5,s}^A$ | $a_4^2 + a_4 a_5 + a_5^2 \geq a_s^2$ |
| $C_{3,6,s}^A$ | $a_4^2 + a_5^2 + a_4(a_5 - a_6) + a_5 a_6 + a_6^2 \geq a_s^2$ |
| $C_{4,5,s}^A$ | $a_5 \geq a_s$ |
| $C_{4,6,s}^A$ | $a_5^2 + a_5 a_6 + a_6^2 \geq a_s^2$ |
| $C_{5,6,s}^A$ | $a_6 \geq a_s$ |

*Table 2.* Type B conditions when $n = 6$.

## Type B Conditions

| Condition | Description |
|---|---|
| $C_{0,1,s}^B$ | False |
| $C_{0,2,s}^B$ | $(1 - a_2)/2 \geq 0 \wedge (\sqrt{3}/2 \cdot (1 + a_2) \leq a_s)$ |
| $C_{0,3,s}^B$ | $(2 + a_2 - a_3)/2 \geq 0 \wedge (\sqrt{3}/2 \cdot (a_2 + a_3) \leq a_s)$ |
| $C_{0,4,s}^B$ | $(1 + 2a_2 + a_3 - a_4)/2 \geq 0 \wedge (\sqrt{3}/2 \cdot (-1 + a_3 + a_4) \leq a_s)$ |
| $C_{0,5,s}^B$ | $(-1 + a_2 + 2a_3 + a_4 - a_5)/2 \geq 0 \wedge (\sqrt{3}/2 \cdot (-1 - a_2 + a_4 + a_5) \leq a_s)$ |
| $C_{1,2,s}^B$ | False |
| $C_{1,3,s}^B$ | $(a_2 - a_3)/2 \geq 0 \wedge (\sqrt{3}/2 \cdot (a_2 + a_3) \leq a_s)$ |
| $C_{1,4,s}^B$ | $(2a_2 + a_3 - a_4)/2 \geq 0 \wedge (\sqrt{3}/2 \cdot (a_3 + a_4) \leq a_s)$ |
| $C_{1,5,s}^B$ | $(a_2 + 2a_3 + a_4 - a_5)/2 \geq 0 \wedge (\sqrt{3}/2 \cdot (-a_2 + a_4 + a_5) \leq a_s)$ |
| $C_{2,3,s}^B$ | False |
| $C_{2,4,s}^B$ | $(a_3 - a_4)/2 \geq 0 \wedge (\sqrt{3}/2 \cdot (a_3 + a_4) \leq a_s)$ |
| $C_{2,5,s}^B$ | $(2a_3 + a_4 - a_5)/2 \geq 0 \wedge (\sqrt{3}/2 \cdot (a_4 + a_5) \leq a_s)$ |
| $C_{3,4,s}^B$ | False |
| $C_{3,5,s}^B$ | $(a_4 - a_5)/2 \geq 0 \wedge (\sqrt{3}/2 \cdot (a_4 + a_5) \leq a_s)$ |
| $C_{4,5,s}^B$ | False |

The proofs of Lemma 7 (necessary properties) and Lemma 8 (sufficient conditions) in this section establish the formal geometric foundation for discovering Trapped Regular Point Lemmas. These predicates constitute one of the two primary paths for systematically constructing new verification functions in our framework.

## C.2. Proof of Theorems in Section 3

**Theorem 6** (restated). Let $H \subset \mathcal{W}$ be a bounded axis-aligned hyperrectangle and let $f$ be any function in $\mathcal{F}_{\text{ver}}$. A key property is that the global maximum of $f$ over $H$ is attained at one of the vertices. Consequently, if

$$\max_{\boldsymbol{w} \in \mathcal{V}(H)} f(\boldsymbol{w}) \le 0, \tag{4}$$

then $f(\boldsymbol{w}) \le 0$ for all $\boldsymbol{w} \in H$, which implies that the underlying splitting function $g(\boldsymbol{w}) \le 0$.

*Proof.* Since the set of vertices $\mathcal{V}(H)$ is finite, the value $M = \max_{\boldsymbol{v} \in \mathcal{V}(H)} f(\boldsymbol{v})$ is well-defined. Consider the sublevel set of $f$ at level $M$:
$$\mathcal{L}_M = \{\boldsymbol{w} \in H \mid f(\boldsymbol{w}) \le M\}.$$
By the definition of $\mathcal{F}_{\text{ver}}$, $\mathcal{L}_M$ is an orthogonally convex set. By our definition of $M$, it is clear that all vertices of the hyperrectangle are contained in this set, i.e., $\mathcal{V}(H) \subseteq \mathcal{L}_M$.

To prove that $f(\boldsymbol{w}_0) \le M$ for any $\boldsymbol{w}_0 \in H$, it suffices to show that $H \subseteq \mathcal{L}_M$. We rely on the following geometric lemma:

**Lemma 19.** *Let $S \subseteq \mathbb{R}^n$ be an orthogonally convex set and $H \subset \mathbb{R}^n$ be a bounded axis-aligned hyperrectangle. If $\mathcal{V}(H) \subseteq S$, then $H \subseteq S$.*

*Proof of Lemma.* We proceed by induction on the dimension $n$.

- **Base Case ($n = 1$):** $H$ is an interval $[a, b]$. Since $S$ is orthogonally convex, its intersection with the line containing $H$ is a connected interval. Since the endpoints (vertices) $a, b \in S$, the entire interval $[a, b]$ must be contained in $S$.

- **Inductive Step:** Assume the statement holds for dimension $n - 1$. Let $H = [a, b] \times H'$, where $H'$ is an $(n-1)$-dimensional hyperrectangle. The vertices of $H$ are the union of $\{a\} \times \mathcal{V}(H')$ and $\{b\} \times \mathcal{V}(H')$.

  Consider the slice of $S$ at $x_1 = a$, defined as $S_a = \{\boldsymbol{y} \in \mathbb{R}^{n-1} \mid (a, \boldsymbol{y}) \in S\}$. Since $S$ is orthogonally convex in $\mathbb{R}^n$, the slice $S_a$ is orthogonally convex in $\mathbb{R}^{n-1}$.

  Since $\{a\} \times \mathcal{V}(H') \subseteq \mathcal{V}(H) \subseteq S$, it follows that $\mathcal{V}(H') \subseteq S_a$. By the inductive hypothesis, $H' \subseteq S_a$. This implies that the entire face $\{a\} \times H'$ is contained in $S$. By the same logic, the face $\{b\} \times H'$ is also contained in $S$.

  Now, take any point $\boldsymbol{w} = (x_1, \boldsymbol{w}') \in H$. Consider the line segment connecting $(a, \boldsymbol{w}')$ and $(b, \boldsymbol{w}')$. This segment is parallel to the first coordinate axis. Since the endpoints lie in the faces identified above, they are both in $S$. By the orthogonal convexity of $S$, the entire segment is in $S$, implying $\boldsymbol{w} \in S$.

$\square$

Applying the lemma to $\mathcal{L}_M$, we conclude that $H \subseteq \mathcal{L}_M$. Therefore, for every $\boldsymbol{w}_0 \in H$, $f(\boldsymbol{w}_0) \le M$. If $M \le 0$, it immediately follows that $f(\boldsymbol{w}_0) \le 0$ everywhere in $H$. $\square$

**Theorem 20.** *A series of $F \in \mathcal{F}_{ver}$ can be derived by establishing one of the following two types of lemmas:*

1. ***Trapped Regular Point Lemma:*** *A proposition asserting that when the parameters $\boldsymbol{w}$ satisfy certain **linear** constraints, an implicit regular point is guaranteed to lie within a specific bounded polygonal region.*

   *Formally, let $v_0$ denote the trapped regular point in the lemma. Then for each splitting $\tau = (V^*, S^-, S^+, t^*)$, we can derive a verification function $\tilde{F}_\tau$ from the splitting function $F_\tau$, as long as $t^*$ contains $v_0$ as an endpoint of an edge.*

2. ***Valid 4-Point Steiner Tree Lemma:*** *A proposition asserting that when $\boldsymbol{w}$ satisfies certain **orthogonally convex** conditions, a specific 4-point Steiner tree structure is valid.*

   *Formally, let $X_1, X_2, X_3, X_4$ denote the 4 points in the lemma. Then for each splitting $\tau = (V^*, S^-, S^+, t^*)$, we can derive a verification function $\tilde{F}_\tau$ from the splitting function $F_\tau$, as long as $S^+ = SMT(X_1, X_2, X_3, X_4)$.*

*Proof.* Our objective is to explain how a series of verification functions $F \in \mathcal{F}_{\text{ver}}$ can be derived from these two types of lemmas. Notice that both linear constraints and orthogonally convex conditions restrict the parameter vector $\boldsymbol{w}$ to an orthogonally convex set $\Omega$.

Consequently, for the first type of lemma, let the trapped regular point be denoted by $v_0$. We can consider the set $t^*$ that contains $v_0$ as an endpoint of an edge and its corresponding splitting function:

$$F_\tau(\boldsymbol{w}, \rho) = \rho \cdot L_{t^*} + L_{S^+} - L_{S^-}.$$

Since $v_0$ is confined within a certain polygonal region, we can derive an upper bound $\widetilde{L}_{t^*}$ for $L_{t^*}$ by estimating the upper bounds of the edges associated with $v_0$. Next, we examine whether the function

$$\widetilde{F}_\tau(\boldsymbol{w}, \rho) = \rho \cdot \widetilde{L}_{t^*} + L_{S^+} - L_{S^-}$$

satisfies the definition of a verification function, where the underlying splitting function is $F_\tau(\boldsymbol{w}, \rho)$. If it does, we have successfully obtained a new verification function.

For the second type of lemma, let the vertices of the 4-point Steiner tree be $X_1, X_2, X_3, X_4$. We can consider the set $S^+$ that includes this Steiner tree and its corresponding splitting function:

$$F_\tau(\boldsymbol{w}, \rho) = \rho \cdot L_{t^*} + L_{S^+} - L_{S^-}.$$

Analogous to the treatment of the first type of lemma, when $\boldsymbol{w} \in \Omega$, we can derive a new verification function by estimating an upper bound $\widetilde{L}_{S^+}$ for $L_{S^+}$ and then checking if

$$\widetilde{F}_\tau(\boldsymbol{w}, \rho) = \rho \cdot L_{t^*} + \widetilde{L}_{S^+} - L_{S^-}$$

satisfies the definition of a verification function. By Lemma 17 and the fact that the coordinates of $U$ and $V$ in the statement of the lemma are all affine functions of $\boldsymbol{w}$, we conclude that $\widetilde{L}_{S^+}$ is convex. Restricting the convex function to the orthogonally convex set $\Omega$ yields a verification function. □

---

**Theorem 9** (restated). Let $\{C_i^A\}_{i=1}^k \subseteq \mathcal{C}^A$ and $\{C_j^B\}_{j=1}^l \subseteq \mathcal{C}^B$ be subsets of conditions. Suppose a linear constraint $C$ on $\boldsymbol{w}$ satisfies the implication: $C \wedge \left( \bigwedge_{i=1}^k C_i^A \right) \implies \bigvee_{j=1}^l C_j^B$. Then, if $\boldsymbol{w}$ satisfies $C$, the regular point $A_n$ is trapped inside the corresponding polygonal region.

---

*Proof.* This follows directly from the definitions of Lemma 7 and Lemma 8. □

This section proves two cornerstone components of our theoretical framework:

1. **Theorem 6 (Vertex-Maximization Property)** is the foundation of our entire verification strategy. It fundamentally addresses the challenge of computational tractability by reducing the verification problem over a continuous parameter domain to a check on its discrete vertices[1].

2. **Pathways for Constructing New Verification Functions:** The efficacy of Theorem 6 relies on a sufficiently rich set of verification functions, $\mathcal{F}_{\text{ver}}$. Our framework systematically augments this set by establishing two types of lemmas:

   - **Trapped Regular Point Lemma:** Theorem 9 formally establishes this pathway. It leverages the A/B predicates from Section C.1 to transform the search for such lemmas into a formal logical deduction task.
   - **Valid 4-Point Steiner Tree Lemma:** This second pathway is designed to handle more complex 4-point splits. Based on Theorem 16, it operates by identifying additional geometric constraints that ensure the corresponding complex splitting function satisfies the shape constraints required by Theorem 6.

---

[1] The algorithm implementing this vertex-checking strategy is provided in Appendix F.1.

## C.3. Proof of Convexity for Splitting Functions

In this section, we provide the proof that for any split where the component $S^+$ connects a set of terminals of size at most 3 (i.e., $|S^+| \leq 3$), the resulting splitting function $F(\boldsymbol{w}, \rho)$ is a convex function of the geometric parameters $\boldsymbol{w}$.

Recall the definition of the splitting function:

$$F(\boldsymbol{w}, \rho) = \rho \cdot L_{t^*}(\boldsymbol{w}) + L_{S^+}(\boldsymbol{w}) - L_{S^-}(\boldsymbol{w}).$$

We analyze the convexity of each term individually. Since we directly use edge lengths as parameters, the coordinates of points in the local substructure are affine functions of the parameter vector $\boldsymbol{w}$.

### C.3.1. LINEARITY OF $L_{S^-}$

The term $L_{S^-}$ represents the sum of the lengths of the edges removed from the original Steiner tree. In our local parameterization, the edge lengths of the initial tree $S$ are directly used as the parameters (or linear combinations thereof). Consequently, $L_{S^-}(\boldsymbol{w})$ is an affine function of $\boldsymbol{w}$ and therefore $-L_{S^-}(\boldsymbol{w})$ preserves convexity.

### C.3.2. CONVEXITY OF $L_{t^*}$

The term $L_{t^*}$ represents the length of the connector graph $t^*$, which consists of edges connecting specific pairs of points from $V$. Let $\boldsymbol{u}(\boldsymbol{w})$ and $\boldsymbol{v}(\boldsymbol{w})$ be the coordinates of two such points. The Euclidean distance between them is:

$$d(\boldsymbol{u}, \boldsymbol{v}) = \|\boldsymbol{u}(\boldsymbol{w}) - \boldsymbol{v}(\boldsymbol{w})\|_2.$$

The Euclidean norm $\| \cdot \|_2$ is a convex function and $\boldsymbol{u}(\boldsymbol{w}) - \boldsymbol{v}(\boldsymbol{w})$ is affine in $\boldsymbol{w}$. The composition of a convex function with an affine mapping is convex. Thus, the distance between any pair of points is convex in $\boldsymbol{w}$. Since $L_{t^*}$ is a sum of such distances (with $\rho \geq 0$), $\rho \cdot L_{t^*}(\boldsymbol{w})$ is convex.

### C.3.3. CONVEXITY OF $L_{S^+}$ FOR $|S^+| \leq 3$

The term $L_{S^+}$ denotes the length of the Steiner Minimal Tree for a subset of boundary points $V' \subset \mathbb{R}^2$. We consider the cases where $|V'| \leq 3$.

**Case 1:** $|V'| \leq 2$. If $|V'| = 2$, $L_{S^+}$ is simply the Euclidean distance between the two points. If $|V'| < 2$, the length is 0. In both cases, the convexity follows the same logic as Section C.3.2.

**Case 2:** $|V'| = 3$. Let $V' = \{A, B, C\}$. The length of the Steiner Minimal Tree is obtained by introducing at most one Steiner point $q$. The length function is defined by the minimization problem:

$$L_{S^+}(\boldsymbol{w}) = \inf_{\boldsymbol{q} \in \mathbb{R}^2} \left( \|A(\boldsymbol{w}) - \boldsymbol{q}\|_2 + \|B(\boldsymbol{w}) - \boldsymbol{q}\|_2 + \|C(\boldsymbol{w}) - \boldsymbol{q}\|_2 \right).$$

Let $g(\boldsymbol{w}, \boldsymbol{q}) = \|A(\boldsymbol{w}) - \boldsymbol{q}\|_2 + \|B(\boldsymbol{w}) - \boldsymbol{q}\|_2 + \|C(\boldsymbol{w}) - \boldsymbol{q}\|_2$. Note that the term $\|A(\boldsymbol{w}) - \boldsymbol{q}\|_2$ is the Euclidean norm of an affine function of the joint vector $(\boldsymbol{w}, \boldsymbol{q})$. Therefore, it is jointly convex in $(\boldsymbol{w}, \boldsymbol{q})$. Since the sum of convex functions is convex, $g(\boldsymbol{w}, \boldsymbol{q})$ is jointly convex in $\boldsymbol{w}$ and $\boldsymbol{q}$.

We invoke the standard result from convex analysis: *if $g(x, y)$ is jointly convex in $(x, y)$ and $C$ is a convex set, then the function $h(x) = \inf_{y \in C} g(x, y)$ is convex in $x$.* Applying this to our case, $L_{S^+}(\boldsymbol{w}) = \inf_{\boldsymbol{q}} g(\boldsymbol{w}, \boldsymbol{q})$ is a convex function of $\boldsymbol{w}$.

### C.3.4. CONCLUSION

The splitting function $F(\boldsymbol{w}, \rho)$ is the sum of a convex term ($\rho L_{t^*}$), a convex term ($L_{S^+}$), and an affine term ($-L_{S^-}$). Since the sum of convex functions is convex, $F(\boldsymbol{w}, \rho)$ is convex with respect to $\boldsymbol{w}$.

This section proves the convexity of a class of basic splitting functions. As these functions inherently satisfy the constraints required by Theorem 6, they serve as the initial members of our verification set $\mathcal{F}_{\text{ver}}$. This set is then subsequently augmented by new functions discovered via the two pathways outlined in the previous subsection.

## C.4. Theoretical Foundation for the Monotonicity Check

The vertex-checking strategy of Theorem 6 is not directly applicable to unbounded parameter regions. To address this case, we must first verify the function's monotonicity along any unbounded dimensions, thereby reducing the problem to one on a bounded domain. The lemmas presented in this section (Lemmas 21, 22, and 23) provide the necessary theoretical guarantees for this critical monotonicity check.

We first describe the verification procedure designed for the splits where $|S^+| \leq 3$ that do not involve implicit regular points.

### C.4.1. GEOMETRIC FOUNDATIONS

Let $x$ be the parameter representing the length of a specific edge $e$ in the tree topology. We first establish upper bounds on the partial derivatives of the length components with respect to $x$.

**Lemma 21** (Derivative of Path Length). *Let $t^*$ be a graph connecting points in $V$, and let $L_{t^*}$ be its total length. Then:*

$$\frac{\partial L_{t^*}}{\partial x} \leq \sum_{(u,v) \in t^*} \mathbb{I}(e \in path(u,v)),$$

*where $\mathbb{I}(\cdot)$ is the indicator function, and $path(u,v)$ denotes the unique path between $u$ and $v$ in the full tree structure.*

*Proof.* Let $(u,v)$ be an edge in $t^*$. Its length is the Euclidean distance $\|u(\boldsymbol{w}) - v(\boldsymbol{w})\|$. If the edge $e$ (associated with parameter $x$) does not lie on the path between $u$ and $v$ in the underlying tree, then $u$ and $v$ belong to the same rigid component with respect to $e$. Changing $x$ translates both points identically, so $\|u - v\|$ is constant, and the derivative is 0.

If $e$ lies on the path between $u$ and $v$, increasing $x$ translates one endpoint (say $v$) relative to the other ($u$) by a vector $\mathbf{d}$ with $\|\mathbf{d}\| = 1$ (the unit direction of edge $e$). The derivative is:

$$\frac{\partial \|u-v\|}{\partial x} = \langle \nabla_v \|u - v\|, \mathbf{d} \rangle = \left\langle \frac{v-u}{\|v-u\|}, \mathbf{d} \right\rangle = \cos\theta,$$

where $\theta$ is the angle between the chord $uv$ and the edge $e$. Since $\cos\theta \leq 1$, the derivative is bounded by 1. Summing over all edges in $t^*$ yields the result. $\square$

**Lemma 22** (Derivative of SMT Length). *Let $S^+$ be the Steiner Minimal Tree for a set of terminals $V'$, with $|V'| \leq 3$. Then:*

$$\frac{\partial L_{S^+}}{\partial x} \leq 1.$$

*Proof.* If $|V'| \leq 2$, $L_{S^+}$ is a single Euclidean distance (or 0), and the result follows from Lemma 21.

Consider the case $|V'| = 3$, say $V' = \{A, B, C\}$. The SMT consists of a Steiner point $q$ connecting to $A, B, C$. The length is $L = \|A - q\| + \|B - q\| + \|C - q\|$. By the Envelope Theorem, the derivative of the optimal value function with respect to a parameter is equal to the partial derivative of the objective function evaluated at the optimum. Thus, we do not need to consider the change in the optimal position of $q$. If edge $e$ separates the terminals (e.g., separating $A$ from $\{B, C\}$), increasing $x$ translates $A$ relative to the rest of the system by a unit vector $\mathbf{d}$. The derivative is:

$$\frac{\partial L}{\partial x} = \frac{\partial \|A-q\|}{\partial x} = \left\langle \frac{A-q}{\|A-q\|}, \mathbf{d} \right\rangle = \cos\theta \leq 1.$$

If $e$ does not separate the terminals, they move as a rigid body, and the derivative is 0. In all cases, the upper bound is 1. $\square$

### C.4.2. SPECIAL ANALYSIS ON THE MONOTONICITY OF $f$

We extend Lemma 22 to $S^+$ connecting a set of terminals whose size is larger than 3.

**Lemma 23** (Derivative of SMT Length with respect to $f$). *Let $S^+$ be the Steiner Minimal Tree for a set of terminals $V'$. Then:*

$$\frac{\partial L_{S^+}}{\partial f} \leq 1.$$

*Proof.* If $Q \notin V'$, then clearly $\frac{\partial L_{S+}}{\partial f} = 0$. Now we focus on the case where $Q \in V'$. Fix the values of all variables except $f$. Let $h(x)$ denote the value of $L_{S+}$ when $f = x$. We then prove that for any $0 \le x_1 < x_2$, $h(x_2) - h(x_1) \le x_2 - x_1$.

Let $Q_1$ and $Q_2$ denote the positions of point $Q$ when $f = x_1$ and $f = x_2$, respectively. Let $V''$ be the set of vertices in $V'$ excluding $Q$. Since $SMT(V'' \cup \{Q_1\})$ together with the edge $Q_1 Q_2$ forms a Steiner tree for $V'' \cup \{Q_2\}$, we have $L_S(V'' \cup \{Q_2\}) \le L_S(V'' \cup \{Q_1\}) + |Q_1 Q_2|$, which by definition implies $h(x_2) - h(x_1) \le x_2 - x_1$. $\qquad \square$

Based on Lemma 23, when verifying monotonicity with respect to variable $f$, we can apply the monotonicity check procedure to the splits generated by the LLM using the 4-Point Steiner Tree Lemmas. This forms a key foundation for the reduction strategy for unbounded $f$ in Appendix D.2.

## D. Experimental Setup

### D.1. Baseline Verification Functions

The construction of the baseline verification functions relies on a classical geometric result derived by Du & Hwang (1983). This result serves as a specific, manually proved instance of the "Trapped Regular Point" lemmas that our LLM agent aims to generalize.

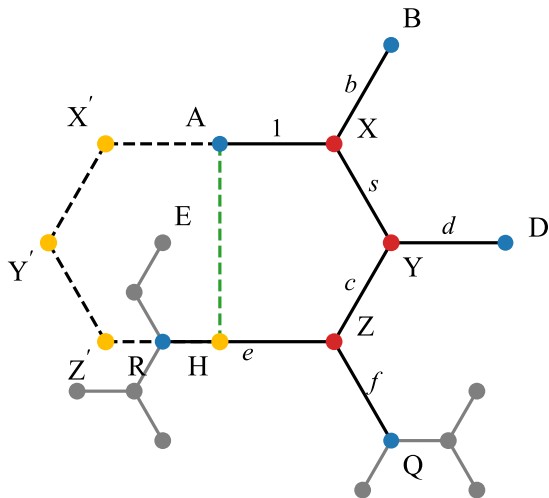

*Figure 11.* Sample of Subtree: $D$ is regular

Recall the subtree of a Steiner Minimal Tree as shown in Figure 11. Let $H$ be the orthogonal projection of $A$ on the line $ZR$. Let the region $AX'Y'Z'H$ be the mirror reflection of the region $AXYZH$ with respect to $AH$.

---

**Du & Hwang (1983) Lemma 2**

**Lemma 24** (Du & Hwang (1983) Lemma 2). *If $c \leq 1$, there must exist a regular point $E$ in the region $AXYZHZ'Y'X'$ that connects to $A$ through $Z$.*

---

**Corollary 25.** *If $c \leq 1$, there must exist an implicit regular point $E$ such that $|AE| \leq \max\{|AY|, |AZ|\}$.*

*Remark*: It is important to note that the proof of Lemma 24 does not depend on the topological status of point $A$. Consequently, this trapping property applies universally to any such chain, regardless of whether the starting node is a regular point or a Steiner point. This allows us to apply the same logic symmetrically to the node $D$, even when $D$ is a Steiner point connected to $U$ and $V$.

Due to the symmetry of the Steiner structure, a similar rule applies to the chain starting at $D$.

---

**Corollary 26.** *If $f \leq d$, there must exist an implicit regular point $F$ such that its distance to the terminal is bounded. The bound depends on whether $D$ is a regular point or a Steiner point:*

- *If $D$ is a regular point, $|DF| \leq \max\{|DZ|, |DQ|\}$.*

- *If $D$ is a Steiner point, $|VF| \leq \max\{|VY|, |VZ|, |VQ|\}$.*

---

Based on the corollaries above, the baseline set $\mathcal{F}_{\text{ver}}^{(0)}$ comprises the verification functions resulted from all splits $\tau = (V^*, S^-, S^+, t^*)$ satisfying the following criteria:

1. The component $S^+$ satisfies $|S^+| \leq 3$.

2. The set of admissible edges in $t^*$ is augmented to include the implicit connections $(A, E)$ and $(D, F)$ (or $(V, F)$). The length functions for these edges are conditional:

- Edge $(A, E)$:

$$L(A, E) = \begin{cases} \max\{|AY|, |AZ|\} & c \leq 1 \\ +\infty & c > 1 \end{cases}$$

- Edge $(D, F)$ (when $D$ is a regular point):

$$L(D, F) = \begin{cases} \max\{|DZ|, |DQ|\} & f \leq d \\ +\infty & f > d \end{cases}$$

- Edge $(V, F)$ (when $D$ is a Steiner point):

$$L(V, F) = \begin{cases} \max\{|VY|, |VZ|, |VQ|\} & f \leq d \\ +\infty & f > d \end{cases}$$

**Impact on Monotonicity.** When a split utilizes implicit edges $(A, E)$, $(D, F)$, or $(V, F)$, their length bounds depend on local geometric parameters. Establishing the required non-increasing property for these variables is non-trivial. Consequently, our system performs the monotonicity checks (as detailed in Appendix F.2) only for variables with respect to which the implicit edge lengths are constant.

**Explanation of Steiner Ratio greater than** $0.824$ **for Baseline Settings.** Though we only cited technical lemmas from Du & Hwang (1983) as the baseline, and that work only claimed a proof for a Steiner ratio lower bound of $\rho = 0.8$, our initial ratio reached $0.8282$, even surpassing the previous best-known lower bound of $0.824$ by Chung & Graham (1985). This improvement is due to (1) prior work considered smaller topological structures. Our baseline setting directly considered topologies with a maximum chain length of 4, as shown in Figure 10, while previous human experts, constrained by computational capabilities, only considered topologies with a maximum chain length of 3. This allowed us to more accurately cover the parameter space, resulting in a higher lower bound; (2) prior work relied on manual proofs considering a relatively limited set of verification functions, whereas we leveraged computational mathematics to exhaustively enumerate all possible splits; (3) previous work employed looser scaling techniques, while we utilized more precise computational algorithms, eliminating gaps caused by arithmetic tricks.

### D.2. Problem Decomposition and Search Space Reduction

During the iterative improvement phase, the LLM Agent focuses exclusively on refining the existence conditions for the implicit point $E$ (associated with the chain starting at $A$). The conditions for the symmetric implicit point $F$ (associated with the chain starting at $D$) remain fixed to the baseline criterion ($f \leq d$).

Consequently, the global verification problem is partitioned into four distinct sub-problems based on the topological status of $D$ and the domain of the parameter $f$:

1. **Case 1:** $D$ is a regular point, and $f \in [0, d]$ ($F$ exists).

2. **Case 2:** $D$ is a regular point, and $f \in (d, +\infty)$ ($F$ does not exist).

3. **Case 3:** $D$ is a Steiner point, and $f \in [0, d]$ ($F$ exists).

4. **Case 4:** $D$ is a Steiner point, and $f \in (d, +\infty)$ ($F$ does not exist).

**Reduction Strategy for Unbounded** $f$**.** For Cases 2 and 4, where $f > d$, the implicit connection $(D, F)$ (or $(U, F)$ and $(V, F)$) is invalid and thus excluded from the connector graph $t^*$. In these scenarios, we apply a strict filtering policy: we only admit splits whose splitting functions are **monotonically non-increasing** with respect to $f$.

Hence, in the region $f \geq d$, it suffices to verify the boundary $f = d$. For each function $F$, define $F^\partial$ as the reduced function obtained from $F$ by setting $f = d$ and removing $f$ from the input variables. The verification target is thus reduced to

$$\max_{\boldsymbol{w} \in \mathcal{W}^\partial} \min_{F \in \mathcal{F}} F^\partial(\boldsymbol{w}) \leq 0,$$

where $\mathcal{W}^\partial$ denotes the reduced parameter space obtained by removing the $f$-coordinate from $\mathcal{W}$, so that $\mathcal{W}^\partial = [0, +\infty)^{n-1}$.

Note that $F^\partial$ preserves both convexity and coordinatewise non-increasingness of $F$. In particular, its non-increasingness with respect to $d$ follows because increasing $d$ simultaneously increases the original $d$-entry and the substituted $f$-entry, and $F$ is non-increasing in both variables.

**Representative Regime for Evaluation.** While our system rigorously verifies all four cases to establish the global lower bound, we restrict the detailed visualization and analysis in the main text to **Case 2** ($D$ is Regular, $f \geq d$). This decision is justified by two empirical observations regarding the optimization landscape:

1. **Boundary Dominance:** The verification bottlenecks (regions where the feasible $\rho$ is minimized) predominantly concentrate at the boundary $f = d$.

2. **Topological Degeneracy:** In cases where $D$ is a Steiner point, the most critical constraints typically occur when the terminal edges ($|DU|$ and $|DV|$) approach zero length. In this limit, the structure topologically degenerates to the case where $D$ is a regular point.

Consequently, Case 2 represents the most challenging and representative slice of the parameter space. Focusing on this regime allows us to demonstrate the system's efficacy on the "hardest" instances of the problem without redundancy.

# E. Proofs of LLM-Generated Lemmas

In this section, we present the formal proofs for the geometric lemmas generated by the LLM agent during the iterative process. As discussed in Section D.2, the LLM proposes these lemmas based on the representative regime where $D$ is a **regular point** and the parameter $f$ is fixed at its boundary value $f = d$.

*Table 3. Results of Each Step*

| Results of Each Step | | | |
|---|---|---|---|
| **Step** | **Steiner Ratio** | **Proposed Lemma Type** | **Description** |
| 1 | 0.8282 | / | / |
| 2 | 0.8500 | Trapped Regular Point | $C^B_{0,4,2}$ |
| 3 | 0.8502 | Trapped Regular Point | $C^A_{2,4,3}, C^B_{0,4,3}$ |
| 4 | 0.8519 | 4-Point Steiner Tree | $(DQ)$-$(RA)$ |
| 5 | 0.8536 | 4-Point Steiner Tree | $(AD)$-$(QR)$ |
| 6 | 0.8540 | Trapped Regular Point | $C^B_{2,4,3}$ |
| 7 | 0.8542 | 4-Point Steiner Tree | $(BD)$-$(QR)$ |
| 8 | 0.8558 | 4-Point Steiner Tree | $(AD)$-$(QR)'$ |
| 9 | 0.8559 | Trapped Regular Point | $C^A_{0,5,5}, C^A_{1,5,5}, C^B_{0,5,5}$ |

To apply these lemmas to the global verification problem (spanning all four cases defined in Section D.2), we adhere to the following transplantation policies:

1. **Trapped Regular Point Lemmas (Implicit $E$):** These lemmas rely exclusively on the local geometry of the chain starting at $A$. The validity of the implication $C \implies \text{Trapped}(E)$ is independent of the topology of $D$ or the value of $f$. Therefore, these lemmas are **universally transplanted** to all four cases.

2. **4-Point Steiner Tree Lemmas:** These lemmas (e.g., verifying the existence of $(AD)$-$(QR)$ trees) were derived and symbolically verified under the strict assumption $f = d$.

   - **Cases 2 & 4 ($f \geq d$):** As per our reduction strategy, the verification functions for the unbounded domain $f \in (d, +\infty)$ are mapped to the boundary $f = d$. Thus, these lemmas are **valid and applied**.
   - **Cases 1 & 3 ($f \leq d$):** In these regimes, the geometric validity of a 4-point topology depends on the specific value of $f \in [0, d]$. Since the symbolic verification was not performed for variable $f$, we adopt a conservative policy: we **exclude** these high-complexity 4-point tree lemmas in these cases, relying instead on the baseline splits and trapped point lemmas.

Edge lengths follow the notation in Figure 11. For example, $|AY| = \sqrt{1 + s + s^2}$ and $|AZ| = \sqrt{1 + s + s^2 - c + cs + c^2}$.

> **Step 1**
>
> **Lemma 27** (Step 1, Side $A$)**.** *If $c \leq 1$, then there exists a regular point $E$ such that $|AE| \leq \max\{|AY|, |AZ|\}$.*
>
> **Lemma 28** (Step 1, Side $D$)**.** *If $f \leq d$, then there exists a regular point $F$ such that $|DF| \leq \max\{|DZ|, |DQ|\}$.*

Note that these 2 lemmas are the same as Corollary 25 and Corollary 26, respectively, when $D$ is a regular point. In particular, they are not model-generated lemmas. We restated them here for a clearer step-by-step proof.

> **Step 2**
>
> **Lemma 29** (Step 2)**.** *If $e < 1 - c + 2/\sqrt{3} \cdot s$ and $c + e > 1$, then there exists a regular point $E$ such that $|AE| \leq \max\{1, |AY|, |AZ|, |AR|, \sqrt{3}/2 \cdot (c + e - 1)\}$.*

*Proof.* If $R$ is a regular point, the statement is proved. We now assume $R$ is a Steiner point. For a Minimal Steiner Tree, this implies the existence of a subsequent edge emanating from $R$ in the clockwise direction. Let $H$ be the foot of the

perpendicular from point $A$ to the line containing this edge. Note that $C_{0,4,2}^B$ is $1+2s+c-e>0$ and $\sqrt{3}/2\cdot(-1+c+e)\le s$. We show that $C_{0,4,2}^B$ holds under the lemma conditions.

First, from $e<1-c+2/\sqrt{3}\cdot s$, we have $1+2s+c-e>1+2(\sqrt{3}/2\cdot(-1+c+e))+c-e=1-\sqrt{3}+(1+\sqrt{3})c+(\sqrt{3}-1)e$. Note that if $c<1$, then by Lemma 27 we already have $|AE|\le\max\{|AY|,|AZ|\}$, so we only need to consider the case $c\ge 1$. Thus $1+2s+c-e>1-\sqrt{3}+(1+\sqrt{3})c+(\sqrt{3}-1)e\ge 2+(\sqrt{3}-1)e>0$. Second, we have $\sqrt{3}/2\cdot(-1+c+e)<\sqrt{3}/2\cdot(2/\sqrt{3}\cdot s)=s$. Thus, $C_{0,4,2}^B$ holds. Then by Theorem 9, there exists a regular point $E$ trapped inside the polygonal region $AXYZRH$. Since $|AH|=\sqrt{3}/2\cdot(c+e-1)$ by simple calculation, we have $|AE|\le\max\{1,|AY|,|AZ|,|AR|,\sqrt{3}/2\cdot(c+e-1)\}$. $\qquad\square$

---

**Step 3**

**Lemma 30** (Step 3). *If $e<1+(2/\sqrt{3}-1)\cdot c$ and $c+e>1$, then there exists a regular point $E$ such that $|AE|\le\max\{1,|AY|,|AZ|,|AR|,\sqrt{3}/2\cdot(c+e-1)\}$.*

---

*Proof.* If $R$ is a regular point, the statement is proved. We now assume $R$ is a Steiner point. For a Minimal Steiner Tree, this implies the existence of a subsequent edge emanating from $R$ in the clockwise direction. Let $H$ be the foot of the perpendicular from point $A$ to the line containing this edge. Note that $C_{0,4,3}^B$ is $1+2s+c-e>0$ and $\sqrt{3}/2\cdot(-1+c+e)\le c$. We show that $C_{0,4,3}^B$ holds under the lemma conditions.

The first part is same as Lemma 29. For the second part, from $e<1+(2/\sqrt{3}-1)\cdot c$, we have $\sqrt{3}/2\cdot(-1+c+e)<\sqrt{3}/2\cdot((2/\sqrt{3}-1)\cdot c+c)=c$. Thus, $C_{0,4,3}^B$ holds. Then by Theorem 9, there exists a regular point $E$ trapped inside the polygonal region $AXYZRH$, so we can conclude the proof same as Lemma 29. $\qquad\square$

---

**Step 4**

**Lemma 31** (Step 4). *The region defined by the system of inequalities $\mathcal{S}$ below is an orthogonally convex set. Furthermore, for any $w$ in this region, the $(RA)$-$(DQ)$ type Steiner tree is valid.*

$$\mathcal{S}:\begin{cases} I_1: & e+c\ge 1 \\ I_2: & e\le s+c+d+2 \\ I_3: & e\ge 2-c-d+s \\ I_4: & e\le 3s \\ I_5: & (2+c^2+3d+2s+3cs+3ds+2s^2)-(2+c+3d+s)e\ge 0 \\ I_6: & 2(\vec{RD}\cdot\vec{AQ})+|\vec{RD}||\vec{AQ}|\ge 0 \\ I_7: & 2(\vec{RV}\cdot\vec{RA})+|\vec{RV}||\vec{RA}|\ge 0 \\ I_8: & 2(\vec{AV}\cdot\vec{AR})+|\vec{AV}||\vec{AR}|\ge 0 \end{cases}$$

---

*Proof.* We perform the verification in two parts: first demonstrating that these conditions imply the geometric existence of the tree, and second proving that the defined region is orthogonally convex.

We employ **Cylindrical Algebraic Decomposition (CAD)** to rigorously verify all the following inequalities.

PART 1: GEOMETRIC VALIDITY

We verify the sufficient conditions from Theorem 16. To enable algebraic verification, all angular conditions $\theta\le 120°$ are transformed into the equivalent scalar product inequality:

$$2(\vec{u}\cdot\vec{v})+|\vec{u}||\vec{v}|\ge 0.$$

**1. Convexity of Quadrilateral $RADQ$:**

- $\vec{RA}\times\vec{AD}\le 0$, $\vec{AD}\times\vec{DQ}\le 0$, and $\vec{DQ}\times\vec{QR}\le 0$: Verified unconditionally.

- $\vec{QR} \times \vec{RA} \leq 0$: Verified to hold when $c + e \geq 1$, which is explicitly enforced by inequality $I_1$.

## 2. Simpson Point Angles:

- $\angle UDQ \leq 120°$: The condition $I_2$ implies that $\vec{DU} \cdot \vec{DQ} \geq 0$ (i.e., the angle is $\leq 90°$), which sufficiently implies $\leq 120°$.

- $\angle UQD \leq 120°$: The condition $I_3$ implies that $\vec{QU} \cdot \vec{QD} \geq 0$ (i.e., the angle is $\leq 90°$), which sufficiently implies $\leq 120°$.

- $\angle VRA \leq 120°$: This is the direct geometric interpretation of inequality $I_7$.

- $\angle VAR \leq 120°$: This is the direct geometric interpretation of inequality $I_8$.

## 3. Intersection Angle:

- The condition that the angle between diagonals $RD$ and $AQ$ is $\leq 120°$ corresponds directly to inequality $I_6$.

Part 2: Orthogonal Convexity Check

We verify that each defining inequality in $\mathcal{S}$ is monotonic with respect to the coordinate axes within the feasible region.

**Linear Inequalities** $(I_1, I_2, I_3, I_4)$: These are linear combinations of variables. Linear functions are unconditionally monotonic, thus defining orthogonally convex half-spaces.

**Inequality** $I_5$: Let $g_5(\boldsymbol{w})$ be the LHS of $I_5$.

- Variables $d, e$: The function $g_5$ is linear with respect to $d$ and $e$, ensuring monotonicity.

- Variable $s$: We compute $\frac{\partial g_5}{\partial s}$. Symbolic verification confirms $\frac{\partial g_5}{\partial s} \geq 0$ whenever $e \leq s + c + d + 2$. This condition is ensured by $I_2$.

- Variable $c$: We compute $\frac{\partial g_5}{\partial c}$. Symbolic verification confirms $\frac{\partial g_5}{\partial c} \geq 0$ whenever $e \leq 3s$. This condition is ensured by $I_4$.

**Inequality** $I_6$: Let $G(\boldsymbol{w}) = \frac{\vec{RD} \cdot \vec{AQ}}{|\vec{RD}||\vec{AQ}|}$ be the cosine of the angle between the diagonals. The condition is equivalent to $G(\boldsymbol{w}) \geq -1/2$. We analyze the connectivity of the feasible region along lines parallel to each coordinate axis.

- **Variables** $c$ **and** $e$: Symbolic differentiation confirms that $\frac{\partial G}{\partial c} \leq 0$ and $\frac{\partial G}{\partial e} \geq 0$ unconditionally. Thus, $G$ is monotonic along the $c$ and $e$ axes, ensuring the feasible region is a connected interval (orthogonally convex) in these directions.

- **Variable** $s$: The symbolic engine indicates that the sign of $\frac{\partial G}{\partial s}$ depends on $c$:

$$\text{sgn}\left(\frac{\partial G}{\partial s}\right) = \begin{cases} \geq 0 & \text{if } c \geq 1 \\ \leq 0 & \text{if } c \leq 1 \end{cases}$$

To prove orthogonal convexity, consider any line $L$ parallel to the $s$-axis. On such a line, the parameters $c, d, e$ are fixed constants. Consequently, the sign of the derivative with respect to $s$ is constant along the entire line $L$. Thus, $G$ restricted to $L$ is monotonic, and the sublevel set $\{s \in L \mid G(\dots, s, \dots) \geq -1/2\}$ is a connected interval.

- **Variable** $d$: Similarly, the behavior depends on $c$:
  - If $c \geq 1$: Symbolic verification shows $\frac{\partial G}{\partial d} \geq 0$. $G$ is monotonic, so the intersection is an interval.
  - If $c < 1$: Symbolic verification confirms that $G(\boldsymbol{w}) \geq -1/2$ is satisfied for all admissible $d$. The intersection is the entire domain $[0, +\infty)$, which is a connected interval.

In all cases, the intersection with lines parallel to the $d$-axis is connected.

**Inequality** $I_7$: Let $G(\boldsymbol{w}) = \frac{\vec{RV} \cdot \vec{RA}}{|\vec{RV}||\vec{RA}|}$. The condition is equivalent to $G(\boldsymbol{w}) \geq -1/2$. We analyze the properties under the assumption that $c + e \geq 1$ (Inequality $I_1$).

- **Variables** $s, d, e$: Symbolic differentiation confirms that within the region $c + e \geq 1$:

$$\frac{\partial G}{\partial s} \leq 0, \quad \frac{\partial G}{\partial d} \leq 0, \quad \frac{\partial G}{\partial e} \geq 0.$$

  Since the function is monotonic with respect to each of these variables individually, the feasible region is orthogonally convex along these axes.

- **Variable** $c$: The behavior depends on the fixed values of $s$ and $e$:
  - If $e \leq s + 1$: Symbolic verification shows $\frac{\partial G}{\partial c} \geq 0$. The function is monotonic, so the valid set for $c$ is a connected interval.
  - If $e > s + 1$: Symbolic verification confirms that $G(\boldsymbol{w}) \geq -1/2$ holds unconditionally for all valid $c$. The valid set is the entire domain, which is a connected interval.

  Thus, for any fixed $s, d, e$, the set of valid $c$ values is connected.

**Inequality** $I_8$: Let $\boldsymbol{u} = \vec{AV}$ and $\boldsymbol{v} = \vec{AR}$. The condition is equivalent to the angle inequality $\angle(\boldsymbol{u}, \boldsymbol{v}) \leq 120°$. We verify this by analyzing the inclination angles $\phi$ and $\psi$ of $\boldsymbol{u}$ and $\boldsymbol{v}$ respectively.

**Geometric Constraint:** We explicitly verify that under the given constraints, the $y$-components of $\vec{AV}$ and $\vec{AR}$ are always non-positive. Consequently, the vectors lie in the lower half-plane, and their inclination angles $\phi, \psi$ are well-defined and restricted to the interval $[-180°, 0]$. This ensures no discontinuity in the angle definition.

The condition is thus equivalent to:

$$-120° \leq \Phi(\boldsymbol{w}) \leq 120°, \quad \text{where } \Phi(\boldsymbol{w}) = \phi(\boldsymbol{w}) - \psi(\boldsymbol{w}).$$

To prove orthogonal convexity, it suffices to show that for any axis-parallel line, $\Phi$ is monotonic (or the condition is unconditionally true). We utilize the derivative formula for the inclination angle $\phi$ of a vector $\boldsymbol{u}$:

$$\frac{\partial \phi}{\partial x} = \frac{\det(\boldsymbol{u}, \partial_x \boldsymbol{u})}{|\boldsymbol{u}|^2}.$$

- **Variable** $c$: The behavior depends on the parameters $e$ and $s$:
  - If $e \geq s + 1$: Symbolic verification shows $\frac{\partial \phi}{\partial c} \leq 0$ and $\frac{\partial \psi}{\partial c} \geq 0$. Thus, $\frac{\partial \Phi}{\partial c} \leq 0$. The difference is monotonic, so the valid set is an interval.
  - If $e < s + 1$: Symbolic verification confirms that the angle condition holds unconditionally. The valid set is the entire domain.

  Since the condition $e \geq s + 1$ is independent of $c$, any line parallel to the $c$-axis falls entirely into one of these two cases.

- **Variable** $s$: We rely on Inequality $I_1$ ($c + e \geq 1$). Symbolic verification confirms that when $c + e \geq 1$:

$$\frac{\partial \phi}{\partial s} \leq 0, \quad \frac{\partial \psi}{\partial s} \geq 0 \implies \frac{\partial \Phi}{\partial s} \leq 0.$$

  Thus, $\Phi$ is monotonic along the $s$-axis.

- **Variable** $d$: Symbolic verification shows $\frac{\partial \psi}{\partial d} = 0$. The sign of $\frac{\partial \phi}{\partial d}$ depends on the sum $c + s$:

$$\text{sgn}\left(\frac{\partial \Phi}{\partial d}\right) = \text{sgn}\left(\frac{\partial \phi}{\partial d}\right) = \begin{cases} \geq 0 & \text{if } c + s \geq 1 \\ \leq 0 & \text{if } c + s \leq 1 \end{cases}$$

  Consider any line parallel to the $d$-axis. Along this line, $c$ and $s$ are fixed constants. Thus, the derivative's sign is constant, implying $\Phi$ is monotonic along that line.

- **Variable** $e$: Symbolic verification shows $\frac{\partial \phi}{\partial e} = 0$ and $\frac{\partial \psi}{\partial e} \leq 0$. Thus, $\frac{\partial \Phi}{\partial e} \geq 0$. The function is monotonic along the $e$-axis.

**Final Conclusion**: We have demonstrated that for every inequality $I_1$ through $I_8$, the defining constraints satisfy the coordinate-wise monotonicity property (either directly or via case-based logic where the case condition is constant along the axis). Therefore, the intersection of these regions is an orthogonally convex set.

$\square$

---

**Step 5**

**Lemma 32** (Step 5). *If $c \geq 1$ and $e \geq 1$, then $(AD)$-$(QR)$ type Steiner tree is valid.*

---

*Proof.* We verify the sufficient conditions for the existence of the $(AD)$-$(QR)$ topology (Theorem 16) using the symbolic computation engine `Mathematica`. To enable algebraic verification, all angular conditions $\theta \leq 120°$ are transformed into the equivalent scalar product inequality:

$$2(\vec{u} \cdot \vec{v}) + |\vec{u}||\vec{v}| \geq 0.$$

We employ **Cylindrical Algebraic Decomposition (CAD)** to rigorously verify that the following inequalities hold under the assumptions $c \geq 1$ and $e \geq 1$:

**1. Convexity of Quadrilateral $ADQR$** (via cross products):

- $\vec{AD} \times \vec{DQ} \leq 0$ and $\vec{DQ} \times \vec{QR} \leq 0$: Verified (Unconditional).
- $\vec{QR} \times \vec{RA} \leq 0$: Verified (Requires $c \geq 1$).
- $\vec{RA} \times \vec{AD} \leq 0$: Verified (Unconditional).

**2. Simpson Point Angles** (via transformed inequality):

- $\angle UQR \leq 120°$, $\angle VAD \leq 120°$, $\angle VDA \leq 120°$: Verified (Unconditional).
- $\angle URQ \leq 120°$: Verified (Requires $e \geq 1$).

**3. Intersection Angle** (via transformed inequality):

- Angle between diagonals $AQ$ and $DR \leq 120°$: Verified (Requires $c \geq 1$).

**4. Orthogonal Convexity**: The validity region is defined by the intersection of linear inequalities. Since these are intersections of convex sets, the resulting domain is convex. $\square$

---

**Step 6**

**Lemma 33** (Step 6). *If $e < (2/\sqrt{3} - 1) \cdot c$, then there exists a regular point $E$ such that $|AE| \leq \max\{|AY|, |AZ|, |AR|\}$, and $|DE| \leq \max\{|DY|, |DZ|, |DR|, \sqrt{3/4 \cdot (c+e)^2 + 3/2 \cdot (c+e)d + d^2}\}$.*

---

*Proof.* We first analyze point $R$ under condition $C_{2,4,3}^B$. If $R$ is a regular point, the statement is proved. We therefore proceed by considering the case where $R$ is a Steiner point. For a Minimal Steiner Tree, this implies the existence of a subsequent edge emanating from $R$ in the clockwise direction. Let $H$ be the foot of the perpendicular from point $Y$ to the line containing this edge. It follows that a trapped regular point, which we denote by $E$, exists within the polygonal region $YZRH$. The distance $|AM|$ is therefore bounded by:

$$|AE| \leq \max\{|AY|, |AZ|, |AR|, |AH|\}$$

It is clear that $|AH| \leq |AR|$, a consequence of the geometric construction. The reasoning is that the projections of the segments $YX$ and $XA$ onto the line $RH$ are co-directional with the vector $\vec{RH}$.

Similarly, for the distance between $D$ and the trapped point $E$, we have the inequality:

$$|DE| \leq \max\{|DY|, |DZ|, |DR|, |DH|\}$$

A calculation yields $|YH| = \frac{\sqrt{3}}{2}(c + e)$. Given that $\angle DYH = 150°$, an application of the Law of Cosines to $\triangle DYH$ gives:

$$\begin{aligned}
|DH|^2 &= |DY|^2 + |YH|^2 - 2|DY||YH|\cos(150°) \\
&= |DY|^2 + |YH|^2 + \sqrt{3}|DY||YH| \\
&= d^2 + \frac{3}{2}d(c + e) + \frac{3}{4}(c + e)^2.
\end{aligned}$$

$\square$

**Step 7**

**Lemma 34** (Step 7). *If $d \geq b$, then $(BD)$-$(QR)$ type Steiner tree is valid.*

*Proof.* We verify the sufficient conditions for the existence of the $(BD)$-$(QR)$ topology (Theorem 16) using the symbolic computation engine `Mathematica`. As in the previous proof, all angular conditions $\theta \leq 120°$ are transformed into the equivalent scalar product inequality:

$$2(\vec{u} \cdot \vec{v}) + |\vec{u}||\vec{v}| \geq 0.$$

We employ **Cylindrical Algebraic Decomposition (CAD)** to rigorously verify that the following inequalities hold under the assumption $d \geq b$:

**1. Convexity of Quadrilateral $BDQR$ (via cross products):**

- $\vec{BD} \times \vec{DQ} \leq 0$, $\vec{DQ} \times \vec{QR} \leq 0$, $\vec{QR} \times \vec{RB} \leq 0$, and $\vec{RB} \times \vec{BD} \leq 0$: All verified to hold unconditionally within the geometric domain.

**2. Simpson Point Angles (via transformed inequality):**

- $\angle URQ \leq 120°$ and $\angle VBD \leq 120°$: Verified (Unconditional).
- $\angle UQR \leq 120°$: Verified (Requires $d \geq b$).
- $\angle VDB \leq 120°$: Verified (Requires $d \geq b$).

**3. Intersection Angle (via transformed inequality):**

- Angle between diagonals $BQ$ and $DR \leq 120°$: Verified (Unconditional).

**4. Orthogonal Convexity**: The validity region is defined by the linear inequality $d \geq b$, which is automatically convex. $\square$

**Step 8**

**Lemma 35** (Step 8). *The region defined by the system of inequalities $\mathcal{S}$ below is an orthogonally convex set. Furthermore, for any $w$ in this region, the $(AD)$-$(QR)$ type Steiner tree is valid.*

$$\mathcal{S}: \begin{cases}
I_1: & d(c + e - 1) + e(s + c) \geq 0 \\
I_2: & e \geq \frac{1}{2}\left(\sqrt{(c + s + 2d)^2 + 4d} - (c + s + 2d)\right) \\
I_3: & -(1 + s/2 - c/2 + d/2)(c/2 + e + d) + \frac{3}{4}c(s + c + d) \geq 0 \\
I_4: & s + e \geq 1 \\
I_5: & c \leq 1 + d + s/2 + e/2
\end{cases}$$

*Proof.* We perform the verification in two parts: first demonstrating that these conditions imply the geometric existence of the tree, and second proving that the defined region is orthogonally convex.

We employ **Cylindrical Algebraic Decomposition (CAD)** to rigorously verify all the following inequalities.

PART 1: GEOMETRIC VALIDITY

We verify the sufficient conditions from Theorem 16.

**1. Convexity of Quadrilateral $ADQR$:**

- The convexity conditions at vertices $A, D, Q$ are verified to hold unconditionally.

- At vertex $R$, the condition $\vec{QR} \times \vec{RA} \leq 0$ is algebraically equivalent to inequality $I_1$.

**2. Simpson Point Angles:**

- $\angle UQR \leq 120°$, $\angle VAD \leq 120°$, and $\angle VDA \leq 120°$: Verified unconditionally.

- $\angle URQ \leq 120°$: Solving the algebraic inequality corresponding to this angle condition yields exactly inequality $I_2$.

**3. Intersection Angle:**

- We impose a stronger condition than required: we enforce that the angle between diagonals $AQ$ and $DR$ is $\leq 90°$ (which implies $\leq 120°$). This corresponds to the condition $\vec{AQ} \cdot \vec{DR} \geq 0$. Algebraic expansion confirms this is equivalent to inequality $I_3$.

PART 2: ORTHOGONAL CONVEXITY CHECK

A set defined by the intersection of inequalities is orthogonally convex if, for each defining function $g_k(\boldsymbol{w}) \geq 0$ and each variable $w_i$, the function $g_k$ is monotonic with respect to $w_i$ (ensuring the cross-section is a connected interval). We verify this property for the system $\mathcal{S}$ using symbolic differentiation.

**Inequality $I_1$:** $g_1 = d(c + e - 1) + e(s + c)$.

- The function is linear in each variable individually. Linear functions are monotonic, satisfying the condition.

**Inequality $I_2$:** $e \geq f(c, s, d)$, where $f$ is the RHS.

- Variable $e$: Linear (monotonic).

- Variables $c, s, d$: We examine the partial derivatives of the bound $f$. Symbolic verification confirms:

$$\frac{\partial f}{\partial c} \leq 0, \quad \frac{\partial f}{\partial s} \leq 0, \quad \frac{\partial f}{\partial d} \geq 0.$$

Since the bound is monotonic in each parameter, the region defined by $e \geq f(c, s, d)$ is orthogonally convex.

**Inequality $I_3$:** $g_3 =$ LHS of $I_3$.

- Variables $s, e$: $g_3$ is linear in $s$ and $e$ (monotonic).

- Variable $c$: We compute $\frac{\partial g_3}{\partial c}$. Symbolic verification shows that $\frac{\partial g_3}{\partial c} \geq 0$ whenever $s + e \geq 1$. This condition is explicitly enforced by inequality $I_4$.

- Variable $d$: We compute $\frac{\partial g_3}{\partial d}$. Symbolic verification shows that $\frac{\partial g_3}{\partial d} \leq 0$ whenever $c \leq 1 + d + s/2 + e/2$. This condition is explicitly enforced by inequality $I_5$.

**Inequalities $I_4$ and $I_5$:**

- These are linear inequalities, which inherently define orthogonally convex regions.

**Conclusion**: The region is the intersection of sets defined by functions that are monotonic in every coordinate axis (either unconditionally or conditioned on other inequalities in the set). Therefore, the intersection is orthogonally convex. □

---

**Step 9**

**Lemma 36** (Step 9). *If $e < s + 1$, then there exists a regular point $E$ such that $|AE| \leq \max\{|AY|, |AZ|, |AR|, e + c - 1\}$.*

---

*Proof.* If $R$ is a regular point, the statement is proved. We now assume $R$ is a Steiner point. Let $RN$ denote the next edge emanating from $R$ in clockwise order. Let $W$ be the intersection of the ray $RN$ and the ray $XA$.

We proceed by case analysis.

**Case 1: Point $W$ lies on the line segment $AX$.** Applying a technique identical to that used in the proof of Lemma 8, we can show that there exists a trapped regular point within the polygon $WXYZR$. Since $|AW| \leq |AX| < |AY|$, the distance from point $A$ to this trapped point is bounded by:

$$\max\{|AY|, |AZ|, |AW|\}$$

**Case 2: Point $W$ does not lie on the line segment $AX$.** In this case, a calculation yields $|AW| = e + c - 1 > 0$ and $|RW| = s + c$. Our objective is to prove the existence of a trapped regular point within the polygon $WXYZR$.

First, we prove by contradiction that $N$ must lie on the line segment $RW$. Assume, for the sake of contradiction, that $N$ does not lie on the segment $RW$. The condition $e < s + 1$ implies $|AW| < |RW|$. This would allow for a structural modification of the Steiner Minimal Tree (SMT)—by rerouting a path through $AW$ instead of one corresponding to $RW$—that results in a shorter total length. This contradicts the minimality hypothesis of the SMT. Therefore, $N$ must lie on the line segment $RW$.

Next, we analyze the properties of point $N$: If $N$ is a regular point, our objective is met. If $N$ is a Steiner point, let $NT$ be the next edge emanating from $N$ in clockwise order.

- If $NT$ intersects $AW$, let $M$ be the point of intersection. The condition $e < s + 1$ implies $|AM| < |NM|$. By a similar argument as before, a shorter tree could be constructed, which leads to a contradiction.

- If $NT$ does not intersect $AW$, then should $T$ also be a Steiner point, the extension of the subsequent clockwise edge from $T$ must intersect either $XY$ or $YZ$. By Lemma 18, it follows that a trapped regular point, let us call it $E$, exists within the polygon $WXYZR$.

In this final scenario, the distance $|AE|$ is bounded as follows:

$$|AE| \leq \max\{|AY|, |AZ|, |AR|, |AW|\}, \quad \text{where } |AW| = e + c - 1$$

This completes the proof.

□

# F. Algorithms and Implementation Details

## F.1. Detail Algorithm for Reward Model

In this section, we present the detailed pseudocode and a comprehensive description of the core algorithm for our reward model. Given a set of verification functions $\mathcal{F}$, the algorithm will determine whether $\rho$ can be obtained based on $\mathcal{F}$.

---

**Feasibility Oracle**

**Algorithm 1** Feasibility Oracle

**Input:** Set of functions $\mathcal{F} \subset \mathcal{F}_{\text{ver}}$, parameter $\rho$.
**Output: True** if $\rho$ is feasible based on verification functions $\mathcal{F}$, **False** otherwise
    // Initialize with $2^n$ regions covering the space
1 **Initialization:** $Q \leftarrow \{\prod_{i=1}^{n} I_i \mid I_i \in \{[0,1], [1, +\infty)\}\}$
2 **while** $Q$ *is not empty* **do**
3      **if** *iteration count exceeds threshold* **then**
4          **return False**
5      Extract region $B = \prod_{i=1}^{n} [l_i, r_i]$ with $B \cap \text{Domain} \neq \emptyset$ from $Q$. Define $I_\infty \leftarrow \{i \mid r_i = +\infty\}$ and $V \leftarrow \text{Vertices}(B)$. Set $verified \leftarrow$ **False**.
6      **for** $F \in \mathcal{F}$ **do**
            // Check monotonicity conditions
7          **if** $F$ *is monotonically decreasing in all* $i \in I_\infty$ **then**
8              **if** $\max_{v \in V} F(v, \rho) \leq 0$ **then**
9                  $verified \leftarrow$ **True**
10      **if *not*** $verified$ **then**
11          $k^* \leftarrow \arg\max_i (\text{if } i \in I_\infty \text{ then } 1/l_i \text{ else } r_i - l_i)$
12          $m \leftarrow$ if $k^* \in I_\infty$ then $2 \cdot l_{k^*}$ else $(l_{k^*} + r_{k^*})/2$
13          Split $B$ along dimension $k^*$ at $m$ into sub-regions $B_{left}$ and $B_{right}$   Push $B_{left}$ and $B_{right}$ to $Q$
14 **return True**

---

The algorithm recursively partitions $\mathcal{W}$ into hyperrectangles, certifying that every region is covered by a verification function via the following steps:

1. **Initialization of Region Partitioning.** We maintain the queue $Q$ of hyperrectangles that partition the search space. We initialize $Q$ with the fundamental orthant splits to ensure the initial partition covers the entire unbounded space $\mathcal{W}$.

2. **Monotonicity Check for Unbounded Regions.** (Line 7) To handle regions extending to $+\infty$, we verify that the verification function $f$ is monotonically non-increasing along the unbounded dimension. This ensures that the maximum value is attained at the finite lower bound. The implementation details of the monotonicity check are provided in Appendix F.2.

3. **Feasibility Verification for Bounded Regions.** (Line 8) For the finite vertices, we invoke the Vertex Maximum Property. By verifying $f(\boldsymbol{w}) \leq 0$ at all vertices, we mathematically guarantee that the inequality holds for every point $\boldsymbol{w}$ within the region.

4. **Refinement via Subdivision.** (Lines 10-13) Unverified regions are subdivided along the dimension with the longest edge. For unbounded dimensions $[l, +\infty)$, we define a "logical length" of $1/l$ and apply a *geometric doubling strategy*, partitioning the interval into a bounded segment $[l, 2l]$ and a remaining unbounded tail $[2l, +\infty)$. This recursive process continues until the entire space is covered, leaving no unverified gaps.

## F.2. Monotonicity Verification Algorithm

The vertex-checking strategy of Theorem 6 is not directly applicable to unbounded parameter regions. To address this case, we must first verify the function's monotonicity along any unbounded dimensions, thereby reducing the problem to one on a bounded domain. Based on the lemmas in Appendix C.4 (Lemmas 21, 22, and 23), we define the procedure to verify $\frac{\partial F}{\partial x} \leq 0$, which is presented in Algorithm 2.

**Monotonicity Check Procedure**

---

**Algorithm 2** Monotonicity Check Procedure

**Input:** Split $\tau = (V^*, S^-, S^+, t^*)$, Edge $e$ with length parameter $x$
**Output:** **True** if $\frac{\partial F_\tau}{\partial x} \leq 0$ is guaranteed; **False** otherwise
`// 1.  Contribution from removed structure` $S^-$ `(Negative term)`
1  $\delta_{S^-} \leftarrow \mathbb{I}(e \in S^-)$
`// 2.  Contribution from connector` $t^*$ `(Positive term)`
2  $N_{t^*} \leftarrow \sum_{(u,v) \in t^*} \mathbb{I}(e$ lies on the path between $u$ and $v)$
`// 3.  Contribution from the component` $S^+$ `(Positive term)`
3  $\delta_{S^+} \leftarrow 0$
4  **for** *each pair of distinct terminals* $(u, v)$ *in* $S^+$ **do**
5     **if** *e lies on the path between $u$ and $v$* **then**
6         $\delta_{S^+} \leftarrow 1$
`// Verification Logic`
7  **if** $\delta_{S^-} + N_{t^*} + \delta_{S^+} = 0$ **then**
8     **return True** ;                        `// Variable` $x$ `is not involved in F`
9  **if** $N_{t^*} + \delta_{S^+} \leq 1 \wedge \delta_{S^-} = 1$ **then**
10    **return True**
11 **else**
12    **return False**

---

Here, $\delta_{S^-}$ indicates whether $e$ belongs to $S^-$, $N_{t^*}$ is the upper bound in Lemma 21, and $\delta_{S^+}$ indicates whether $e$ lies on the path between some pair of distinct terminals in $S^+$. These quantities are then combined to certify whether $\frac{\partial F}{\partial x} \leq 0$ is guaranteed.

**Proposition 37** (Correctness of Algorithm 2). *Suppose that either*

1. *$S^+$ connects at most three terminals, or*

2. *the parameter under consideration is $x = f$.*

*If Algorithm 2 returns **True**, then*

$$\frac{\partial F_\tau}{\partial x} \leq 0.$$

*Proof.* We assume $\rho \leq 1$, which is valid for the Steiner ratio problem. Recall the derivative of the splitting function:

$$\frac{\partial F_\tau}{\partial x} = \rho \frac{\partial L_{t^*}}{\partial x} + \frac{\partial L_{S^+}}{\partial x} - \frac{\partial L_{S^-}}{\partial x}.$$

We analyze the bounds using the variables computed in Algorithm 2:

- Since $S^-$ is a fixed tree topology and $x$ is the length of edge $e$, $\frac{\partial L_{S^-}}{\partial x} = \delta_{S^-}$.

- By Lemma 21, $\frac{\partial L_{t^*}}{\partial x} \leq N_{t^*}$.

- For the $S^+$ term, if $\delta_{S^+} = 0$, then $e$ lies on no path between any pair of terminals in $S^+$, so $L_{S^+}$ is independent of $x$ and $\frac{\partial L_{S^+}}{\partial x} = 0$. If $\delta_{S^+} = 1$, then $\frac{\partial L_{S^+}}{\partial x} \leq 1$ follows from Lemma 22 when $S^+$ connects at most three terminals, and from Lemma 23 when $x = f$. Therefore, in either case, $\frac{\partial L_{S^+}}{\partial x} \leq \delta_{S^+}$.

Substituting these into the derivative:

$$\frac{\partial F_\tau}{\partial x} \leq \rho \cdot N_{t^*} + \delta_{S^+} - \delta_{S^-} \leq N_{t^*} + \delta_{S^+} - \delta_{S^-}. \tag{5}$$

Whenever Algorithm 2 returns **True**, either $N_{t^*} + \delta_{S^+} \leq 1$ and $\delta_{S^-} = 1$, or $\delta_{S^-} = N_{t^*} = \delta_{S^+} = 0$. In either case, equation 5 implies

$$\frac{\partial F_\tau}{\partial x} \leq 0.$$

Thus, Algorithm 2 provides a rigorous sufficient condition for monotonicity. $\qquad\square$

## F.3. Strategies for Handling Complex Splits

### F.3.1. METHODS FOR HANDLING SPLITS WITH IMPLICIT REGULAR POINTS

When a split involves implicit regular points and utilizes edges connected to them (referred to as implicit edges), the splitting function cannot be evaluated directly, since its value depends on the locations of these implicit regular points. We therefore analyze instead the monotonicity of the verification functions generated by the Trapped Regular Point Lemma. We only perform monotonicity checks for the baseline verification functions; verification functions generated by other lemmas are not regarded as monotone.

For a baseline verification function, Algorithm 2 is applied only to variables with respect to which the implicit edge lengths are constant. For these variables, Proposition 37 remains valid. If an implicit edge length depends on a variable, we do not regard the corresponding verification function as monotone with respect to that variable. Under the settings in Section D, the variables with respect to which each implicit edge length is not constant are listed in Table 4.

*Table 4.* Variables with respect to which each implicit edge length is not constant.

| Implicit edge | Nonconstant variables |
| --- | --- |
| $(A, E)$ | $c, s$ |
| $(V, F)$ | $v, d, c, f$ |
| $(U, E)$ | $c, d, s, u, e$ |
| $(V, E)$ | $c, d, s, v, e$ |

When $D$ is a Steiner point

| Implicit edge | Nonconstant variables |
| --- | --- |
| $(A, E)$ | $c, s$ |
| $(D, F)$ | $d, c, f$ |
| $(D, E)$ | $c, d, s, e$ |

When $D$ is a regular point

This conservative strategy is justified by the following observation: the verification bottlenecks (i.e., regions where the feasible $\rho$ is minimized) typically occur near the origin of the parameter space $\mathcal{W}$. Therefore, complex splits only need to be applied to bounded regions, rather than unbounded ones. As a result, we adopt a conservative strategy regarding monotonicity for complex splits.

### F.3.2. METHODS FOR HANDLING SPLITS WITH 4-POINT STEINER TREE

When a split comprises an $S^+$ that connects four terminals, the monotonicity of variable $f$ can be verified using Algorithm 2, as established in Proposition 37. For all other variables, monotonicity is not assumed.

This conservative strategy follows the same rationale as discussed above.

## F.4. Enumeration of Valid Splits

To generate a rich candidate set of splitting functions, we perform an algorithmic enumeration of topologically valid splits. The procedure is formally defined in Algorithm 3.

**Definitions used in Algorithm:**

- $G$: The graph representing the local Steiner substructure.

- $V_{exp}$: The set of explicit regular points (e.g., $\{A, B, D\}$ or $\{A, B, U, V\}$).

- $W$: The union of $V_{exp}$ and the boundary connection points $\{Q, R\}$.

- $V_{imp}$: The set of available implicit regular points $\{E, F\}$.

---

**Split Enumeration**

---

**Algorithm 3** Split Enumeration

**Input:** Graph $G$, Terminals $V_{exp}$, Boundary Set $W$, Implicit Points $V_{imp}$
**Output:** Final list of valid splits $\mathcal{T}_{final}$
// **Phase 1:** Pruning (Generate $S^-$)

1 Initialize $\mathcal{T}_1 \leftarrow \varnothing$
2 **for** *each non-empty subset $V^* \subseteq V_{exp}$* **do**
3      **for** *each subset of edges $S^- \subseteq E(G)$* **do**
4          **if** *all edges incident to $V^*$ in $G$ are included in $S^-$* **then**
5              Add $(V^*, S^-)$ to $\mathcal{T}_1$
     // **Phase 2:** Re-optimization (Generate $S^+$)
6 Initialize $\mathcal{T}_2 \leftarrow \varnothing$
7 **for** *each tuple $(V^*, S^-) \in \mathcal{T}_1$* **do**
8      Let $G' = (V(G), E(G) \setminus S^-)$
9      Find connected components $C_1, \ldots, C_k$ of $G'$ that contain at least one vertex from $W \setminus V^*$
     // Enumerate all combinations of representatives
10      **for** *each $(p_1, \ldots, p_k) \in V(C_1) \times \cdots \times V(C_k)$* **do**
11          $S^+ \leftarrow \text{SMT}(\{p_1, \ldots, p_k\})$
12          Add $(V^*, S^-, S^+)$ to $\mathcal{T}_2$
     // **Phase 3:** Re-connection (Generate $t^*$)
13 Initialize $\mathcal{T}_{final} \leftarrow \varnothing$
14 Let $V_{conn} = V_{exp} \cup V_{imp}$
15 **for** *each tuple $(V^*, S^-, S^+) \in \mathcal{T}_2$* **do**
16      **for** *each subgraph $t^*$ on vertices $V_{conn}$* **do**
         // Topology Check via Graph Contraction
17          Construct graph $H$ from $t^*$: Contract all vertices in $V_{conn} \setminus V^*$ into a single super-node $\Omega$
18          **if** *$H$ is a spanning tree on the set $V^* \cup \{\Omega\}$* **then**
19              Add $(V^*, S^-, S^+, t^*)$ to $\mathcal{T}_{final}$
20 **return** $\mathcal{T}_{final}$

---

The algorithm enumerates splittings in three phases.

First, it chooses every non-empty subset $V^* \subseteq V_{exp}$ and enumerates edge sets $S^-$ such $S^-$ contains all edges incident to vertices in $V^*$.

Second, let

$$G' = (V(G), E(G) \setminus S^-).$$

The algorithm identifies all connected components of $G'$ that contain at least one vertex from $W \setminus V^*$, where $W$ consists of all explicit regular points together with the boundary points. Since every path from a point in $V \setminus V_{exp}$ to the local graph $G$ enters through a boundary point, the components containing $W \setminus V^*$ capture all locally relevant residual pieces. The algorithm then chooses one representative vertex from each such component and, for each resulting tuple of representatives, takes $S^+$ to be the SMT on those vertices.

Finally, the algorithm enumerates candidate forests $t^*$ for reconnecting $V^*$ to the remaining regular points. To test whether a candidate has the required topology, all vertices in $V_{conn} \setminus V^*$ are contracted into a single super-node $\Omega$. The candidate is accepted if the resulting multigraph is a spanning tree on $V^* \cup \{\Omega\}$, with self-loops and parallel edges taken into account. This guarantees that every removed vertex is connected to the outside, while no two outside vertices are connected through $t^*$.

Thus, the algorithm systematically generates a broad set of valid splittings by exploring different choices of $V^*$, $S^-$, $S^+$, and $t^*$.

# G. LLM Agent and Prompts

## G.1. Prompt for Searching Trapped Regular Point Lemmas

### Prompt for Searching Trapped Regular Point Lemmas

**PROMPT START**

**Role:** *You are an expert in Computational Geometry and Symbolic Logic.*

**Task:** You need to derive a linear condition on three variables $(a_2, a_3, a_4)$ that guarantees a polygonal chain is "trapped" within a bounded region, assuming the chain satisfies Minimum Steiner Tree properties.

**Tools:** You have access to a `Mathematica` execution environment.

You must follow the **6 Phases** below strictly.

**PHASE 1: Geometric Definitions**

Construct the coordinates of a polygonal chain $A_0, A_1, \ldots, A_n$ and a point $D$ based on lengths $a_i$.

1. **Coordinate System:**

   - $A_0 = (0,0)$
   - $A_1 = (1,0)$ (Since $a_1 = 1$)

2. **Recursive Definition:** For $k \geq 2$, point $A_k$ is derived from $A_{k-1}$ by moving distance $a_k$ at an inclination angle of $-60 \times (k-1)$ degrees.

   - $A_2 = A_1 + a_2(\cos(-60°), \sin(-60°))$
   - $A_3 = A_2 + a_3(\cos(-120°), \sin(-120°))$
   - ... and so on.

3. **Point D:** $D$ is connected to $A_2$ with length $d$ at angle $0°$.

   - $D = A_2 + (d, 0)$

**PHASE 2: The Logical Implication Problem**

You are working with two types of geometric properties.

**1. Properties (A) - Steiner Minimality:**

- **Definition:** For any $0 \leq u < s \leq v \leq n$, the Euclidean distance satisfies $|A_u A_v| \geq a_s$.

- **Meaning:** These are the constraints that the sequence $a_i$ always satisfies.

**2. Properties (B) - The Polygon Trap:**

- **Definition:** For $0 \leq u < s \leq v < n$, let $H$ be the foot of the perpendicular from $A_u$ to the line containing $A_v A_{v+1}$.

- **The Trap:** *IF* $H$ lies on the ray $A_v A_{v+1}$ *AND* the distance $|A_u H| < a_s$, *THEN* the entire tail of the chain $(A_n)$ is trapped inside the polygon defined by vertices $A_u, A_{u+1}, \ldots, A_v, H$.

- **Meaning:** These are your goals. Proving one of these allows you to bound the distance to $A_n$.

**CRITICAL INDEX CONSTRAINT:**

For both property types $A(u,v)$ and $B(u,v)$, the index $s$ (associated with the threshold length $a_s$) **MUST** satisfy the condition

$$u < s \leq v.$$

- *Example:* For a condition involving $A_3$ and $A_4$ (i.e., $u = 3, v = 4$), the **only** valid $s$ is 4. You cannot use $a_5$.

**The Logical Goal:**
Find a condition $C(a_2, a_3, a_4)$ such that:

$$\forall a_5, a_6, \ldots (> 0): \quad [\text{Selected A-props are True}] \implies [\text{At least one Selected B-prop is True}].$$

**PHASE 3: Condition Selection Strategy (Look-Up Only)**
**CRITICAL INSTRUCTION:** Do NOT attempt to derive the algebraic expressions for conditions $A(u, v, s)$ or $B(u, v, s)$ yourself. Use the exact algebraic forms provided in the list below.

1. **Analyze the Box:** Check the provided Box values against the $B$ conditions in the list. Identify which ones are likely to hold (i.e., where the inequality is satisfied or close to satisfied).

2. **Select Constraints:** Select a small subset (e.g., 1 $B$ property and 0–3 $A$ properties) that seem most relevant to the Box.

3. **Formulate Logic:** You want to prove that the $A$ constraints prevent $a_5, a_6$ from violating the $B$ goal. It is OK that $B$ goals do not involve $a_5, a_6$, in which case you may not need $A$ properties.

**IMPORTANT Tip:** You should first check $B$ properties with $v \leq 4$ and identify if there are any that already hold. In these cases, you do not need to derive upper bound for $a_5$.
**Tips:** In order to pose upper bound on $a_5$, you can typically use $A(0, 5, 5)$.
**Variable Instantiation (WARNING):**

- The conditions in the list below contain the variable `a_s`. When you select a condition $A(u, v)$ or $B(u, v)$, you must replace `a_s` with a specific $a_k$.

- **CONSTRAINT:** You must ensure $u < k \leq v$.

- **ERROR TRAP: Do NOT** use $B(3, 4)$ with $a_5$. Since $u = 3, v = 4$, the condition $s \leq v$ requires $s \leq 4$. Using $s = 5$ is geometrically invalid.

**The Provided List (Algebraic Forms):**
```
Example: A(u, v, s) represents the (A) condition for points |A_u A_v| >= a_s
A(0, 1): 1 >= a_s
A(0, 2): 1 + a2 + a2^2 >= a_s^2
A(0, 3): 1 + a2 + a2^2 - a3 + a2*a3 + a3^2 >= a_s^2
A(0, 4): a2^2 + a3^2 + a2*(1 + a3 - a4) + a3*(-1 + a4) + (-1 + a4)^2 >= a_s^2
A(0, 5): 1 + a2^2 + a3^2 - 2*a4 + a4^2 + a2*(1 + a3 - a4 - 2*a5) + a3*(-1 + a4 -
    a5) - a5 + a4*a5 + a5^2 >= a_s^2
A(0, 6): 1 + a2^2 + a3^2 - 2*a4 + a4^2 - a5 + a4*a5 + a5^2 + a3*(-1 + a4 - a5 -
    2*a6) + a2*(1 + a3 - a4 - 2*a5 - a6) + a6 - a4*a6 + a5*a6 + a6^2 >= a_s^2
A(1, 2): a2 >= a_s
A(1, 3): a2^2 + a2*a3 + a3^2 >= a_s^2
A(1, 4): a2^2 + a3^2 + a2*(a3 - a4) + a3*a4 + a4^2 >= a_s^2
A(1, 5): a2^2 + a3^2 + a4^2 + a2*(a3 - a4 - 2*a5) + a3*(a4 - a5) + a4*a5 + a5^2
    >= a_s^2
A(1, 6): a2^2 + a3^2 + a4^2 + a4*a5 + a5^2 + a3*(a4 - a5 - 2*a6) + a2*(a3 - a4 -
    2*a5 - a6) - a4*a6 + a5*a6 + a6^2 >= a_s^2
A(2, 3): a3 >= a_s
A(2, 4): a3^2 + a3*a4 + a4^2 >= a_s^2
A(2, 5): a3^2 + a4^2 + a3*(a4 - a5) + a4*a5 + a5^2 >= a_s^2
A(2, 6): a3^2 + a4^2 + a5^2 + a3*(a4 - a5 - 2*a6) + a4*(a5 - a6) + a5*a6 + a6^2
    >= a_s^2
A(3, 4): a4 >= a_s
A(3, 5): a4^2 + a4*a5 + a5^2 >= a_s^2
A(3, 6): a4^2 + a5^2 + a4*(a5 - a6) + a5*a6 + a6^2 >= a_s^2
A(4, 5): a5 >= a_s
A(4, 6): a5^2 + a5*a6 + a6^2 >= a_s^2
A(5, 6): a6 >= a_s
```

```
B(0, 1): False
B(0, 2): (1 - a2)/2 >= 0 && ((Sqrt[3]*(1 + a2))/2 <= a_s)
B(0, 3): (2 + a2 - a3)/2 >= 0 && ((Sqrt[3]*(a2 + a3))/2 <= a_s)
B(0, 4): (1 + 2*a2 + a3 - a4)/2 >= 0 && ((Sqrt[3]*(-1 + a3 + a4))/2 <= a_s)
B(0, 5): (-1 + a2 + 2*a3 + a4 - a5)/2 >= 0 && ((Sqrt[3]*(-1 - a2 + a4 + a5))/2
    <= a_s)
B(1, 2): False
B(1, 3): (a2 - a3)/2 >= 0 && ((Sqrt[3]*(a2 + a3))/2 <= a_s)
B(1, 4): (2*a2 + a3 - a4)/2 >= 0 && ((Sqrt[3]*(a3 + a4))/2 <= a_s)
B(1, 5): (a2 + 2*a3 + a4 - a5)/2 >= 0 && ((Sqrt[3]*(-a2 + a4 + a5))/2 <= a_s)
B(2, 3): False
B(2, 4): (a3 - a4)/2 >= 0 && ((Sqrt[3]*(a3 + a4))/2 <= a_s)
B(2, 5): (2*a3 + a4 - a5)/2 >= 0 && ((Sqrt[3]*(a4 + a5))/2 <= a_s)
B(3, 4): False
B(3, 5): (a4 - a5)/2 >= 0 && ((Sqrt[3]*(a4 + a5))/2 <= a_s)
B(4, 5): False
```

**The Required Box:**
Your condition must cover this region:
{Bottleneck}

**PHASE 4: Mathematica Derivation**
Use the tool to solve the implication.
**Step 4.1: Attempt Derivation**
Run the following logic in Mathematica using the expressions copied from the list:

```
(* Define the logic: ForAll tail_vars, (Steiner_Constraints) implies (Trap_Goal)
    *)
Reduce[
  ForAll[{a5, a6},
    a5 > 0 && a6 > 0 && <Selected_A_Props>,
    <Selected_B_Props>
  ] && a2 > 0 && a3 > 0 && a4 > 0,
  {a2, a3, a4}
]
```

*Tip:* If the result is non-linear, relax it to a linear inequality that still covers the Box.
**Step 4.2: Verification**
Verify your derived condition `<Cond>`:

```
Reduce[
  !ForAll[{a5, a6}, a5>0 && a6>0 && <Selected_A_Props>, <Selected_B_Props>]
  && a2 > 0 && a3 > 0 && a4 > 0
  && <Cond>,
  {a2, a3, a4}
]
```

The output **MUST** be `False`.
**Important:** You MUST verify the implication in the **ENTIRE** domain, not in the Test Box.

**PHASE 5: Upper Bound Calculation**
If the logic in Phase 4 holds, the point $A_n$ is trapped in the polygon $A_u \ldots A_v H A_u$ defined by your selected $B$ property. You must now calculate the specific coordinates of these vertices to determine the maximum distance.

1. **Identify and Fix Vertices ($A_u \ldots A_v$):**

   - **Case $v < 5$:** The points $A_1, \ldots, A_4$ are fully determined by the input parameters $a_2, a_3, a_4$. Use their standard coordinates.
   - **Case $v = 5$:** The point $A_5$ depends on the variable $a_5$. To fix this:

- In Phase 4, you should have ensured the condition implies a bound on $a_5$. Specifically, you may **ADD another goal** to your derivation: $a_5 \leq a_2 + a_3$.
  - For the Upper Bound calculation here, assume the "worst case" where $a_5$ reaches this limit: set $a_5 = a_2 + a_3$.
  - Calculate the coordinates of $A_5$ using this substituted value.

2. **Calculate Vertex H:**

   - $H$ is the foot of the perpendicular from $A_u$ to the line defined by $A_v$ and direction $-60v°$.
   - Since $A_u$ and $A_v$ are now fixed (based on steps above), explicitly calculate the $(x, y)$ coordinates of $H$.

3. **Affine Constraint (CRITICAL):**

   - You **MUST** ensure that the coordinates $(x, y)$ of **every** vertex in the trap polygon $(A_u, \ldots, A_v, H)$ are expressed as **Affine Functions** of $a_2, a_3, a_4$.
   - *Format:* $x = C_0 + C_1 a_2 + C_2 a_3 + C_3 a_4$.
   - This ensures the upper bound function remains simple and robust.

4. **Compute Upper Bounds:**

   - $MaxDist_A = \max(|A_0 P|)$ for all $P \in \{A_u, \ldots, A_v, H\}$.
   - $MaxDist_D = \max(|DP|)$ for all $P \in \{A_u, \ldots, A_v, H\}$.

**PHASE 6: Implementation & Output Requirements**

**Detailed Derivation:** Show how you choose the $A$ / $B$ properties and how you derived the final condition.

**Upper Bound:** If $A_5$ is involved in the trapping polygon, **EXPLICITLY** state the upper bound of $a_5$.

**Mathematica Output:** Paste the code and the output confirming the `False` result for the implication checks.

**Constraints:**

1. **Linearity:** The final condition on $a_2, a_3, a_4$ must be linear.

2. **Generality:** Do not hardcode the box values. The condition must be algebraic.

3. **Safety:** You DO NOT need to include inequalities like $a_2 > 0$ at 0.

**Final Output:** Provide a C++ code block containing these exact functions:

```
// Returns true if a2, a3, a4 satisfy the derived linear condition.
bool X_cond(double a2, double a3, double a4) {
  // Your linear inequality here
}

// Returns the upper bound of dist(A0, An) based on the trap polygon vertices.
double AX_upper_bound(double a2, double a3, double a4) {
  // Return max distance to polygon vertices
}

// Returns the upper bound of dist(D, An).
double DX_upper_bound(double a2, double a3, double a4, double d) {
  // Return max distance from D to polygon vertices
}
```

**If you fail to find a valid condition after rigorous attempts, return `false` and `0.0`.**

**CRITICAL WARNING:** Do NOT attempt to do anything that violates the above rules. Feel free if you indeed fail to find a valid condition.

**— END OF PROMPT —**

## G.2. Prompt for Searching Valid 4-Point Steiner Tree Lemmas

---

**Prompt for Searching Valid 4-Point Steiner Tree Lemmas**

**PROMPT START**
**Role:** *You are an expert Computational Geometer and Mathematician.*
**Objective:** Derive a **simplified, robust condition** $P(b, c, d, s, e)$ that guarantees the existence of a specific Steiner Tree.
**Constraint:** The condition you find must be "Axis-Parallel Segment Closed" (APSC).
**Tools:** You have access to a `Mathematica` execution environment and a `Python` execution environment.

**PHASE 1: Geometric Definitions**
Construct the following points in the 2D plane based on these vector instructions.

- **A**: $(0, 0)$

- **X**: From A, move length 1 at angle $0°$.

- **Y**: From X, move length $s$ at angle $-60°$.

- **B**: From X, move length $b$ at angle $60°$.

- **D**: From Y, move length $d$ at angle $0°$.

- **P**: From Y, move length $c$ at angle $-120°$.

- **Q**: From P, move length $d$ at angle $-60°$.

- **R**: From P, move length $e$ at angle $180°$.

The domain of the variables is $b \geq 0$ && $c \geq 0$ && $d \geq 0$ && $s \geq 0$ && $e \geq 0$.
*Note: Although Points X, Y, P are used for construction, your final task focuses on the Steiner Tree for points* **{A}, {B}, {C}, {D}**.

**PHASE 2: The Existence Logic (Type {A}{B}-{C}{D})**
You are investigating the existence of an ({A}{B})-({C}{D}) type Steiner Tree. For this specific type, define the auxiliary points $U$ and $V$ as follows:

1. **Point U**: Rotate point **{B}** $60°$ counter-clockwise around **{A}**.

2. **Point V**: Rotate point **{D}** $60°$ counter-clockwise around **{C}**.

**The Exact Existence Conditions** ($E_{exact}$) **are:**

1. **Convexity:** The quadrilateral formed by {A, B, C, D} must be convex.

2. **Internal Angles:** The angles $\angle UCD$, $\angle UDC$, $\angle VAB$, and $\angle VBA$ must all be $\leq 120°$.

3. **Crossing Angle:** Let $\theta$ be the angle between diagonal vectors $\vec{AC}$ and $\vec{BD}$. The condition is $\theta \leq 120°$.

   - *Relaxation Hint:* You may relax this third condition to $\theta \leq 90°$ (which is $\vec{AC} \cdot \vec{BD} \geq 0$) if it helps simplify the math.

The condition for "convex quadrilateral" can be formulated as follows: For any three adjacent points x, y, z (in clockwise order), the cross product of vectors xy and yz is less than or equal to zero.
The condition that the angle between vectors u and v is less than or equal to 120 degrees can be expressed using the cosine rule or the dot product of vectors, namely: vectors u and v form an angle $\leq 120°$ iff

$$2 \operatorname{dot}(u, v) + (|u| \, |v|) \geq 0.$$

You may relax it to $\operatorname{dot}(u, v) \geq 0$ if possible.

---

**PHASE 3: Deriving the Relaxed Condition P**
The exact conditions above are too complex. You must find a **sufficient condition** $P(b, c, d, s, e)$ such that $P \implies E_{exact}$.
**Requirements for Condition P (APSC Property):**
**Definition:** A condition $P$ satisfies APSC if, for any line parallel to one of the coordinate axes, the set of points within the domain (variables $\geq 0$) that satisfy the condition forms a single connected interval.
**To satisfy APSC, $P$ must be the conjunction of inequalities from the following allowed categories only:**

1. **Linear or Multilinear:** Inequalities that are linear in each variable separately.

2. **Quadratic:** Inequalities in the form $Ax^2 + Bx + C \geq 0$ (or $\leq 0$), subject to monotonicity rules explained in the prompt.

3. **Functional Boundaries:** Inequalities in the form $e > f(b, c, s, d)$ or $e < f(b, c, s, d)$, where monotonicity of $f$ must be verified.

**Coverage Requirement:**
Your condition $P$ must be general (not hardcoded numbers), but it must strictly cover the following "Test Box": {Bottleneck}
**Important:** You MUST verify the APSC Property in the **ENTIRE** domain, not in the Test Box.

**PHASE 4: Verification Protocol**
You must verify your work using Mathematica logic.
**Step A:** Write Mathematica code to compute the exact expressions for the 6 existence inequalities (Convexity + Angles).
**Step B:** Define your relaxed condition `condP`.
**Step C:** Verify Implication. You must run a command equivalent to this logic (check each inequality separately):

```
Reduce[ !<ExactCondition_i> && condP && b>=0 && c>=0 && d>=0 && s>=0 && e>=0, {
    vars} ]
```

If this returns `False`, your condition is valid. If it returns `True` or an expression, your condition is invalid.
*Tips:* Reformulate $2 \operatorname{dot}(u, v) + (|u| \, |v|) \geq 0$ into $\operatorname{dot}(u, v) \geq 0 \, || \, 4 \operatorname{dot}(u, v)^2 \leq |u|^2 |v|^2$ for faster speed when verifying.
*Constraint:* Do not put all checks in one `Reduce` call. It will time out. Check each inequality individually against $P$.
**Important:** You MUST verify the implication in the **ENTIRE** domain, not in the Test Box.

**PHASE 5: Output Requirements**

1. **Detailed Derivation:** Show how you calculated coordinates and how you derived the relaxed inequalities.

2. **APSC Proof:** Provide a logical argument explaining why your condition $P$ satisfies the APSC property, specifically addressing the Linearity/Monotonicity requirements for each part of your condition.

3. **Mathematica Output:** Paste the code and the output confirming the `False` result for the implication checks.

4. **C++ Implementation:**

Provide a C++ code block containing these exact functions:

```
// Returns true if your condition $P$ is met.
bool steiner_cond(double b, double c, double d, double s, double e) {
    // Your condition here
}

// Returns the total length of the ({A}{B}-{C}{D}) type Steiner tree, which is
    exactly |UV|, the distance between U and V.
```

```
double steiner_length(double b, double c, double d, double s, double e) {
    // Return the distance between U and V
}
```

**Important:** Do NOT use the Test Box values inside the C++ code. The code must be a general algebraic formula.

**If you fail to find a valid condition after rigorous attempts, return `false` and `0.0`.**

**— END OF PROMPT —**

# H. A Simplified Version for Mathematicians

If you are interested solely in the mathematical results of this work and are not concerned with the content related to LLMs, you may proceed directly to this section. We will present the complete logical structure of the mathematical proofs in the most concise manner possible.

## H.1. Preliminaries

### H.1.1. DEFINITIONS AND NOTATIONS

We first introduce the necessary definitions and notations.

**Definition 1** (Minimum Spanning Tree). *Consider a set $V$ of $n$ points in the Euclidean plane $\mathbb{R}^2$. A spanning tree on $V$ is a connected, acyclic graph with vertex set $V$. When the length of each edge is defined as the Euclidean distance between its endpoints, a spanning tree that minimizes the total length is called a **Minimum Spanning Tree**.*

**Definition 2** (Steiner Minimal Tree). *Consider a set $V$ of $n$ points in the Euclidean plane. The shortest network interconnecting all points in $V$, where the length of each edge is measured by Euclidean distance, is necessarily a tree, referred to as a **Steiner Minimal Tree**. A Steiner Minimal Tree may contain auxiliary vertices not in $V$. These additional vertices are called **Steiner points**, while the points in $V$ are referred to as **regular points**.*

Then we formally define the Steiner ratio and the Steiner ratio conjecture.

**Definition 3** (Steiner Ratio). *The Steiner ratio is defined as the infimum of the ratio of the length of the Steiner Minimal Tree to the length of the Euclidean Minimum Spanning Tree over all finite sets of points $V$:*

$$\rho_{Steiner} = \inf_V L_S(V)/L_m(V),$$

*where $L_S(V)$ and $L_m(V)$ denote the lengths of Steiner Minimal Tree and Minimum Spanning Tree, respectively.*

**Conjecture (Gilbert-Pollak).** Gilbert & Pollak (1968) famously conjectured that this ratio is lower-bounded by: $\rho_{\text{Steiner}} = \sqrt{3}/2 \approx 0.866$.

### H.1.2. BASIC PROPERTIES

Let $V$ be a finite set of points in the Euclidean plane. We denote the lengths of the Steiner Minimal Tree (SMT) and the Minimum Spanning Tree (MST) on $V$ as $L_S(V)$ and $L_m(V)$, respectively. For an arbitrary geometric graph $G$, let $L_G$ denote the sum of the Euclidean lengths of its edges.

We first recall some fundamental geometric properties of SMTs.

**Lemma 11** (Gilbert & Pollak (1968), Basic Properties of SMTs). *Let $S$ be a Steiner Minimal Tree for a set $V$. The following properties hold:*

1. *Every Steiner point in $S$ has a degree of exactly 3, and the three incident edges meet at $120°$ angles.*

2. *If $|V| = n$, then $S$ contains at most $n - 2$ Steiner points. A tree with exactly $n - 2$ Steiner points is called a **Full Steiner Tree (FST)**.*

3. *Any SMT can be decomposed into a union of edge-disjoint Full Steiner Trees. Consequently, it suffices to consider only Full Steiner Trees when proving lower bounds on the Steiner ratio.*

### H.1.3. INDUCTION-BASED METHOD

In this section, we present the induction-based method for proving the lower bound on the Steiner ratio. We recommend readers to read Section 2.2 for a informal but more intuitive explanation of the method.

Based on Lemma 11, we restrict our analysis to Full Steiner Trees without loss of generality. A defining topological feature of a Full Steiner Tree is that all regular points (terminals) are leaves with a degree of 1. We use **induction** to establish the lower bound. The base case where $|V| \leq 4$ is shown by Pollak (1978). Thus, we may assume $|V| \geq 5$.

Let $V$ be the set of regular points, and let $S$ be $\text{SMT}(V)$. To complete the induction step, we focus on a terminal substructure of $S$. Consider a leaf $A$ of maximum depth (choosing an arbitrary point as root) in $S$. Let $X$ be the parent of $A$. Since

$A$ is at maximum depth, the sibling of $A$ must also be a terminal, denoted as $B$. Let $Y$ be the parent of $X$. The sibling component $D$ incident to $Y$ cannot have a height exceeding that of $X$; consequently, $D$ is necessarily either a single terminal or a Steiner point connected to two terminals.

This results in the two cases illustrated in Figure 10: in the left case, $D$ is a single regular point; in the right case, $D$ is a Steiner point connecting two regular points, $U$ and $V$. The notations $(R)$ and $(Q)$ denote the residual subtrees connected to the main structure at nodes $R$ and $Q$. The labels on the edges denote their Euclidean lengths. Because the Steiner ratio is scale-invariant, we normalize the scale of the tree such that the edge $AX$ has length 1 ($|AX| = 1$).

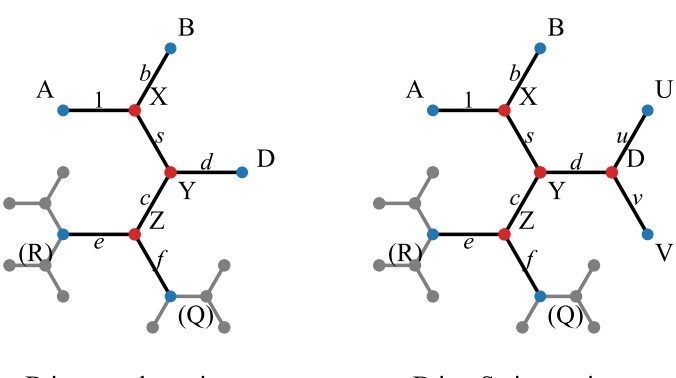

| D is a regular point | D is a Steiner point |

Figure 10. Local substructures of a Steiner Minimal Tree.

**Definition 12** (Parameter Space). *Given a local structure in Figure 10, let $n$ be the number of edges excluding the normalized edge $AX$. The **parameter space** $\mathcal{W}$ is defined as $[0, +\infty)^n$, where the $i$-th entry represents the length of the $i$-th edge. (Therefore $\boldsymbol{w} \in \mathcal{W}$ is something like $(b, c, d, s, \cdots)$.)*

Since we only consider full Steiner trees, for any $\boldsymbol{w} \in \mathcal{W}$, the positions of all points in the local structure are uniquely determined and depend affinely on $\boldsymbol{w}$.

Next, we formalize the notion of a *splitting*, which was introduced informally in Section 2.2. We then state the reduction argument of Du & Hwang (1983) in terms of this notion.

**Definition 13** (Splitting). *Given a Steiner tree $S$ for a set of points $V$, a tuple $\tau = (V^*, S^-, S^+, t^*)$ is called a **splitting** if:*

1. *$V^* \subsetneq V$ is a non-empty subset of $V$.*
2. *$S^-$ is a subset of the edges of $S$. Let $S_{rem} = S \setminus S^-$. No point in $V^*$ is incident to $S_{rem}$.*
3. *$S^+$ is a graph (may contain auxiliary vertices not in $S$) such that $S^+ \cup S_{rem}$ forms a valid Steiner tree for $V \setminus V^*$.*
4. *$t^*$ is a forest interconnecting $V^*$ and $V \setminus V^*$ using only vertices in $V$. It must satisfy: (a) For every $x \in V^*$, there exists a path in $t^*$ connecting $x$ to some $y \in V \setminus V^*$. (b) There are no paths in $t^*$ connecting any two distinct points $y_1, y_2 \in V \setminus V^*$.*

**Lemma 14** (Du & Hwang (1983) Lemma 3). *Let $S = \mathrm{SMT}(V)$ be the Steiner Minimal Tree for a set $V$ of $n$ points. Suppose that for all point sets with fewer than $n$ points, the Steiner ratio is at least $\rho'$. If there exists a splitting $\tau = (V^*, S^-, S^+, t^*)$ satisfying $(L_{S^-} - L_{S^+})/L_{t^*} \geq \rho'$, then the Steiner ratio for $V$ also satisfies $L_S(V)/L_m(V) \geq \rho'$.*

Based on Lemma 14, we reformulate the verification of a candidate lower bound $\rho$ for the Steiner ratio as a minimax condition.

**Corollary 15** (Equivalent Formulation of the Lower Bound). *For any splitting $\tau = (V^*, S^-, S^+, t^*)$ and geometric configuration $\boldsymbol{w} \in \mathcal{W}$, we define the **splitting function** $F_\tau(\boldsymbol{w}, \rho)$ as:*

$$F_\tau(\boldsymbol{w}, \rho) = \rho \cdot L_{t^*} + L_{S^+} - L_{S^-}.$$

*Let $\mathcal{F} = \{F_\tau : \tau \text{ is a splitting}\}$ denote the collection of all splitting functions. The lower bound $\rho_{Steiner} \geq \rho$ is established if:*

$$\max_{\boldsymbol{w} \in \mathcal{W}} \min_{F \in \mathcal{F}} F(\boldsymbol{w}, \rho) \leq 0. \tag{6}$$

We say that a value $\rho$ is *feasible* if the condition equation 6 holds.

Intuitively, equation 6 states that for *any* possible geometric configuration of the tree ($\max_{\boldsymbol{w}}$), there must exist *at least one* valid strategy for splitting the tree ($\min_F$) such that the splitting function is non-positive.

## H.2. Verification Function and Algorithm

### H.2.1. VERIFICATION FUNCTION

To verify the feasibility of a candidate lower bound $\rho$, we need to check the condition in Corollary 15. However, directly verifying this condition is challenging due to the high dimensionality of the parameter space $\mathcal{W}$ and the complexity of the splitting functions. To address this, we introduce a *verification function* that serves as an upper bound on the maximum of the minimum of the splitting functions.

**Definition 5** (Verification Function). *A function $f : \mathcal{W} \to \mathbb{R} \cup \{+\infty\}$ is called a **verification function** if it satisfies the following two conditions:*

1. **Shape Constraint:** *The restriction of $f$ to any line parallel to a coordinate axis is unimodal (first non-increasing and then non-decreasing).*

2. **Bounding Constraint:** *There exists a splitting function (defined in Theorem 4) $g \in \mathcal{F}$ such that $g(\boldsymbol{w}) \leq f(\boldsymbol{w})$, $\forall \boldsymbol{w} \in \mathcal{W}$.*

*We denote the set of all verification functions by $\mathcal{F}_{ver}$.*

*Remark:* The central challenge in verifying $f(\boldsymbol{w}) \leq 0$ throughout each hyperrectangle is that exhaustive point-wise checking is infeasible in a continuous domain. The following theorem addresses this challenge by establishing that the maximum value of our verification function is attained at the vertices. This property effectively reduces the verification problem to a finite set of vertex checks. We now state the theorem as follows:

**Theorem 6** (Vertex Maximum). *Let $H \subset \mathcal{W}$ be a bounded axis-aligned hyperrectangle and let $f$ be any function in $\mathcal{F}_{ver}$. A key property is that the global maximum of $f$ over $H$ is attained at one of its vertices. Consequently, if*

$$\max_{\boldsymbol{w} \in \mathcal{V}(H)} f(\boldsymbol{w}) \leq 0,$$

*then $f(\boldsymbol{w}) \leq 0$ for all $\boldsymbol{w} \in H$, which implies that the underlying splitting function $g(\boldsymbol{w}) \leq 0$. Note that $\mathcal{V}(H)$ is the set of vertices of $H$.*

We'll discuss how to construct verification functions later. For now, we first present the algorithm for verifying the lower bound using verification functions.

### H.2.2. ALGORITHM FOR VERIFYING THE LOWER BOUND

To make verification tractable, we employ a Branch-and-Bound strategy that recursively partitions the continuous parameter space $\mathcal{W}$ into smaller hyperrectangles, as illustrated in Figure 5 for the 2D case. For each hyperrectangle, our objective is to identify a single function $f \in \mathcal{F}$ that ensures $f(\boldsymbol{w}) \leq 0$ globally within the subdomain.

We present the core algorithm below, deferring the full pseudocode and detailed explanations to Appendix F.1. To determine the maximal valid lower bound, we employs a binary search, using a feasibility oracle grounded in Theorem 6. This oracle recursively partitions the continuous parameter space $\mathcal{W}$ into hyperrectangles and certifies each region via the following steps:

(1) *Region Partitioning.* We recursively partition the infinite parameter space into axis-aligned hyperrectangle regions.

(2) *Vertex Verification.* For bounded regions, we select a verification function $f$ and invoke the Vertex Maximum Property. By verifying $f(\boldsymbol{w}) \leq 0$ at all vertices, we mathematically guarantee that the inequality holds for every point $\boldsymbol{w}$ within the region.

(3) *Monotonicity Check for Unbounded Regions.* For unbounded regions, we first verify that $f$ is non-increasing along those unbounded dimensions, ensuring the maximum value occurs at the finite lower boundary (a "slice" of the region). We then apply vertex verification to this bounded slice. Details of the monotonicity check are provided in Appendix F.2.

(4) *Refinement via Subdivision.* Unverified regions are subdivided along one dimension into two halves, and recursively attempt verification.

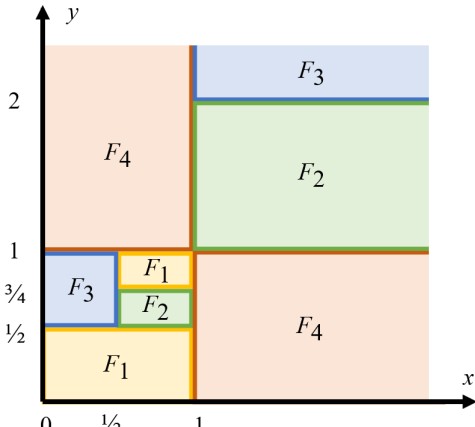

Figure 5. Branch-and-Bound on 2D Plane.

The procedure terminates when either all regions are certified within the region number threshold, in which case the candidate $\rho$ is declared *feasible*, or when the budget is exhausted with remaining uncertified regions, in which case $\rho$ is reported as *infeasible*.

### H.3. Construction of Verification Functions

In this section, we detail the construction of verification functions. We first introduce how to pre-prepare the list of splitting functions, and how to derive some 'trivial' verification functions directly from the splitting functions. Then we present the two types of geometric lemmas that form the basis for deriving more complex verification functions. Finally, we enumerate the specific geometric lemmas used in our verification process.

#### H.3.1. PRE-PREPARATION OF SPLITTING FUNCTIONS

To generate a rich candidate set of splitting functions, we perform an algorithmic enumeration of topologically valid splits. Details of this procedure are provided in Appendix F.4.

we provide the proof in Appendix C.3 that for any split where the component $S^+$ connects a set of terminals of size at most 3 (i.e., $|S^+| \leq 3$), the resulting splitting function $F(\boldsymbol{w}, \rho)$ is a convex function of the geometric parameters $\boldsymbol{w}$.

Recall the definition of the splitting function:

$$F(\boldsymbol{w}, \rho) = \rho \cdot L_{t^*}(\boldsymbol{w}) + L_{S^+}(\boldsymbol{w}) - L_{S^-}(\boldsymbol{w}).$$

We analyze the convexity of each term individually. We demonstrate that the splitting function $F(\boldsymbol{w}, \rho)$ is the sum of a convex term ($\rho L_{t^*}$), a convex term ($L_{S^+}$), and an affine term ($-L_{S^-}$). Since the sum of convex functions is convex, $F(\boldsymbol{w}, \rho)$ is convex with respect to $\boldsymbol{w}$. As these functions inherently satisfy the constraints required by Theorem 6, they serve as the initial members of our verification set $\mathcal{F}_{\text{ver}}$.

#### H.3.2. 2-TYPE OF GEOMETRIC LEMMAS

We demonstrate that finding more complex verification function reduces to identifying two types of simpler geometric lemmas: **(1) the Trapped Regular Point Lemma** and **(2) the Valid 4-Point Steiner Tree Lemma**.

**Trapped Regular Point Lemma.** Consider traversing a path on the Steiner tree. At each intermediate Steiner point, we consistently select the branch corresponding to a $60°$ clockwise turn ("turning right"). This traversal traces a maximal chain of vertices $A_0, A_1, \ldots, A_n$, where $A_n$ is a regular point and $A_1, \ldots, A_{n-1}$ are Steiner points.

Due to the geometric constraints established in Lemma 11, every internal angle $\angle A_{i-1} A_i A_{i+1}$ along this chain is exactly $120°$. We abstract this configuration as the **Steiner Spiral Chain**, as illustrated in Figure 6. For analysis, we denote the edge lengths as $a_i = |A_{i-1} A_i|$ for $i = 1, \ldots, n$, and normalize the system such that $a_1 = 1$.

Crucially, this chain is non-intersecting and constitutes the Steiner Minimal Tree for the set of vertices $\{A_0, \ldots, A_n\}$. Our

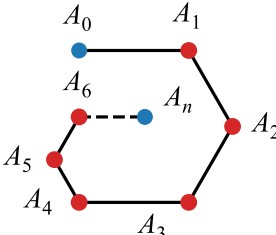

Figure 6. The Steiner Spiral Chain structure

goal is to derive conditions under which $A_n$ is trapped inside a bounded polygon.

**Valid 4-Point Steiner Tree Lemma.** Consider four points $A$, $B$, $C$, and $D$ forming a convex quadrilateral in clockwise order. We focus on the $(AB)$-$(CD)$ topology, defined as follows:

**Definition 10** (Steiner Tree Type). *The $(AB)$-$(CD)$ type Steiner tree consists of two Steiner points $S$ and $T$, and five edges: $(A,S), (B,S), (S,T), (T,C), (T,D)$. The edges incident to each Steiner point meet at $120°$ angles, as illustrated in Figure 7.*

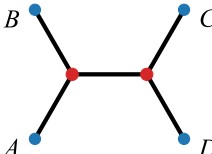

Figure 7. $(AB)$-$(CD)$ type Steiner tree.

We adopt the Valid 4-Point Steiner Tree Theorem from Du et al. (1987) to determine the existence of the $(AB)$-$(CD)$ topology. Let $U$ be the vertex of an equilateral triangle $\triangle ABU$ constructed such that $A, B, U$ appear in counterclockwise order. Similarly, let $V$ be the vertex of an equilateral triangle $\triangle CDV$ constructed such that $C, D, V$ appear in counterclockwise order. The existence of the $(AB)$-$(CD)$ topology is governed by the following conditions:

**Theorem 16** (Validity of $(AB)$-$(CD)$ Type Steiner Tree). *We say an $(AB)$-$(CD)$ type Steiner tree is valid if:*

1. *$A, B, C$ and $D$ form a convex quadrilateral.*

2. *$\max(\angle UCD, \angle UDC, \angle VAB, \angle VBA) \leq 120°$.*

3. *The intersection angle of the diagonals, $\angle AOB$, is at most $120°$ (where $O$ is the intersection of $AC$ and $BD$).*

When these conditions are met, the length of the tree follows a concise formula:

**Lemma 17** (Length of $(AB)$-$(CD)$ Type Steiner Tree). *The length of the $(AB)$-$(CD)$ type Steiner tree is precisely given by $|UV|$, where $U$ denote the position of point $B$ after segment $AB$ is rotated counterclockwise $60°$ about point $A$ and $V$ denote the position of point $D$ after segment $CD$ is rotated counterclockwise $60°$ about point $C$.*

Finally, we formally state the translation process from these geometric lemmas to the verification functions.

**Theorem 20** (From Geometric Lemmas to Verification Functions). *A series of $F \in \mathcal{F}_{ver}$ can be derived by establishing one of the following two types of lemmas:*

1. ***Trapped Regular Point Lemma:*** *A proposition asserting that when the parameters $\mathbf{w}$ satisfy certain **linear** constraints, an implicit regular point is guaranteed to lie within a specific bounded polygonal region.*

   *Formally, let $v_0$ denote the trapped regular point in the lemma. Then for each splitting $\tau = (V^*, S^-, S^+, t^*)$, we can derive a verification function $\tilde{F}_\tau$ from the splitting function $F_\tau$, as long as $t^*$ contains $v_0$ as an endpoint of an edge.*

2. ***Valid 4-Point Steiner Tree Lemma:*** *A proposition asserting that when $\mathbf{w}$ satisfies certain **orthogonally convex** conditions, a specific 4-point Steiner tree structure is valid.*

*Formally, let $X_1, X_2, X_3, X_4$ denote the 4 points in the lemma. Then for each splitting $\tau = (V^*, S^-, S^+, t^*)$, we can derive a verification function $\tilde{F}_\tau$ from the splitting function $F_\tau$, as long as $S^+ = SMT(X_1, X_2, X_3, X_4)$.*

### H.3.3. LOGICAL CRITERIA FOR THE GEOMETRIC LEMMAS

The preceding theorem reduces the construction of new verification functions to the proof of two kinds of geometric lemmas. We now record the logical criteria used to obtain such lemmas. The statements below are purely geometric: they only involve implications among constraints on the local parameters.

**A criterion for trapped regular points.** We use the notation of the Steiner spiral chain

$$A_0, A_1, \ldots, A_n,$$

where $A_0, A_n$ are regular points, $A_1, \ldots, A_{n-1}$ are Steiner points, and $a_i = |A_{i-1}A_i|$ with $a_1 = 1$.

**Lemma 7** (Type A: Intrinsic Properties) *For any $0 \leq u < s \leq v \leq n$, the inequality*

$$|A_u A_v| \geq a_s$$

*is always satisfied. We denote this condition by $C^A_{u,v,s}$.*

**Lemma 8** (Type B: Sufficient Trapping Conditions]) *For $0 \leq u < s \leq v < \min(6, n)$, let $H$ be the orthogonal projection of $A_u$ onto the line $A_v A_{v+1}$. Define $C^B_{u,v,s}$ to be the conjunction of the following two conditions:*

$$H \in \overrightarrow{A_v A_{v+1}}, \qquad |A_u H| < a_s.$$

*If $C^B_{u,v,s}$ holds, then $A_n$ is trapped inside the polygonal region*

$$A_u A_{u+1} \cdots A_v H A_u.$$

Let $\mathcal{C}^A$ and $\mathcal{C}^B$ denote the collections of all Type-A and Type-B predicates, respectively.

**Theorem 9** (Trapping criterion) *Let $\{C^A_i\}^k_{i=1} \subseteq \mathcal{C}^A$ and $\{C^B_j\}^\ell_{j=1} \subseteq \mathcal{C}^B$ be finite subfamilies. Suppose that a linear constraint $C$ on the local parameter vector $\boldsymbol{w}$ satisfies*

$$C(\boldsymbol{w}) \wedge \left( \bigwedge_{i=1}^{k} C^A_i \right) \implies \bigvee_{j=1}^{\ell} C^B_j.$$

*Then, if $\boldsymbol{w}$ satisfies $C$, the regular point $A_n$ is trapped inside the corresponding polygonal region.*

Thus, to prove a Trapped Regular Point Lemma, it suffices to exhibit a linear condition $C$ and verify the implication in Theorem 9. If $P_1, \ldots, P_\ell$ are the corresponding trapping polygons, then for any fixed point $x$ appearing in the comparison forest $t^*$ one obtains the distance bound

$$|xA_n| \leq \max_{1 \leq j \leq \ell} \max_{p \in \mathcal{V}(P_j)} |xp|,$$

where $\mathcal{V}(P_j)$ denotes the vertex set of $P_j$. This bound is then used as an explicit upper bound for the relevant edge length in $L_{t^*}$.

**A criterion for valid four-point Steiner trees.** The second class of lemmas concerns the validity of a prescribed four-point Steiner topology. Let

$$T = (X_1 X_2) - (X_3 X_4)$$

be such a topology, where the points $X_1, X_2, X_3, X_4$ are determined by the local parameters $\boldsymbol{w}$. Let $\mathcal{Q}_T(\boldsymbol{w})$ denote the conjunction of the three validity conditions in Theorem 16, applied to the ordered quadruple $(X_1, X_2, X_3, X_4)$.

Thus, to prove a Valid 4-Point Steiner Tree Lemma, it is enough to find an orthogonally convex condition $C(\boldsymbol{w})$ such that

$$C(\boldsymbol{w}) \implies \mathcal{Q}_T(\boldsymbol{w}).$$

Indeed, under this implication, Theorem 16 guarantees that the topology $T$ is valid throughout the region defined by $C$. Moreover, by Lemma 17, if $\widehat{U}_T$ and $\widehat{V}_T$ are the two equilateral vertices associated with $X_1X_2$ and $X_3X_4$, respectively, then the length of this four-point Steiner tree is $|\widehat{U}_T\widehat{V}_T|$.

Consequently, on the region $C$, the term $L_{S^+}$ in the corresponding splitting function can be evaluated by this explicit formula. By Theorem 20, such a lemma yields new verification functions whenever the reconstruction part $S^+$ of the splitting is this four-point topology.

The concrete lemmas listed below are instances of these two criteria: the trapped-point lemmas arise from implications of the form $C \wedge A_{\mathrm{sel}} \Longrightarrow B_{\mathrm{sel}}$,, whereas the four-point lemmas arise from implications of the form $C \Longrightarrow \mathcal{Q}_T$.

### H.3.4. LIST OF GEOMETRIC LEMMAS

In this section, we enumerate the geometric lemmas (generated by the LLM agent and human verified).

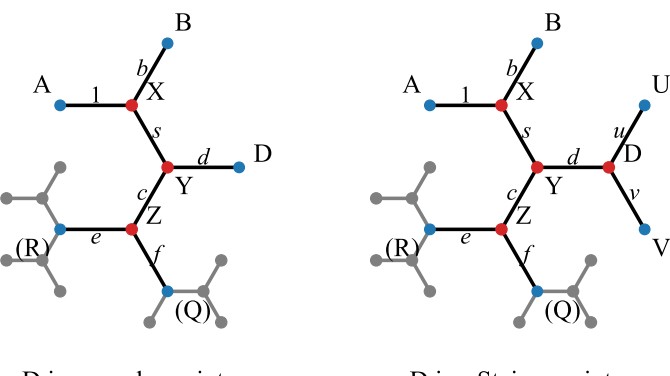

D is a regular point        D is a Steiner point

Figure 10. Two cases of the local substructure.

Edge lengths follow the notation in Figure 10. For example, $|AY| = \sqrt{1 + s + s^2}$ and $|AZ| = \sqrt{1 + s + s^2 - c + cs + c^2}$.

**Lemma 27** (Step 1, Side $A$). *If $c \leq 1$, then there exists a regular point $E$ such that $|AE| \leq \max\{|AY|, |AZ|\}$.*

**Lemma 28** (Step 1, Side $D$). *If $f \leq d$, then there exists a regular point $F$ such that*

- $|DF| \leq \max\{|DZ|, |DQ|\}$, *when $D$ is a regular point.*
- $|VF| \leq \max\{|VY|, |VZ|, |VQ|\}$, *when $D$ is a Steiner point.*

**Lemma 29** (Step 2). *If $e < 1 - c + 2/\sqrt{3} \cdot s$ and $c + e > 1$, then there exists a regular point $E$ such that $|AE| \leq \max\{1, |AY|, |AZ|, |AR|, \sqrt{3}/2 \cdot (c + e - 1)\}$.*

**Lemma 30** (Step 3). *If $e < 1 + (2/\sqrt{3} - 1) \cdot c$ and $c + e > 1$, then there exists a regular point $E$ such that $|AE| \leq \max\{1, |AY|, |AZ|, |AR|, \sqrt{3}/2 \cdot (c + e - 1)\}$.*

**Lemma 31** (Step 4). *The region defined by the system of inequalities $\mathcal{S}$ below is an orthogonally convex set. Furthermore, for any $w$ satisfying $f = d$ in this region, the $(RA)$-$(DQ)$ type Steiner tree is valid.*

$$\mathcal{S}: \begin{cases} I_1: & e + c \geq 1 \\ I_2: & e \leq s + c + d + 2 \\ I_3: & e \geq 2 - c - d + s \\ I_4: & e \leq 3s \\ I_5: & (2 + c^2 + 3d + 2s + 3cs + 3ds + 2s^2) - (2 + c + 3d + s)e \geq 0 \\ I_6: & 2(\vec{RD} \cdot \vec{AQ}) + |\vec{RD}||\vec{AQ}| \geq 0 \\ I_7: & 2(\vec{RV} \cdot \vec{RA}) + |\vec{RV}||\vec{RA}| \geq 0 \\ I_8: & 2(\vec{AV} \cdot \vec{AR}) + |\vec{AV}||\vec{AR}| \geq 0 \end{cases}$$

**Lemma 32** (Step 5). *If $c \geq 1$ and $e \geq 1$ and $f = d$, then $(AD)$-$(QR)$ type Steiner tree is valid.*

**Lemma 33** (Step 6). *If $e < (2/\sqrt{3}-1) \cdot c$, then there exists a regular point $E$ in the polygon $YZRH$, where $H$ is the orthogonal projection of $Y$ onto the ray starting at $R$ with inclination $+120°$. Therefore, $|AE| \leq \max\{|AY|, |AZ|, |AR|, |AH|\}$, and*

- *$|DE| \leq \max\{|DY|, |DZ|, |DR|, |DH|\}$, when $D$ is a regular point.*
- *$|UE| \leq \max\{|UY|, |UZ|, |UR|, |UH|\}$ and $|VE| \leq \max\{|VY|, |VZ|, |VR|, |VH|\}$, when $D$ is a Steiner point.*

**Lemma 34** (Step 7). *If $d \geq b$ and $f = d$, then $(BD)$-$(QR)$ type Steiner tree is valid.*

**Lemma 35** (Step 8). *The region defined by the system of inequalities $\mathcal{S}$ below is an orthogonally convex set. Furthermore, for any $\boldsymbol{w}$ satisfying $f = d$ in this region, the $(AD)$-$(QR)$ type Steiner tree is valid.*

$$
\mathcal{S} : \begin{cases}
I_1 : & d(c + e - 1) + e(s + c) \geq 0 \\
I_2 : & e \geq \frac{1}{2}\left(\sqrt{(c + s + 2d)^2 + 4d} - (c + s + 2d)\right) \\
I_3 : & -(1 + s/2 - c/2 + d/2)(c/2 + e + d) + \frac{3}{4}c(s + c + d) \geq 0 \\
I_4 : & s + e \geq 1 \\
I_5 : & c \leq 1 + d + s/2 + e/2
\end{cases}
$$

**Lemma 36** (Step 9). *If $e < s + 1$, then there exists a regular point $E$ such that $|AE| \leq \max\{|AY|, |AZ|, |AR|, e + c - 1\}$.*

Proofs of these lemmas are provided in Appendix E.

## H.4. Iterative Construction of the Verification Set

We now summarize how the final family of verification functions is constructed. This subsection is not needed for the logical validity of any individual geometric lemma; rather, it explains how the lemmas above are assembled into a certificate for the global lower bound.

For a finite family $\mathcal{F}_{\text{ver}}$ of verification functions and a candidate value $\rho$, write

$$
\Phi(\mathcal{F}_{\text{ver}}, \rho) := \max_{\boldsymbol{w} \in \mathcal{W}} \min_{f \in \mathcal{F}_{\text{ver}}} f(\boldsymbol{w}, \rho).
$$

By Corollary 15 and the definition of verification functions, the inequality

$$
\Phi(\mathcal{F}_{\text{ver}}, \rho) \leq 0
$$

is a sufficient certificate for the lower bound $\rho_{\text{Steiner}} \geq \rho$.

The construction starts from an initial family $\mathcal{F}_{\text{ver}}^{(0)}$, obtained from the basic splitting functions and the baseline trapped-point estimates. Suppose that after the $k$-th stage we have constructed a finite family $\mathcal{F}_{\text{ver}}^{(k)}$. The feasibility oracle described in Theorem 6 and Algorithm 1 is then applied to candidate values of $\rho$. Equivalently, it attempts to certify the statement

$$
\forall \boldsymbol{w} \in \mathcal{W}, \qquad \exists f \in \mathcal{F}_{\text{ver}}^{(k)} \text{ such that } f(\boldsymbol{w}, \rho) \leq 0.
$$

If the statement is certified for a value $\rho$, then $\rho$ is a valid lower bound with respect to the current family. One may then test a larger value of $\rho$.

If the oracle fails to certify a larger trial value, it produces a finite collection of hyperrectangles

$$
H_1, \ldots, H_m \subset \mathcal{W}
$$

which are not yet covered by the current family. We replace these uncovered regions by their smallest axis-aligned enclosing hyperrectangle

$$
\Omega_k = \prod_r \left[ \min_{1 \leq i \leq m} \inf_{\boldsymbol{w} \in H_i} w_r, \max_{1 \leq i \leq m} \sup_{\boldsymbol{w} \in H_i} w_r \right],
$$

and regard $\Omega_k$ as the current obstruction to improving the bound.

The next step is to prove additional geometric lemmas whose hypotheses cover this obstruction. Concretely, one seeks a condition $C(\boldsymbol{w})$ such that

$$\Omega_k \subseteq \{\boldsymbol{w} : C(\boldsymbol{w})\},$$

and such that one of the following two implications is valid:

$$C \wedge A_{\mathrm{sel}} \Longrightarrow B_{\mathrm{sel}}, \qquad \text{or} \qquad C \Longrightarrow \mathcal{Q}_T.$$

The first implication gives a Trapped Regular Point Lemma; the second gives a Valid 4-Point Steiner Tree Lemma. By **Theorem 20**, each newly proved lemma produces a finite collection $\Delta\mathcal{F}_k$ of additional verification functions. We then set

$$\mathcal{F}_{\mathrm{ver}}^{(k+1)} = \mathcal{F}_{\mathrm{ver}}^{(k)} \cup \Delta\mathcal{F}_k.$$

The iterative construction may therefore be summarized as

$$\mathcal{F}_{\mathrm{ver}}^{(k)} \xrightarrow{\;\rho\text{-feasibility test}\;} (\rho_k, \Omega_k) \xrightarrow{\;\text{new geometric lemmas}\;} \Delta\mathcal{F}_k \xrightarrow{\;\text{augmentation}\;} \mathcal{F}_{\mathrm{ver}}^{(k+1)}.$$

Only the final certificate is used in the proof. Namely, after finitely many stages, the accumulated family $\mathcal{F}_{\mathrm{ver}}^{(K)}$ is checked over the whole parameter space, separately in each of the cases described below. Once this final check proves

$$\Phi(\mathcal{F}_{\mathrm{ver}}^{(K)}, \rho_*) \le 0,$$

Corollary 15 gives the rigorous lower bound

$$\rho_{\mathrm{Steiner}} \ge \rho_*.$$

In our certificate, the final value is $\rho_* = 0.8559$. The intermediate boxes $\Omega_k$ only guide the construction of new lemmas; the validity of the final bound follows solely from the proved lemmas and the final covering certificate.

## H.5. Verification Procedure

Since there are two possible local substructures, and Lemma 28 determines whether $F$ exists—namely, $F$ exists when $f \le d$—we divide the verification into four cases according to the topological status of $D$ and the range of the parameter $f$:

1. **Case 1:** $D$ is a regular point, and $f \in [0, d]$ ($F$ exists).

2. **Case 2:** $D$ is a regular point, and $f \in (d, +\infty)$ ($F$ does not exist).

3. **Case 3:** $D$ is a Steiner point, and $f \in [0, d]$ ($F$ exists).

4. **Case 4:** $D$ is a Steiner point, and $f \in (d, +\infty)$ ($F$ does not exist).

**Reduction Strategy for Unbounded $f$.** For Cases 2 and 4, where $f > d$, the implicit connection $(D, F)$ (or $(U, F)$ and $(V, F)$) is invalid and thus excluded from the graph $t^*$. In these scenarios, we apply a strict filtering policy: we only admit splits whose splitting functions are **monotonically non-increasing** with respect to $f$.

Hence, in the region $f \ge d$, it suffices to verify the boundary $f = d$. For each function $F$, define $F^\partial$ as the reduced function obtained from $F$ by setting $f = d$ and removing $f$ from the input variables. The verification target is thus reduced to

$$\max_{\boldsymbol{w} \in \mathcal{W}^\partial} \min_{F \in \mathcal{F}} F^\partial(\boldsymbol{w}) \le 0,$$

where $\mathcal{W}^\partial$ denotes the reduced parameter space obtained by removing the $f$-coordinate from $\mathcal{W}$, so that $\mathcal{W}^\partial = [0, +\infty)^{n-1}$.

Note that $F^\partial$ preserves both convexity and coordinatewise non-increasingness of $F$. In particular, its non-increasingness with respect to $d$ follows because increasing $d$ simultaneously increases the original $d$-entry and the substituted $f$-entry, and $F$ is non-increasing in both variables.

It remains to handle the four cases above separately. For each case, we construct the corresponding verification functions using Lemma 27–36, with the boundary-reduced forms applied in Cases 2 and 4, and then verify the resulting inequalities via Algorithm 1. This completes the verification procedure.

The complete implementation for generating and verifying the certificates is available at `https://github.com/keyisi2006/Steiner-Ratio/tree/main/certificate`, allowing the computer-assisted verification to be checked independently.

