# OpenReview forum: "Towards Solving the Gilbert-Pollak Conjecture via Large Language Models"
_ICML.cc/2026/Conference — ICML 2026 regular_

### Official Review · Reviewer_mrHu · 2026-03-12

**Soundness:** 3
**Presentation:** 1
**Significance:** 3
**Originality:** 4
**Overall Recommendation:** 4
**Confidence:** 4

**Summary:**

We are given n points in the Euclidean plane. These points form a complete edge weighted graph by giving each edge the Euclidean distance between each edge points. A minimum spanning tree is a tree that spans the n points and whose sum of edge length is minimum among all spanning trees. In a Steiner tree, we can add additional points to the n points, so-called Steiner points. This can result in a smaller spaning tree. (Consider three points forming a triangle, the minimum spanning tree has to take two edges. We can add a Steiner point inside the triangle and then connect each point ot the Steiner point.) A Steiner minimum tree is a minimum spanning tree over all possibilities to add Steiner points to the n points. (Potentially, this could be an unbounded number of points, but it is easy to see that the number of Steiner points is always bounded by n-2.) The Gilbert-Pollak conjecture states the the ratio of the weights of a Steiner minimum tree and a minimum spanning tree, the so-called Steiner ratio, is at least \sqrt{3}/2. In the 80s, researchers tried to establish lower bounds for this ratio. In the present paper, the authors significantly improve these lower bounds using an LLM based architecture plus manual proof verification.

**Compliance With Llm Reviewing Policy:**

Affirmed.

**Key Questions For Authors:**

1) Why and to what extend are your methods superior to classical optimization techniques that one could have used. The last lower bounds are from the 80s. Computing power has advanced since then. My understanding is that one only has to study a small finite number of points. For instance, one could try to phrase this in the existential theory of the reals. Or one could try to find verification functions using classical optimization techniques.

2) Is it correct that all the lower bounds follow from considering configurations of at most 6 points or some other small finite number?

3) What can other researchers learn from your architecture? There is a little bit in the second column on page 8, but is there more.

4) Why can you start with the higher ratio of 0.8282. There are some hints on page 36, but not enough to figure it own on ones own.

Minor comments:

- page 3, Figure 4: What are the yellow points?
- page 5, line 260, left col: Why 60 degrees and not 120?
- page 6, figure bottom right: Why does the output use C++ syntax?
- page 8, Figure 9: I found this SOTA-style presentation a little ridiculous
- page 9, references: Steiner should be capitalized (several times)
- page 13, Lemma 11: Provide a reference
- page 13, line 682: define maximum depth (is there a root?)
- page 13: Figure 10 looks a little bit lost here
- page 14: It is not clear what Lemma 15 refers to.
- page 23: The section refers to an algorithm that you find on page 33
- page 24: This section again contains forward references to F.2
- page 24, line 1284: What does policy mean in the context of a proof?
- page 31, last paragraph: seems to be an artifact
- page 39/40: In the prompt, there are A/B conditions encoded. Where do they come from. Aren't we supposed to find them?

**Limitations:**

yes

**Strengths And Weaknesses:**

Soundness:

The submission appears to be technically sound to me. I think the overall framework is correct and I also verified some of the proofs by hand, but could not do all due to time constraints. However, the presentation makes it hard to follow the ideas of the paper. The proposed architecture is sound, since it did what it should do, and the output can be verified manually.

Part D of the appendix seems to contain the human verified findings by the LLM. I verified some of them, they seem to be sound. However, I do not see any proof of any new lower bound for the ratio. My current understanding is that after the proof Lemma 24, one should already be able to prove a lower bound of 0.85. But I do not see it derived anywhere. This would be extremely helpful in my opinion for the understanding of the proof.


Presentation:

I find the presentation hard to follow, and I am also missing certain things. There are a lot of forward references and more explanations of the complex matter would be helpful.

- page 4, Definition 5: Splitting function does not seem to be defined.
- page 5, Theorem 6: \mathcal{V}(H) is not defined, I guess it is the set of vertices.
- page 16/17: Here I would find it helpful to get more information how to derive the concrete A/B-type conditions from Lemma 7 and 16. (Nitpicking: Why do you start with listing the B-Type Conditions.) I guess you run over all u,v,s and replace the distances by new names.
- page 19: Theorem 18 is not properly formulated. What is the quantification of "a series". Does it have a property? Otherwise you could repeat the same function.
- Section E: I would find more explanations of the algorithm very helpful.

Significance:

There are three contributions in my opinion.

1) The improved lower bounds for the Steiner ratio. While this is impressive from the viewpoint of combinatorics and computational geometry, it is rather irrelevant for the ICML community.

2) The proof of concept that LLMs are capable to attack research level questions, even ones that are open for a long time. I think this is still impressive, we have seen a few such results but not too many so far.

3) The overall architecture for finding the proof. This is of course tailor-made to the problem. I think one can learn from this architecture if one wants to use LLMs to prove research level theorems, but this could be worked out better in the paper. Moreover, the problem benefits from the fact that the LLM only has to study a small number of points (this is at least my understanding), which makes it easier. In particular, you do not have to reason about arbitarily large graphs.

Originality:

The paper is genuinely original. It finds new mathematically proven lower bounds for an old problem from computational geometry. While the framework builds on prior work by Du and Huang, it extends this framework in a nontrivial way and then sets up a highly nontrivial architecture to use an LLM to prove new lower bounds using this framework.

---

> ### Author Rebuttal · Authors · 2026-03-31
>
> > **Q1**: Why and to what extend are your methods superior to classical optimization techniques that one could have used.
>
> **Classical methods can handle verification, but not lemma discovery.**
> Our pipeline has 2 stages: (1) *verification*—verifying a bound given a set of functions, and (2) *discovery*—searching new lemmas. Classical tools are effective for stage (1). However, stage (2) involves combining from hundreds of geometric conditions. Moreover, the selected conditions must be simplified and fit the need of Theorem 6. This selection and simplification procedure is not arbitrary traditional optimizer output. The LLM can effectively navigate this search space, and all eight new lemmas that enabled our improvements were discovered through LLM-guided search.
>
> > **Q2**: Is it correct that all the lower bounds follow from considering configurations of at most 6 points or some other small finite number?
>
> Yes, but not globally. The proof works by induction on arbitrary $n$; each inductive step only analyzes a constant-size local subtree (Fig. 10).
>
> > **Q3**: What can other researchers learn from your architecture?
>
> Two architectural insights may transfer to other problems.
>
> - **Constructing a structured reasoning space.** Rather than asking the LLM for an end-to-end proof, we decompose the search into a compositional space of lemmas. The LLM performs selection and simplification; exact tools then verify each candidate. This design plays to the LLM's strength in creative search while avoiding hallucination in formal reasoning.
> - **Targeted reflection on reasoning failures.** Verifier failures are summarized as reflection signals and fed back to the LLM. This search–verify–reflect loop may be applicable to other mathematical problems where exact verifiers exist and failures can be localized.
>
> > **Q4**: Why can you start with the higher ratio of 0.8282.
>
> The 0.8282 ratio comes from extending the local structure, exhaustive split enumeration, and replacing manual scaling  with a precise Branch-and-Bound algorithm. So this improvement is mainly due to the computational method eliminating human inaccuracy.
>
>
> **Weaknesses**
>
> > I do not see any proof of any new lower bound for the ratio ... one should already be able to prove a lower bound of 0.85. But I do not see it derived anywhere.
>
> To prove a lower bound of, e.g., 0.85, one collects all baseline verification functions (Appendix F.1) together with those derived from the LLM-discovered lemmas via Theorem 18, then runs Algorithm 1 with $\rho=0.85$. The algorithm returns True, certifying the bound. So Lemma 24 has already provided the proof for the geometric part, and the remaining part is computational certification. We will make it explicit in the revision.
>
> **Weaknesses (Presentation)**
>
> > **W1**: Definition 5: Splitting function does not seem to be defined.
>
> The splitting function is defined in Theorem 4 (l.154-156). We will add a clearer reference.
>
> > **W2**:Theorem 6: \mathcal{V}(H) is not defined, I guess it is the set of vertices.
>
> Yes, we will add the definition.
>
> > **W3**: Here I would find it helpful to get more information how to derive the concrete A/B-type conditions from Lemma 7 and 16.
>
> The A/B conditions are derived by enumerating all tuples $(u, v, s)$ and substituting the actual edge lengths (in terms of $a_i$) into the inequalities from Lemma 7 and 8. We will add a more detailed explanation in the revision.
>
> > **W4**: Theorem 18 is not properly formulated. What is the quantification of "a series".
>
> "A series" refers to the set of verification functions obtained by pairing the lemma with each splitting. We reformulate the Trapped Regular Point Lemma part as follows:
>
> Suppose a lemma establishes that when $w \in \mathcal W$ satisfies linear condition $P(w)$, the implicit regular point $E$ is guaranteed to lie within polygon $Γ$. Then for each preprocessed splitting $\tau$, if $AE \in t^\*$, we can derive a verification function $\tilde F\_\tau$ that upper bounds $F\_\tau$, by replacing $|AE|$ in $L\_{t^*}$ with $\max\_{V \in Γ} |AV|$.
>
> We will update it in the paper.
>
> > **W5**:  I would find more explanations of the algorithm very helpful.
>
> We will add more detailed explanations in the revision.
>
> **Questions in Minor Comments**
>
> - Fig 4 yellow points: auxiliary points in $S^+$.
> - 60° (l.260): exterior angle; the interior angle is 120°.
> - C++ syntax (p.6): The code on p.6 uses Python-style code for conciseness; the actual implementation uses C++ for computational efficiency.
> - Maximum depth (l.682): defined by choosing an arbitrary point as root. It can be seen that no matter which point is taken as the root, the proof will hold true.
> - "Policy" (l.1284): the lemmas in Appendix D are stated for Case 2; "policy" refers to how to translate them to other cases.
> - A/B conditions in prompt (p.39/40): derived as described above; the LLM selects and combines them, not discovers them.
> - We will also address all remaining comments in the revision, thanks!

---

> > ### Author Rebuttal · Reviewer_mrHu · 2026-04-03
> >
> > Thanks for your clarifications. I think the presentation can be improved significantly. Hence, I will keep my score, and wish you the best of luck.

---

### Official Review · Reviewer_L5Ui · 2026-03-12

**Soundness:** 3
**Presentation:** 2
**Significance:** 4
**Originality:** 4
**Overall Recommendation:** 5
**Confidence:** 3

**Summary:**

This paper considers the Gilbert-Pollak conjecture, an easily stated conjecture
in combinatorial planar geometry.  Given a set $V$ of points in the Euclidean
plain, consider first the minimum spanning tree $T$ of the set, where the cost
of the tree is the sum of the Euclidean lengths of the edges in the tree.  Then
consider the minimum cost Steiner tree $S$: this is a minimum cost spanning tree
of a potentially larger set $V \cup V'$, where the set $V'$ of *Steiner points*
is chosen to minimize the total cost of $S$.  By definition, $cost(S) \leq
cost(T)$.  Gilbert and Pollak, in the 1960s, conjetured that, in the plain, it
also holds that $cost(S) \geq (3/4)^{1/2} \cdot cost(T)$.  The number
$(3/4)^{1/2}$ is about 0.866.  The best proved bound so far, however, is about
0.824, due to Chung and Graham.


The goal of this paper is to improve this bound, and to do so by leveraging LLMs
as a guided search tool.  The paper comes up with two broad templates for
combinatorial-geometric lemmas (partly inspired from the cited work of Du and
Hwang, 1983), which can be combined in a clever way to obtain lower bounds on
the Gilbert-Pollak constant.  It then uses LLMs (experiments performed using
GPT-5 and Gemini Pro, with a claimed cost of "a few hundred dollars", l. 88),
augmented with symbolic computation tools such as Mathematica for automatically
checking and verifying their work, to generate provable instantiations of such
templates, leading in several steps to a new bound of 0.8559, thus improving on
the bound of Chung and Graham.

**Compliance With Llm Reviewing Policy:**

Affirmed.

**Final Justification:**

My original review was already quite positive.  Two main concerns regarding the paper are: (a) the readability is not optimal. Between the paper and the rebuttal by the authors, I could clear up all the things I was confused by, but it would be good to highlight definitions, global conventions/assumptions etc. in a way that a reader can easily find them.  (b) Given the timeline for ICML reviewing, it is virtually impossible to check every detail, especially of the LLM generated parts of the proof (which are the highlights of the paper).  Thus, I hope also that irrespective of whether the paper is presented at ICML, the authors would consider sending it through a round of more leisurely mathematics journal review as well.

**Key Questions For Authors:**

- As also mentioned in the supplementary material, the inductive framework
  described in Section 2.1 comes from the paper of Du and Hwang (1983, Lemma 3).
  If so, it would be good to cite this prominently in the main paper itself.

- l.1033. "Notice that both linear constraints and orthogonally convex
  conditions restrict the parameter vector w to an 1034 orthogonally convex set
  Ω." This statement is unclear.  I am not sure why a constraint such as $a_2 -
  a_3 > 0$ on the entries of $W$ leads to an orthogonally convex region.

  In general, both the statement and the proof of Theorem 18 are a bit hard to
  understand.  For example, $t^{*}$ is supposed to be a set of edges, but then
  the proof talks of the vertex $v_{0}$ belonging to it (on l. 1036).

  Similarly, should the set $\Omega$ in the proof appear in the hyotheses of
  Theorem 18?


- As mentioned above, several terms have been used without giving a full
  definition. Sometimes, this makes things hard to check.  For example, consider
  the crucial object $\mathcal{W}$, described informally as "space of
  edge-length vectors associated with the tree topology" is.  Now, e.g. on
  l.1138 in supplementary, its rather delicate properties are used (there, it is
  claimed, for example, that "$u(\mathbf{w}) − v(\mathbf{w})$ is affine in
  $\mathbf{w}$", where $\mathbf{w}$ ranges over $\mathcal{W}$).  I think this
  property holds only because of the property that in a *full* Steiner tree, all
  regular points have degree one. Further, this property seems to have been
  implicitly assumed in other places as well (e.g. in Section C.4.1, where it
  seems to be needed for $u(\mathbf{w})$ to be well-defined).  If so, please be
  explicit about this.

**Limitations:**

Yes

**Strengths And Weaknesses:**

The main strength of this paper is that it gives a fairly detailed example of
principled LLM use, combined with domain-specific mathematical insight, to make
progress on a combinatorial-geometric problem that had not seen much progress in
the last few decades, perhaps due to (as the paper also suggests several times),
the tedious verification that would have been needed to refine previous
"gadgets".  It is not clear that LLMs can be a off-the-shelf solution for such
cases; and the paper presents a very satisfying story where a "lemma template"
based framework is used to make an LLM perform guided search that is further
modulated through a "reward" structure, and access to symbolic verification
programs such as Mathematica (and its powerful $\mathsf{Reduce}$ function for
consistency checking).  While the combinatorial insights in the paper might be
special to the problem considered, I believe the paper has the potential to
inspire further progress in the use of LLMs in proving mathematical theorems via
a broadly similar path as that taken in the paper.

In terms of the weakness, I feel one of the main weaknesses is presentation.
While I did not spot any bugs in the parts I was able to check in the limited
time available for the review, several parts took much longer to understand
because of implicit conventions.  Many of these are described in the comments
below, and in the questions for authors.


Also, while the paper is generally very precise in citing previous work and
clearly delineating the new contribution, there were a couple of instances
where I felt there could be improvements:

- One caveat, that seems to have been mentioned only in the supplementary is
that a generalization (helped with computational exploration) of an idea of Du
and Hwang (1983, Lemma 3: this corresponds to the inductive framework described
in Section 2.2) already improves upon the previous best known bound (see line
1944 in the supplementary section).  Thus, it seems that if the goal is to just
get an improvement on the previous best bound, LLMs are not really needed, and
the new template framework + a little bit of computation is enough?  It would be
good to discuss this in the main paper.

- The inductive framework in Section 2.2, which underlies the whole paper, seems
  to be based on the paper of Du and Hwang, 1983. This seems to have been
  generously acknowledged in the supplementary material, but if this is indeed
  so, it would be good to mention it in the main paper as well.


Finally, given the timelines of ICML reviewing, it is hard to check all the
details of the LLM generated steps (though the paper carefully lists them all
out).  This is not really a limitation of the paper, though perhaps the paper
might be helped by a round of slower traditional mathematics journal reviewing
as well.


### Comments
- p.3: Several terms used in the statement of Theorem 4 are undefined at that
  point (and seem to be not formally defined) even in the supplement:
  "edge-length vectors associated with the tree topology", $t^{*}$ (which is
  only informally described in the preceding paragraphs), the "splitting" tuple,
  etc.  Thus, the reader has to try to deduce their meanings from the context.
  Given the centrality of this result to the development in the paper, all these
  terms should be defined explicitly, preferably in the main paper itself.

- p. 8, l. 427-428. "...more than 80% of LLM-proposed lemmas are effective when
  instantiated as verification functions." What does "effective" mean here? Does
  it mean that 80% of the lemmas are correct? Or just that 80% of the them are
  "useful" for the final proof in some subjective way?

- l.751.  It seems that for conformity with Theorem 6 of Du at al, $V$ should be defined so that $D$, $C$, $V$ are in counter-clockwise order: please check! (This should be so also because otherwise because if we follow the convention actually given in the paper, then $U$ and $V$ seem to lie on the "same side" of the picture, which is not what is needed in the argument of Lemma 15 on l. 770).

### Minor comments


- p. 12, l. 634. A double quote is typeset incorrectly.

- p. 13, Lemma 11.  Please add a reference for this.  I believe paragraphs
  3.2-3.4 and 3.6 of the original Gilbert-Pollak paper would work as a
  reference.

- l.1104. A period is missing at the end of the paragraph.

- It seems that the proof of Lemma 15 appears twice, on pages 15 and 20.

---

> ### Author Rebuttal · Authors · 2026-03-31
>
> > **Q1**: As also mentioned in the supplementary material, the inductive framework described in Section 2.1 comes from the paper of Du and Hwang (1983, Lemma 3). If so, it would be good to cite this prominently in the main paper itself.
>
> Thank you for the suggestion. We will add this citation in our revised version.
>
> > **Q2 (1)**: l.1033. "Notice that both linear constraints and orthogonally convex conditions restrict the parameter vector w to an 1034 orthogonally convex set Ω." This statement is unclear [...]  should the set $\Omega$ in the proof appear in the hyotheses of Theorem 18?
>
> $a\_2, a\_3, a\_4$ are entries of $w$ (e.g. $a\_2$ may be the first dimension of $w$), so constraints like $a\_2 - a\_3 > 0$ are linear in $w$, defining a half-space (which is orthogonally convex). In our reformulation of Theorem 18 (please see W4 in our response to Reviewer mrHu), $\Omega$ corresponds to the set satisfying $P(w)$, which already appears in the hypotheses. We will also clarify the construction of verification functions in the proof.
>
> > **Q2 (2)**: $t^*$ is supposed to be a set of edges, but then the proof talks of the vertex $v\_0$ belonging to it (on l. 1036).
>
> Thanks for the careful reading. We realize that the current writing "$v\_0$ belongs to $t^\*$" is not accurate. It should be "$v\_0$ is an endpoint of an edge in $t^\*$." We will fix this in the next version of the paper.
>
> > **Q3**: I think this property holds only because of the property that in a *full* Steiner tree, all regular points have degree one. Further, this property seems to have been implicitly assumed in other places as well.
>
> You are correct. We only consider full Steiner trees. This has been explicitly stated in Lemma 11 on l.676 and l.678. We will put it more clearly.
>
> **Weaknesses**
>
>  > Thus, it seems that if the goal is to just get an improvement ... LLMs are not really needed, and the new template framework + a little bit of computation is enough?
>
> Using a new template with only a small amount of additional computation is not sufficient to yield significant improvements. In our experiment, we can see that without LLMs, exhaustively enumerating previously known splits yields only 0.8282. The leap to 0.8559 requires discovering new geometric lemmas by selecting and combining conditions from hundreds of Type A/B candidates—a combinatorial space that is intractable for manual exploration (Section 3.2.1, l.316). The LLM navigates this space effectively, and all 8 new lemmas enabling our improvement (Appendix D) were discovered through this LLM-driven process.
>
> **Comments**
>
> > **C1**: Several terms used in the statement of Theorem 4 are undefined at that point
>
> Thank you for the careful reading. The notation in our paper is indeed quite dense. We will follow your advice and carefully double-check all details to ensure that every term is properly defined. Here, we state some of the key definitions.
>
> **Definition.** Given a full Steiner tree $S$ for a set of points $V$, a tuple $\tau=(V^*,S^-, S^+, t^\*)$ is called a **splitting** if:
>
> 1. $V^*\subsetneq V$ is a non-empty subset of $V$.
> 2. $S^-$ is a subset of the edges of $S$. Let $S\_{rem}=S\setminus S^-$. No point in $V^*$ is incident to $S_{rem}$.
> 3. $S^+$ is a graph (may contain auxiliary vertices not in $S$) such that $S^+\cup S\_{rem}$ forms a valid Steiner tree for $V\setminus V^*$.
> 4. $t^\*$ is a forest interconnecting $V^\*$ and $V\setminus V^\*$ using only vertices in $V$. It must satisfy: (a) For every $x \in V^\*$, there exists a path in $t^\*$ connecting $x$ to some $y \in V \setminus V^\*$. (b) There are no paths in $t^\*$ connecting any two distinct points $y\_1, y\_2 \in V \setminus V^*$.
>
> **Definition.** Given a local structure in Figure 10, let $n$ be the number of edges excluding $AX$. The parameter space $\mathcal W$ is defined as $[0, +\infty)^n$, where the $i$-th entry represents the length of the $i$-th edge.
>
> Since we only consider full Steiner trees (as stated in Lemma 11), given any configuration $w\in\mathcal W$, positions of all points in the structure are fixed.
>
> > **C2**: p. 8, l. 427-428. "... 80% of LLM-proposed lemmas are effective when instantiated as verification functions." What does "effective" mean here?
>
> "Effective" means both mathematically correct and helpful in improving the lower bound for the current step. We will clarify this in the revision.
>
> > **C3**: l.751. It seems that for conformity with Theorem 6 of Du at al, $V$ should be defined so that $D$, $C$, $V$ are in counter-clockwise order.
>
> We confirm that C, D, V are in counter-clockwise order, as illustrated below:
> ```
>   U
>  / \
> A---B
> |   |
> D---C
>  \ /
>   V
> ```
> However, we found a typo on l.769: "A, U, B are arranged in counter-clockwise order" should be "clockwise order." This is likely the source of confusion. We will correct it and add clearer figure annotations in the revision.

---

> > ### Author Rebuttal · Reviewer_L5Ui · 2026-04-04
> >
> > Thank you for the detailed response and the clarifications.
> >
> > Regarding Comment 3 for l. 751. Yes, my confusion indeed seems to have come from the typo on l. 768 that $A$, $U$ $B$ are arranged *counterclockwise* (as the response says, they should be *clockwise*).
> >
> > Regarding the comment about the improvement without LLMs on l. 1944 in supplementary material.  If I understand the response correctly, the author(s) agree with the comment in the review that
> >
> >   > a generalization (helped with computational exploration) of an idea of Du and Hwang (1983, Lemma 3: this corresponds to the inductive framework described in Section 2.2) already improves upon the previous best known bound (see line 1944 in the supplementary section).
> >
> > but point out that this improvement is small (from 0.8241 to 0.8282), compared to the final improvement to 0.8559.  However, I think it is still important to mention this improvement in the main paper.  I do not think mentioning it it detracts from the message of this paper.   For examples, in the hypothetical scenario where someone else had come up with the improvement to 0.8282, the improvement to 0.8282 and the final improvement to 0.8559 would presumably both be interesting results in their own right.
> >
> >
> > Given my already positive score, I am keeping it as is.  In general, I agree with some of the other reviewers that the presentation/readability of the paper could be much improved.  Further, as I said in my review, it is perhaps impossible to verify all the details of the LLM generated steps (even though the paper carefully lists them all out) within the time-frame of the ICML review period.  So I hope that the authors will consider sending the paper through a round of traditional mathematics reviewing as well, irrespective of whether the results are presented at ICML.

---

### Official Review · Reviewer_LqkH · 2026-03-13

**Soundness:** 3
**Presentation:** 3
**Significance:** 3
**Originality:** 2
**Overall Recommendation:** 4
**Confidence:** 3

**Summary:**

The submitted manuscript presents an AI system that utilizes Large Language Models (LLMs) to establish a tighter lower bound for the Steiner ratio in the Gilbert-Pollak Conjecture. The authors claim to have improved the lower bound from the long-standing 1985 benchmark of 0.824 to a new certified bound of 0.8559. Rather than prompting the LLM to solve the conjecture directly, the system tasks the model with generating rule-constrained geometric lemmas formatted as executable code. These lemmas construct "verification functions" that yield theoretically certified lower bounds. The system also employs a reflection mechanism that identifies bottleneck regions to guide the LLM's iterative refinement.

**Compliance With Llm Reviewing Policy:**

Affirmed.

**Key Questions For Authors:**

1.	In Section 3.3 under the "Translate" step, the manuscript states that proposed lemmas are "manually validated for correctness" before they are translated into verification functions. Could you quantify the exact extent of human intervention required during these iterations? Does this manual validation step represent a significant bottleneck that limits the claim of an automated theoretical discovery process?
2.	The framework successfully leverages the Valid 4-Point Steiner Tree Lemma but notes a "conservative policy" of excluding high-complexity 4-point tree lemmas in certain topological regimes because symbolic verification was not performed for variable parameters. What are the fundamental computational limits of the Mathematica verification oracle (specifically regarding Cylindrical Algebraic Decomposition) if this system were to be scaled to 5-point or larger topologies?
3.	The experiments section notes that the framework was tested with both GPT-5 and Gemini 3 Pro, concluding that the system is "robust and largely insensitive to the choice of LLM backbone". Could you provide a more granular comparative breakdown in the appendix? Specifically, it would be helpful to see the exact lemma proposal success rates, number of iterations to convergence, and token efficiency differences between the distinct models.
4.	The prompts provided in Appendix G for searching both the Trapped Regular Point Lemmas and Valid 4-Point Steiner Tree Lemmas are highly structured and mathematically prescriptive. How sensitive is the LLM agent's overall success rate to variations in the phrasing and constraints of these prompts?

**Limitations:**

No, the authors have not adequately discussed the limitations and potential negative societal impact of their work.

Constructive Suggestions for Improvement: While the paper mentions certain technical constraints throughout the text, there is no dedicated discussion synthesizing the limitations of the proposed framework. The authors should aggregate these into a clear "Limitations" section, specifically addressing the following concerns.
1. The system currently relies on human intervention, as the authors note that proposed lemmas must be "manually validated for correctness" before being translated into verification functions. Acknowledging this as a limitation to the claim of a fully "automated" discovery process is crucial.
2. The authors mention adopting a "conservative policy" of excluding high-complexity 4-point tree lemmas in certain regimes because symbolic verification was not performed for variable parameters. The authors should discuss the computational bottlenecks of their symbolic verification oracle (like Cylindrical Algebraic Decomposition) and how this limits scaling the framework to larger point topologies.
3. The current Impact Statement is overly dismissive, stating: "There are many potential societal consequences of our work, none of which we feel must be specifically highlighted here. While a paper focused on theoretical geometry may not have immediate malicious applications, the authors should replace this boilerplate with a brief, thoughtful reflection. Constructive additions could include acknowledging the broader implications of automating mathematical discovery (e.g., environmental compute costs or the dual-use nature of advanced, autonomous reasoning agents).

**Strengths And Weaknesses:**

Strength:
1.	The approach to overcoming the combinatorial explosion—a classic hurdle in theoretical computer science—is highly effective. By tasking the LLM with generating isolated, parameterized lemmas specifically, the Trapped Regular Point Lemma and Valid 4-Point Steiner Tree Lemma, instead of full end-to-end proofs, the framework significantly reduces the likelihood of hallucination.
2.	The system's reliance on a Branch-and-Bound strategy recursively partitions the continuous parameter space into hyperrectangles to make verification mathematically tractable. Furthermore, the framework offloads all mathematical validation to Mathematica for exact computation, ensuring the final lower bound stands as a standard mathematical proof independent of the generative AI process.
3.	The closed-loop reflection system smartly extracts minimal axis-aligned hyperrectangles (bottleneck regions) from failed verifications and feeds their coordinate intervals back to the LLM. This provides mathematically structured feedback, enabling more effective iterative refinement than generic scalar rewards.

Weakness:
1.	While the paper successfully handles 4-point Steiner tree topologies, the authors note a conservative policy of excluding these high-complexity lemmas in certain regimes because the symbolic verification (via Cylindrical Algebraic Decomposition) was not performed for variable parameters. The scalability of this computational bottleneck to even larger topological structures is not fully addressed.
2.	The authors briefly mention testing across different models, stating the framework is robust and insensitive to the backbone. A more detailed comparative analysis of token consumption, reasoning hours, and lemma success rates between these models would strengthen the empirical section.

---

> ### Author Rebuttal · Authors · 2026-03-31
>
> > **Q1**: Could you quantify the exact extent of human intervention required during these iterations? Does this manual validation step represent a significant bottleneck that limits the claim of an automated theoretical discovery process?
>
> Thanks for the question. The manual validation step doesn't represent a significant bottleneck in our framework. In the experiment, we used LLM to generate about 20 candidate lemmas across one proof trajectory. Human effort was limited to validating these candidates(including checking that the LLM followed prescribed constraints, faithfully executed the Mathematica verification statements). Each validation costs a few minutes for a well-educated mathematical-background student.
>
> > **Q2 & W1**: What are the fundamental computational limits [...] if this system were to be scaled to 5-point or larger topologies?
>
> The cost of Cylindrical Algebraic Decomposition (CAD) becomes a major computational bottleneck when scaling our framework to larger topologies. In our framework, certification is carried out via Branch-and-Bound with vertex verification (Theorem 4), but this procedure still depends on CAD to establish the existence conditions required for lemma proposal and verification. Expanding the local inductive structure to include more edges and points, or moving to 5-point or larger Steiner tree topologies, would significantly increase the computational burden of CAD and make lemma generation substantially more difficult. We view these challenges as important directions for future work and will discuss them in the Limitations section.
>
> > **Q3 & W2**: The experiments section notes that the framework was tested with both GPT-5 and Gemini 3 Pro, concluding that the system is "robust and largely insensitive to the choice of LLM backbone". Could you provide a more granular comparative breakdown in the appendix?
>
> Thanks for the question. We realize that saying our framework is "robust and largely insensitive to the choice of LLM backbone" is too strong and we will remove this statement. Following your suggestion, we conducted more detailed experiments to analyze token consumption, reasoning hours, and lemma success rates between GPT, Gemini and Claude models.  We'll add these results in the next version of the paper.
>
>
> |  | GPT-5 | Gemini-3.1-Pro | Claude-4.6-Opus |
> | -------- | -------- | -------- | -------- |
> |  Lemma success rate | 74% | 81% | 86% |
> |  Iterations to convergence | 16 | 9 | 7 |
> |  Reasoning hours | 6.2h | 10.2h | 2.4h |
> |  Token consumption    | 772K | 414K     | 30K     |
>
> It is worth mentioning that all the models in the table ultimately converged to good results (Steiner ratio >= 0.855). The data in the table is related to the proof path chosen by the model, so it is highly random and _not_ entirely related to the performance of the model.
>
>
> > **Q4**: [...] How sensitive is the LLM agent's overall success rate to variations in the phrasing and constraints of these prompts?
>
> Thanks for the question. According to our experience, the LLM agent's overall success rate is not very sensitive to the formulation of prompts. Across our experiments, we observed that the LLM generated reasonable outputs as long as the prompt clearly described the requirements.
>
> **Questions in Limitations**
>
> >  The authors should aggregate these into a clear "Limitations" section [...]
>
>
> Thanks for your valuable and constructive suggestions. We will follow your suggestion and add a dedicated limitations section in the next version of the paper.
>
> > The current Impact Statement is overly dismissive [...]
>
> Thanks for the suggestion. We will rewrite this section with a brief discussion covering the modest compute cost of our framework and the positive potential of automating mathematical discovery to accelerate progress on open problems.

---

> > ### Author Rebuttal · Reviewer_LqkH · 2026-04-04
> >
> > Thanks for the authors. I will keep my score

---

### Decision · Program_Chairs · 2026-04-30

**Decision:**

Accept (regular)

**Comment:**

This paper is about the conjecture that the so-called Steiner ratio of any set of points in the plane is at least sqrt(3)/2=0.866... The best known lower bound used to be 0.824, and this paper improves it to 0.8559, with a proof generated with a novel interesting LLM-based approach.

While the reviewers mention that the presentation could be improved, they agree that the idea is novel, interesting, and apparently successful. This is a clear accept.